# Statistical Numerical PDE :
# Fast Rate, Neural Scaling Law and When it's Optimal

**Yiping Lu**
Stanford University, CA
yplu@stanford.edu

**Haoxuan Chen**
Caltech, CA
haoxuan@caltech.edu

**Jianfeng Lu**
Duke University, NC
jianfeng@math.duke.edu

**Lexing Ying**
Stanford University, CA
lexing@stanford.edu

**Jose Blanchet**
Stanford University, CA
jose.blanchet@stanford.edu

## Abstract

In this paper, we study the statistical limits of deep learning techniques for solving elliptic partial differential equations (PDEs) from random samples using the Deep Ritz Method (DRM) and Physics-Informed Neural Networks (PINNs). To simplify the problem, we focus on a prototype elliptic PDE: the Schrödinger equation on a hypercube with zero Dirichlet boundary condition, which is applied in quantum-mechanical systems. We establish upper and lower bounds for both methods, which improve upon concurrently developed upper bounds for this problem via a fast rate generalization bound. We discover that the current Deep Ritz Method is sub-optimal and propose a modified version of it. We also prove that PINN and the modified version of DRM can achieve minimax optimal bounds over Sobolev spaces. Empirically, following recent work which has shown that the deep model accuracy will improve with growing training sets according to a power law, we supply computational experiments to show similar-behavior of dimension dependent power law for deep PDE solvers.

## 1 Introduction

Partial differential equations (PDEs) play a prominent role in many disciplines of science and engineering. The recent deep learning breakthrough and the rapid development of sensors, computational power, and data storage in the past decade draws attention to numerically solving PDEs via machine learning methods [36, 35, 48, 18, 57, 29], especially in high-dimension where conventional methods become impractical. Despite the success and popularity of adopting neural networks for solving high-dimensional PDEs, the following question still remain poorly answered.

> *For a given PDE and data driven approximation architecture, how large a sample size and how complex such model is needed for to reach a prescribed performance level?*

In this paper, we aim to establish the numerical analysis of such deep learning based PDE solvers. Inspired by recent works which showed that the empirical performance of a model is remarkably predictable via a power law of the data number, known as the neural scaling law [28, 22, 54], we aim to explore the neural scaling law for deep PDE solvers and compare its performance to Fourier approximation. In this work, we will focus on the deep Ritz method (DRM) [12, 29] and the Physics-Inspired Neural Networks (PINN) approach [57, 48], both are based on minimizing neural network parameters according to some loss funcitonal related to the PDEs.

35thThe Symbiosis of Deep Learning and Differential Equations (DLDE), Conference on Neural Information Processing Systems (NeurIPS 2021Workshop), Sydney, Australia.

To provide theoretically guarantees for DRM and PINN, following [38, 11, 1], we decompose the error into approximation error [68, 59, 55] and generalization error [2, 68, 67]. However instead of the $O(1/\sqrt{n})$ ($n$ is the number of data sampled) slow rate generalization bounds established in prior work [38, 55, 66, 56], we utilize the strongly convex structure of the DRM and PINN objective and provide a $O(1/n)$ fast rate generalization bound [2, 67] which lead us to a non-parametric estimation bound. Our theory also suggests optimal selection of network size with respect to the number of sampled data. Moreover, to illustrate the optimiality of our upper bound, we also establish an information-theoretic lower bound which matches our upper bound for PINN and a modified version of DRM.

## 2  Setting

For simplicity, we consider the static Schrödinger equation with zero Dirichlet boundary conditions on the domain $\Omega$, which we assume to be the unit hypercube in $\mathbb{R}^d$. Let $f \in L^2(\Omega)$, $V \in L^\infty(\Omega)$ and , $g \in L^\infty(\Omega)$. Our focus is on the analysis of Deep-Learning-based numerical methods to solve the elliptic equations

$$
\begin{aligned}
-\Delta u + V u &= f \quad \text{in } \Omega, \\
u &= g \quad \text{on } \partial\Omega.
\end{aligned}
\tag{2.1}
$$

### 2.1  Loss Functions for Solving PDEs and Induced Evaluation Metric

In this paper, we mainly focus on analysing Deep Ritz Methods (DRM) and Physics Informed Neural Network (PINN). In this subsection, we first introduce the objective function and algorithm of the two methods.

**Deep Ritz Methods**   [12, 57] Recall that the equation 2.1 is equivalent to following variational form

$$
u^* = \arg \min_{H_0^1(\Omega)} \boldsymbol{E}^{\mathrm{DRM}}(u) := \frac{1}{2} \int_\Omega |\nabla u|^2 + V|u|^2 \, dx - \int_\Omega f u \, dx,
\tag{2.2}
$$

where $u$ is minimized over $H_0^1(\Omega)$ with boundary condition given by $g$ on $\partial\Omega$.

**Physics Informed Neural Network**   [48, 57]. PINN solves 2.1 via minizing the following objective function

$$
u^* = \arg \min_u \boldsymbol{E}^{\mathrm{PINN} \in H_0^1(\Omega)}(u) := \int_\Omega |\Delta u(x) - V(x)u(x) + f(x)|^2 dx.
$$

The objective function $\boldsymbol{E}^{\mathrm{PINN}}$ can also be viewed as the population risk function and we can train an optimal estimator approximation of the solution to the PDE within a parameterized hypothesis function class $\boldsymbol{F} \subset H_0^1(\Omega)$. In this paper, we also rely on the strong convexity of the PINN objective respect to the $H_2$ norm.

### 2.2  Estimator Setting

**Empirical Loss Minimization**   In order to access the $d$-dimensional integrals, DRM and PINNemploy a Monte-Carlo method on sampled data $(X_i, f_i = f(X_i) + \eta_i)_{i=1}^n, \eta_i \sim \mathcal{N}(0, \sigma)$ for computing the high dimensional integrals, which leads to the so-called *empirical risk minimization* training for neural networks. Define the empirical losses $\boldsymbol{E}_n$ by setting

$$
\boldsymbol{E}_n^{\mathrm{DRM}}(u) = \frac{1}{n} \sum_{j=1}^n \left[ |\Omega| \cdot \left( \frac{1}{2}|\nabla u(X_j)|^2 + \frac{1}{2}V(X_j)|u(X_j)|^2 - f_j u(X_j) \right) \right],
\tag{2.3}
$$

$$
\boldsymbol{E}_n^{\mathrm{PINN}}(u) = \frac{1}{n} \sum_{j=1}^n \left[ |\Omega| \cdot \left( \Delta u(X_j) - V(X_j)u(X_j) + f_j \right)^2 \right],
\tag{2.4}
$$

where $|\Omega|$ represent the Lebesgue measure of the sets.

Once given an empirical loss $\boldsymbol{E}_n'$, we apply the empirical loss minimization to seek the estimation $u_n$, i.e. $u_n = \arg \min_{u \in \boldsymbol{F}} \boldsymbol{E}_n(u)$ where $\boldsymbol{F}$ is the parametrized hypothesis function space we consider.

For example, reproducing kernel Hilbert space[8] and tensor training format[49]. In this paper, we consider sparse neural network and truncated fourier basis, which can achieves min-max optimal estimation rate for the non-parametric function estimation[62, 52, 14, 59, 7, 27, 45].

**Neural Network Function Space**    In this paper, the hypothesis function space $\mathcal{F}$ is expressed by the neural network following [59]. Let us denote the ReLU$^3$ activation by $\eta_3(x) = \max\{x^3, 0\}$ $(x \in \mathbb{R})$ which is used in [12], and for a vector $x$, $\eta(x)$ is operated in an element-wise manner. Define the neural network with height $L$, width $W$, sparsity constraint $S$ and norm constraint $B$ as

$$\Phi(L, W, S, B) := \{(\mathcal{W}^{(L)}\eta_3(\cdot) + b^{(L)}) \circ \cdots (\mathcal{W}^{(2)}\eta_3(\cdot) + b^{(2)}) \circ (\mathcal{W}^{(1)}x + b^{(1)}) \mid$$

$$\mathcal{W}^{(L)} \in \mathbb{R}^{1 \times W}, b^{(L)} \in \mathbb{R}, \mathcal{W}^{(1)} \in \mathbb{R}^{W \times d}, b^{(1)} \in \mathbb{R}^W, \mathcal{W}^{(l)} \in \mathbb{R}^{W \times W}, b^{(l)} \in \mathbb{R}^W (1 < l < L),$$

$$\sum_{l=1}^{L}(\|\mathcal{W}^{(l)}\|_0 + \|b^{(l)}\|_0) \leq S, \max_l \|\mathcal{W}^{(l)}\|_{\infty,\infty} \vee \|b^{(l)}\|_\infty \leq B\}, \tag{2.5}$$

where $\circ$ denotes the function composition, $\|\cdot\|_0$ is the $\ell_0$-norm of the matrix (the number of non-zero elements of the matrix) and $\|\cdot\|_\infty$ is the $\ell_\infty$-norm of the matrix (maximum of the absolute values of the elements).

**Truncated Fourier Basis Estimator**    We also considered the Truncated Fourier basis as our estimator. Suppose the domain we interested $\Omega \subseteq [0,1]^d$. For any $z \in \mathbb{N}^d$, we consider the corresponding Fourier basis function $\phi_z(x) := e^{2\pi i \langle z, x \rangle}$ $(x \in \Omega)$. Any function $f \in L^2(\Omega)$ can be represented as weighted sum of the Fourier basis $f(x) := \sum_{z \in \mathbb{N}^d} f_z \phi_z(x)$ where $f_z := \int_\Omega f(x)\overline{\phi_z(x)}dx$ $(\forall z \in \mathbb{N}^d)$ is the Fourier coefficient. This inspired us to use the Fourier Basis whose index lies in a truncated set $Z_\xi = \{z \in \mathcal{Z} | \|z\|_\infty \leq \xi\}$ to represent the function class $\boldsymbol{F}$ as $\boldsymbol{F}_\xi = \{\sum_{\|z\|_\infty \leq \xi} a_z \phi_z | a_z \in \mathbb{R}, \|z\|_\infty \leq \xi\}$.

# 3   Lower Bounds

**Theorem 3.1** (Lower bound). *We denote $u^*(f)$ to be the solution of the PDE 2.1 and we can access randomly sampled data $\{X_i, f_i\}_{i=1,\cdots,n}$ as described in Section 2.2.*

**DRM Lower Bound.**    *For all estimator $H : (\mathbb{R}^d)^{\otimes n} \times \mathbb{R}^{\otimes n} \to H_\alpha(\Omega)$, we have*

$$\inf_H \sup_{u \in H_\alpha(\Omega)} \mathbb{E}\|H(\{X_i, f_i\}_{i=1,\cdots,n}) - u^*(f)\|_{H_1}^2 \gtrsim n^{-\frac{2\alpha-2}{d+2\alpha-4}}. \tag{3.1}$$

*Given that $n^{-\frac{2(\beta-k)}{d+2\beta}}$ is the minimax rate of estimation of the $k$-th derivative of a $\beta$-smooth density in $L_2$ [34, 47, 43], the lower bound have here is the rate of estimating the right hand side function $f$ in terms of the $H_{-1}$ norm. Given $H_{-1}$ norm error estimate on $f$, we can achieve estimate of $u$ with, which provides an alternative way to understand our upper bound.*

**PINN Lower Bound.**    *For all estimator $H : (\mathbb{R}^d)^{\otimes n} \times \mathbb{R}^{\otimes n} \to H_1(\Omega)$, we have*

$$\inf_H \sup_{u \in H_\alpha(\Omega)} \mathbb{E}\|H(\{X_i, f_i\}_{i=1,\cdots,n}) - u^*(f)\|_{H_2}^2 \gtrsim n^{-\frac{2\alpha-4}{d+2\alpha-4}}. \tag{3.2}$$

# 4   Upper Bounds

**Physics Informed Neural Network.**

**Theorem 4.1.** *(Informal Upper Bound of PINN with Deep Neural Network Estimator) With proper assumptions, consider the sparse Deep Neural Network function space $\Phi(L, W, S, B)$ with parameters $L = O(1)$, $W = O(n^{\frac{d}{d+2s-4}})$, $S = O(n^{\frac{d}{d+2s-4}})$, $B = O(1)$, then the Physics Informed estimator $\hat{u}_{PINN}^{DNN} = \min_{u \in \Phi(L,W,S,B)} \boldsymbol{E}_n^{PINN}(u)$ satisfies the following upper bound with high probability:*

$$\|\hat{u}_{PINN}^{DNN} - u^*\|_{H_2}^2 \lesssim n^{-\frac{2s-4}{d+2s-4}} \log n.$$

**Theorem 4.2.** *(Informal Upper Bound of PINN with Truncated Fourier Series Estimator) With proper assumptions, consider the Physics Informed Neural Network objective with a plug-in Fourier Series estimator $\hat{u}_{PINN}^{Fourier} = \min_{u \in F_\xi(\Omega)} \boldsymbol{E}_n^{PINN}(u)$ with $\xi = \Theta(n^{\frac{1}{d+2s-4}})$, then with high probability we have*

$$\|\hat{u}_{PINN}^{Fourier} - u^*\|_{H_2}^2 \lesssim n^{-\frac{2s-4}{d+2s-4}}.$$

**Deep Ritz Methods.**

**Theorem 4.3.** *(Informal Upper Bound of DRM with Truncated Fourier Series Estimator)With proper assumptions, consider the Deep Ritz objective with a plug in Fourier Series estimator $\hat{u}_{DRM}^{Fourier} = \min_{u \in F_\xi(\Omega)} \boldsymbol{E}_n^{DRM}(u)$ with $\xi = \Theta(n^{\frac{1}{d+2s-2}})$, then with high probability we have*

$$\|\hat{u}_{DRM}^{Fourier} - u^*\|_{H_1}^2 \lesssim n^{-\frac{2s-2}{d+2s-2}}.$$

**Theorem 4.4.** *(Final Upper Bound of DRM with Deep Neural Network Estimator) With proper assumptions, consider the sparse Deep Neural Network function space $\Phi(L, W, S, B)$ with parameters $L = O(1)$, $W = O(n^{\frac{d}{d+2s-2}})$, $S = O(n^{\frac{d}{d+2s-2}})$, $B = O(1)$, then the Deep ritz estimator $\hat{u}_{DRM}^{DNN} = \min_{u \in \Phi(L,W,S,B)} \boldsymbol{E}_n^{DRM}(u)$ satisfies the following upper bound with high probability:*

$$\|\hat{u}_{DRM}^{DNN} - u^*\|_{H_1}^2 \lesssim n^{-\frac{2s-2}{d+2s-2}} \log n.$$

| | Upper Bounds | | | Lower Bound |
|---|---|---|---|---|
| Objective Function | Neural Network | Previous Bound | Fourier Basis | |
| Deep Ritz | $n^{-\frac{2s-2}{d+2s-2}} \log n$ | $n^{-\frac{2s-2}{d+4s-4}} \log n$ [11] | $n^{-\frac{2s-2}{d+2s-2}}$ | $n^{-\frac{2s-2}{d+2s-4}}$ |
| Modified Deep Ritz | $n^{-\frac{2s-2}{d+2s-2}} \log n$ | / | $n^{-\frac{2s-2}{d+2s-4}}$ | $n^{-\frac{2s-2}{d+2s-4}}$ |
| PINN | $n^{-\frac{2s-4}{d+2s-4}} \log n$ | $n^{-\frac{2s-2}{d+4s-4}} \log n$ [25] | $n^{-\frac{2s-4}{d+2s-4}}$ | $n^{-\frac{2s-4}{d+2s-4}}$ |

Table 1: Upper bounds and lower bounds we achieved in this paper and previous work. The upper bound colored in red indicates the convergence rate matches the min-max lower bound.

# 5   Modified Deep Ritz Methods

Comparing the lower bound in Section 3 and the upper bound in Section 4, we find out that the Physics Informed Neural Network achieved min-max optimality while the Deep Ritz Method doesn't. In this section, we proposed a modified version of deep Ritz which can be statistically optimal.

As discussed in Appendix B, the reason behind the suboptimality of DRM comes from the high complexity introduced via the uniform concentration bound of the gradient term in the variational form. At the same time, we further observed that the $\int |\nabla u|^2 dx$ doesn't require any information of observed data, which means that we can easily make another splitted sample to approximate the $\int |\nabla u|^2 dx$ term.

$$\boldsymbol{E}_{N,n}(u) = \frac{1}{N} \sum_{j=1}^{N} \left[ |\Omega| \cdot \frac{1}{2} |\nabla u(X'_j)|^2 \right] + \frac{1}{n} \sum_{j=1}^{n} \left[ |\Omega| \cdot \left( \frac{1}{2} V(X_j)|u(X_j)|^2 - f_j u(X_j) \right) \right]$$

(5.1)

Once we sampled more data for approximating $\int |\nabla u|^2 dx$, we can achieve an near optimal bound for the Truncated Fourier Estimator when $\frac{N}{n} \gtrsim n^{\frac{2}{d+2s-4}}$.

**Theorem 5.1.** *(Informal Upper Bound of DRM with Truncated Fourier Series Estimator)With proper assumptions, consider the Deep Ritz objective with a plug in Fourier Series estimator $\hat{u}_{DRM}^{Fourier} = \min_{u \in F_\xi(\Omega)} \boldsymbol{E}_n^{DRM}(u)$ with $\xi = \Theta(n^{\frac{1}{d+2s-4}})$ and $\frac{N}{n} \gtrsim n^{\frac{2}{d+2s-4}}$, then we have*

$$\|\hat{u}_{DRM}^{Fourier} - u^*\|_{H_1}^2 \lesssim n^{-\frac{2s-2}{d+2s-4}}.$$

All our upper and lower bound is summarized in Table 1. Due to page limit, we put all the discussion in Appendix A and experiments in Appendix C.

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

# Appendix

## A Discussion

### A.1 Related Works

**Neural Scaling Law**    The starting point of our work is the recent observation across speech, vision and text [22, 28, 51, 50] that the empirical performance of a model satisfies a power law scales as a power-law with model size and dataset size. [54] further finds out that the power of the scaling law is depends on the intrinsic dimension of the dataset. Theoretical works [52, 59, 60, 7, 24, 14, 27] explore the optimal power law under the non-parametric curve estimation setting via a plug-in neural network. Our work extend this line of research into solving a PDE.

**Deep Network Based PDE Solver.**    Solving high dimensional partial differential equations (PDEs) have been a long-standing challenge due to the curse of dimensionality. At the same time, deep learning has shown superior flexibility and adaptivity in approximating high dimensional functions which leads to state-of-the-art performances in a wide range of tasks ranging from computer vision to natural language processing. Recent years, pioneer works [18, 48, 36, 57, 29] have tried to utilize the deep neural networks to solve different types of PDEs and achieves impressive results in many tasks [37, 33]. Based on the natural idea to represent solutions of PDEs by (deep) neural networks, different loss functions for solving PDEs are proposed. [18, 19] utilize the Feyman-Kac formulation which turns solving PDE to a stochastic control problem and the weak adversarial network [69] solves the weak formulation via an adversarial network. In this paper, we focus on the convergence rate of the Deep Ritz Method (DRM) [12, 29] and Physic–informed neural network(PINN) [48, 57]. DRM[12, 29] utilize the variational structure of the PDE, similar to the Ritz-Galerkin method in classical numerical analysis of PDEs, and trains the neural network to minimize the variational objective. PINN[48, 57] train the neural network directly to minimize the residual of the PDE.

**Theoretical Guarantee For Machine Learning Based PDE Solver.**    Theoretical convergence results for deep learning based PDE solvers raises wide interest recently. Specifically, [38, 15, 41, 65, 66, 56, 1] investigated the regularity of PDEs approximated by neuarl network and [38, 39] further provided a generalization analysis. [44] considered a prior and an equivalent white noise model[4] and considered the rate of convergence of the posterior. Our paper doesn't need to introduce and

provided a non-asymptotic result. At the same time, [44] can only be applied to linear PDEs while our proof technique can be extend to nonlinear one. All these paper also failed to answer the question that how to determine the network size corresponding to the sampled data number to achieve a desired statistical convergence rate. [23, 40] consider the similar problem for the optimal transport problem, *i.e.* Monge-ampere equation. However the variational problem we considered is different from[23, 40] which leads to technical difference. The most related paper to us is a concurrent paper [11, 25, 26]. However our upper is faster than [11, 25, 26] and we have shown that the lower bound conjecture in [11] is wrong. In this paper, we showed that generalization analysis in [38, 11, 39] are loose due to lack of localization technique[2, 67]. With observation of the strong convexity of the loss function, we follows fast rate results for ERM [52, 67, 14] and provided a near optimal bound for both DRM and PINN.

## A.2 Contribution

In short, we summarize our contribution as following

- In this paper, we first consider the statistical limit of learning a PDE solution from sampled observations. The lower bound showed a non-standard exponent different from non-parametric estimating a function, which breaks the conjecture listed in the concurrent work [11].

- Instead of the $O(1/\sqrt{n})$ slow rate generalization bounds in [38, 11], we utilize the strongly convex nature of the variational form and provide a fast rate generalization bound via the localization methods [63, 2, 30, 58, 67]. We showed that PINN and a modified version of DRM can achieve nearly min-max optimal convergence rate. Our result is listed in Table 1.

- We tested the recently discovered neural scaling law [22, 28, 51, 20] for deep PDE solvers numerically and the empirical results verified our theory.

## A.3 Remark on Our Upper Bound

- To theoretically understand the empirical success of Physics Informed Neural Networks and the Deep Ritz solver, in this section, we aim to prove that the excess risk $\Delta \boldsymbol{E}_n := \boldsymbol{E}(u_n) - \boldsymbol{E}(u^*)$ of a well-trained neural networks on the PINN/DRM loss function will follow a precise power-law scaling relations with the size of the training dataset. Similar to [66, 38, 11, 25, 26], we decompose the excess risk into approximation error and generalization error. Different from the concurrent bound [11, 25], we provided a fast rate $O(1/n)$ by utilizing the strong convexity of the objective function established in Section 2.1 and achieved a faster and near optimal upper bound. We showed that the generalization error can be bounded by the fixed point of the local Rademencher complexity

$$\psi(r) = R_n(\{\mathcal{I}(u) | \|u - u^*\|_A^2 \le r\}),$$

where $R_n$ is the rademencher complexity, $\mathcal{I}(u) = \Delta u + Vu, \|\cdot\|_A = \|\cdot\|_{H_2}$ for PINN and $\mathcal{I}(u) = \|\nabla u\|^2 + Vu, \|\cdot\|_A = \|\cdot\|_{H_1}$ for DRM. We put detailed definition and analysis in Appendix D.4. Then we plug in the approximation and generalization error calculated in Appendix D.3 and Appendix D.2 and finally achieved the following upper bounds.

- There is a common belief that Machine learning based PDE solvers can break the curse of dimensionality [12, 16, 31]. However we obtained an $n^{-\frac{2s-2}{2s-4+d}}$ convergence rate which can become super slow in high dimension. Our analysis showed that it's essential to constrain the function space to break the curse of dimensionality. [38] considered the DRM in Barron spaces. [46] showed that functions in the Barron space enjoy a smoothness $s$ at the same magnitude as $d$, which will also leads to convergence rate independent of the dimension using our upper bound. Neural network can also approximate mixed sparse grid spaces [42, 59], function on manifold [45, 7] without curse of dimensionality. Combined with these approximation bounds, we can also achieve a bound that breaks the curse of dimensionality using Theorem D.12 and D.9. In this paper, we aim to consider the statistical power of the loss function in common function spaces and put the curse of dimensionality as a separate topic.

- Our bound is faster than the concurrent bound [11, 25] for we provided a fast rate $O(1/n)$ by utilizing the strong convexity of the objective function. Comparing to the lower bound

provided in Section3, we show that our bound for PINN is near optimal and we'll let our bound for DRM become near optimal in the next section.

- For upper bound of DRM, due to technique issue, we assume the observation we access is clean, *i.e* $f_i = f(X_i)$. We conjecture that add noising on observation will not effect the rate and leave this to future work.

## B  Intuition Behind the Sub-optimality of the Unmodified Deep Ritz Methods

In this section, we aim to discuss the intuition behind the sub-optimality of the unmodified DRM via using the truncation Fourier basis. To simplify the notation, in this section we consider the following simplest Poisson equation $\Delta u = f$. To illustrate the necessity of the modification we made, we consider the difference between the following two estimators

- **Estimator 1.** We use the truncated Fourier basis estimator to learn the right hand side function $f$ and then we invert the PDE exactly to get the estimated $u$.

- **Estimator 2.** We plug in a parametrization of the truncated fourier basis into the empirical DRM objective

We would like to point out that *estimator 1 isn't build for computational consideration* but we use it to consider the statistical limit of our sampled data. We first show that the estimator 1 can achieve the minmax optimal estimation error.

**Error Of Estimator 1**   Firstly, we show that if one wants to learn the function $u$ in $H_1$ norm, one need to learn the right hand side function $f$ in $H^{-1}$ norm. The $H^{-1}$ norm is defined as the dual norm of the $H_1$ norm, *i.e.* $\|u\|_{H_{-1}} = \max_{\|v\|_{H_1} \leq 1} \langle u, v \rangle$. Once we assume we have an estimation $\hat{f}$ of $f$ in $H_{-1}$, we can have the estimate of $u$ via $\hat{u} := (\Delta)^{-1} \hat{f}$ in $H_1$ norm for

$$\|\nabla u - \nabla \hat{u}\|_{H_1} = \max_{\|v\|_{H_1} \leq 1} \langle \nabla u - \nabla \hat{u}, \nabla v \rangle$$
$$= \max_{\|v\|_{H_1} \leq 1} \langle \Delta u - \Delta \hat{u}, v \rangle$$
$$= \max_{\|v\|_{H_1} \leq 1} \left\langle f - \hat{f}, v \right\rangle = \|f - \hat{f}\|_{H_{-1}}.$$

Estimator 1 using the truncated fourier estimator to estimate the right hand side function $f$. Suppose we can access a random sample of observed data as $\{x_i, f(x_i)\}_{i=1}^n$, then the Fourier coefficient $f_z := \langle u, \phi_z \rangle$ can be estimated as $\hat{f}_z := \frac{1}{n} \sum_{i=1}^n f(x_i)\phi_z(x_i)$. To bound the estimation error of $\hat{f} := \sum_{\|z\|_\infty \leq Z} \hat{f}_z \phi_z$ in $H_{-1}$, we first apply the bias-variance decompoiton as

$$\mathbb{E}\|\hat{f} - f\|_{H^{-1}}^2 \leq \|\mathbb{E}\hat{f} - f\|_{H^{-1}}^2 + \mathbb{E}\|\hat{f} - \mathbb{E}f\|_{H^{-1}}^2$$

We first bound the bias term $\|\mathbb{E}\hat{f} - f\|_{H^{-1}}^2$. We know that $\mathbb{E}\hat{f} = \sum_{\|z\|_\infty \leq Z} f_z \phi_z$. Thus for a truncation set $Z$ to be of the from $\mathcal{Z} := \{z \in \mathbb{N}^d | \|z\|_\infty \leq Z\}$, the bias can be controlled by

$$\| \sum_{\|z\|_\infty > Z} f_z \phi_z \|_{H^{-1}}^2 \leq C \sum_{\|z\|_\infty > Z} f_z^2 z^{-2} \leq \|z\|^{-2(s-1)} \|f\|_{H_{\alpha-2}}^2$$

Next we estimate the variance of the estimator, we decompose the variance in to every term

$$\mathbb{E}\|f - \hat{f}\|_{H_{-1}}^2 \leq \mathbb{E} \sum_{\|z\|_\infty \leq Z} (\hat{f}_z - f_z)^2 \|\phi_z\|_{H_{-1}}^2 \leq \sum_{\|z\|_\infty \leq Z} |z|^{-1}\mathrm{Var}(\hat{f}_z).$$

Finally we achieve a $Z^{-2(s-1)} + \frac{Z^{d-2}}{n}$ upper bound for estimator 1 and with optimal selection of $Z$, we can achieve the min-max optimal convergence rate $n^{-\frac{2s-2}{d+2s-4}}$.

**Difference Between Estimator 1 and Estimator 2**  Next we aim to understand the Deep Ritz Method objective function via plugging in a truncated Fourier series estimator. We consider our estimator $u = \sum \hat{u}_z \phi_z(x)$ lies in the truncated fourier spaces. Then the empirical DRM objective function then can be expressed as

$$\frac{1}{2n} \sum_{i=1}^{n} \left( \sum_z \hat{u}_z \phi_z(x_i) \right)^2 + \sum_z \hat{u}_z \phi_z(x_i) f(x_i). \tag{B.1}$$

We observe that (B.1) is a quadratic formula respect to the fourier coffecient $\mathrm{u} := (u_z)_{\|z\|_\infty \leq Z}$ and we can reformulate it in the following matrix form

$$\min \frac{1}{2} \mathrm{u}^\top \hat{A} + \mathrm{u}^\top \hat{f}, \text{ where } \hat{A} = \left( \frac{1}{n} \sum_{i=1}^{n} \nabla \phi_i(x_i) \nabla \phi_j(x_i) \right)_{\|i\|_\infty \leq Z, \|j\|_\infty \leq Z}. \tag{B.2}$$

Based on the matrix formulation B.2 we can compare the solution for the two estimator

- **Estimator 1:** The Fourier coefficient of the solution of Estimator 1 is

$$\hat{u_1} = \mathrm{diag} \left( \|z\|_2^2 \right)_{\|z\|_\infty \leq Z}^{-1} \hat{f}. \tag{B.3}$$

- **Estimator 2:** The Fourier coefficient of the solution of Estimator 2 is

$$\hat{u_2} = \hat{A}^{-1} \hat{f}. \tag{B.4}$$

We notice that $\mathbb{E}\hat{A} = \left( \|z\|_2^2 \right)_{\|z\|_\infty \leq Z}$, thus we discovered that we further introduce another variance from the sampling of $A$. We directly estimate $\hat{u}_1 - \hat{u}_2$ and showed this term will larger than the final convergence rate. Notice that

$$\|\hat{u}_1 - \hat{u}_2\|_{H_1}^2 = f^\top \left( (\mathbb{E}\hat{A})^{-1} - \hat{A}^{-1} \right)^\top \mathrm{diag} \left( \|z\|_2^2 \right)_{\|z\|_\infty \leq Z} \left( (\mathbb{E}\hat{A})^{-1} - \hat{A}^{-1} \right) f \tag{B.5}$$

Next we aim to bound $\left( (\mathbb{E}\hat{A})^{-1} - \hat{A}^{-1} \right)$. We first use Matrix Bernstein Inequality[61] to bound the $H_1$ distance between $\hat{u}_1$ and $\hat{u}_2$. Using Matrix Bernstein Inequality, we know with high probability $1 - e^{-t}$

$$\left\| \left( (\mathbb{E}\hat{A}) - \hat{A} \right) \right\|_{\mathrm{H}} \leq \sqrt{\frac{Z^d}{n}} + \frac{t}{n}, \tag{B.6}$$

where $\| \cdot \|_{\mathrm{H}}$ is the matrix operator norm respect to the vector $\| \cdot \|_{\mathrm{H}}$ defined as $\|z\|_{\mathrm{H}}^2 = z^\top \mathrm{diag} \left( \|z\|_2^2 \right)_{\|z\|_\infty \leq Z}^{-1} z$. Note that

$$\left( I + (\mathbb{E}\hat{A})^{-1} \left( \hat{A} - (\mathbb{E}\hat{A}) \right) \right) \left( (\mathbb{E}\hat{A})^{-1} - \hat{A}^{-1} \right) = (\mathbb{E}\hat{A})^{-1} \left( \hat{A} - (\mathbb{E}\hat{A}) \right) (\mathbb{E}\hat{A})^{-1} \tag{B.7}$$

When $n$ is large enough, we know that $\frac{1}{2} I \preccurlyeq I + (\mathbb{E}\hat{A})^{-1} \left( \hat{A} - (\mathbb{E}\hat{A}) \right) \preccurlyeq I$ with high probability. Thus the term $\|\hat{u}_1 - \hat{u}_2\|_{H_1}^2$ is at the scale of $\left\| \left( (\mathbb{E}\hat{A}) - \hat{A} \right) \right\|_{\mathrm{H}}^2 \approx \frac{Z^d}{n}$, the same magnitude as what we get from the empirical process approach in our main proof and is large than the $\frac{Z^{d-2}}{n}$ variance term for $\hat{u}_1$. Thus here we conjecture that the our bound for DRM itself is tight and leads to the sub-optimal convergence rate.

## C  Experiments

In this section, we conduct several numerical experiments to verify our theory. We follow the neural network and hyper-parameter setting in [5]. Due to the page limit, we only put the experiments for Deep Ritz Methods here.

## C.1 The Modified Deep Ritz Methods

In this section, we conduct experiments which substantiate our theoretical results for modified Deep Ritz methods. For simplicity, we take $V(x) = 1$ in our experiment. We conduct experiment in 2-dimension and select the solution of the PDE as $u^* = \sum_z \|z\|^{-s} \phi_z(x) \in H^s$. We showed the Log-log plot of $H_1$ loss plotted against the number of sampled data for $s = 4$ in Figure 1. We use an OLS estimator to fit the log-log plot and put the estimated slope and corresponding R2 score in Figure 1. As our theory predicts, the modified Deep Ritz Methods convergences faster than the original one. All the derivation of the two estimators is listed in Appendix B.

## C.2 Dimension Dependent Scaling Law.

We conduct experiments to illustrate that the population loss of well-trained and well-tuned deep Ritz method will scales with the $d$-dimensional training data number $N$ as a power-law $\mathcal{L} \propto \frac{1}{N^\alpha}$. We also scans over a range of $d$ and $\alpha$ and verified an an approximately $\alpha \propto \frac{1}{d}$ scaling law as our theory suggested. We use the same test function in Section C.1 as the solution of our PDE. For simplicity, we take $V(x) = 1$ in our experiment. We trained deep Nitsche method on 20, 80, 320, 1280, 10240 sampled data for 5,6,7,8,9,10 dimensional problems and we plot our result in log–log scale. Result is shown in Figure 2. We discovered the $L \propto n^{\frac{1}{d+2}}$ scaling law in practical situations.

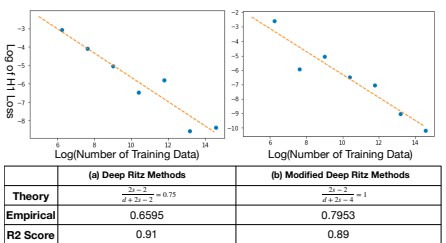

| | (a) Deep Ritz Methods | (b) Modified Deep Ritz Methods |
|---|---|---|
| Theory | $\frac{2s-2}{d+2s-2} = 0.75$ | $\frac{2s-2}{d+2s-4} = 1$ |
| Empirical | 0.6595 | 0.7953 |
| R2 Score | 0.91 | 0.89 |

Figure 1: The Log-Log plot and estimated convergence slope for Modified DRM and DRM using fourier basis, showing the median error over 5 replicates.

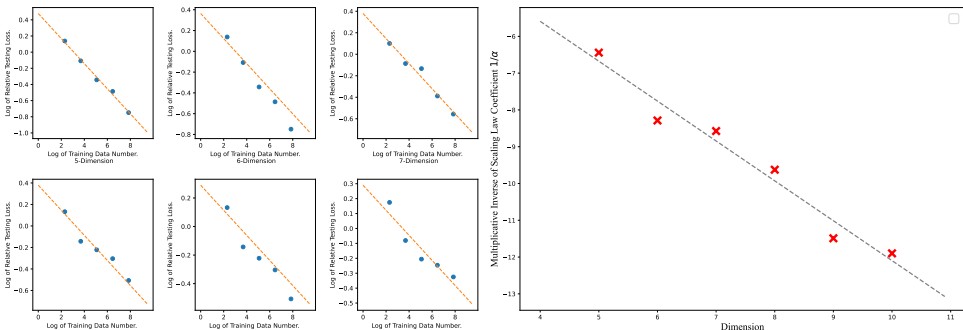

Figure 2: We verify the dimension dependent scaling law empirically and the multiplicative inverse of the scaling law coefficient is highly linear with the dimension $d$, showing the mean error over 2 replicates.

## C.3 Adapation To The Simpler Functions.

[54] have shown that the neural scaling law will adapt to the structure that the target function enjoys, this adaptivity enables the neural network to break the cure of the dimensionality for simple functions in high dimension. [60, 6] also observed this theorticly. For solving PDE, we also observed this adaptivity in practice. Here we tested the following two hypothesis

- **Random Neural Network Teacher.** Following [54], we also tested random neural network using He initialization [21] as the ground turth solution $u^*$. [10] has shown that random deep neural networks are biased towards simple functions and in practice we observed a scaling law at the parametric rate. To be specific, we obtained a linear estimate with slope $\alpha = -0.50679429$ in the log-log curve and with a $R^2$ score 0.96. See Figure3(a).

- **Simple Polynomials.** Neural network can approximate simple polynomials exponentially fast [64], thus we select the ground truth solution to be the following simple polynomial in

10 dimensional spaces,

$$u^*(x) = x_1 x_2 + \cdots + x_9 x_{10}.$$

In this example, we obtained a linear estimate with slope $\alpha = -0.49755418$ in the log-log curve and with a $R^2$ score 0.99. See Figure3(b). .

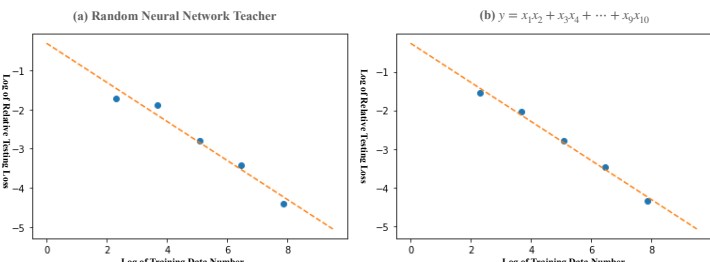

Figure 3: Neural network have the ability to adapt to simple functions and achieves convergence without curse of dimensionality, showing the median error over 5 replicates.

# D   Proof of the Upper Bounds

## D.1   Regularity Result For the PDE model.

### Regularity Results of the DRM Objective Function

**Theorem D.1.** *We consider the static Schrödinger equation on the unit hypercube on $\mathbb{R}^d$ with the zero Direchlet boundary condition:*

$$\begin{aligned} -\Delta u + Vu = f \text{ on } \Omega, \\ u = 0 \text{ on } \partial\Omega. \end{aligned} \tag{D.1}$$

*where $f \in L^2(\Omega)$ and $V \in L^\infty(\Omega)$ with $0 < V_{\min} \le V(x) \le V_{\max} > 0$. There exists a unique weak solution $u_S^*$ to the equivalent variational problem [13]:*

$$u_S^* = \operatorname*{arg\,min}_{u \in H_1^0(\Omega)} \boldsymbol{E}_S^{DRM}(u) := \operatorname*{arg\,min}_{u \in H_1^0(\Omega)} \left\{ \frac{1}{2} \int_\Omega \left[ |\nabla u|^2 + V|u|^2 \right] dx - \int_\Omega fu\,dx \right\}. \tag{D.2}$$

*Then for any $u \in H_1(\Omega)$, we have:*

$$\frac{\min(1, V_{\min})}{2} \|u - u_S^*\|_{H_1(\Omega)}^2 \le \boldsymbol{E}_S^{DRM}(u) - \boldsymbol{E}_S^{DRM}(u_S^*) \le \frac{\max(1, V_{\max})}{2} \|u - u_S^*\|_{H_1(\Omega)}^2. \tag{D.3}$$

*Proof.* To show that $u_S^*$ satisfies estimate D.3, we first claim that for any $u \in H_1(\Omega)$,

$$\boldsymbol{E}_S(u) - \boldsymbol{E}_S(u_S^*) = \frac{1}{2} \int_\Omega |\nabla u - \nabla u_S^*|^2 dx + \frac{1}{2} \int_\Omega V(u_S^* - u)^2 \, dx. \tag{D.4}$$

In fact, by plugging in the first equation of D.1, one has that

$$\begin{aligned} \boldsymbol{E}_S(u_S^*) &= \frac{1}{2} \int_\Omega |\nabla u_S^*|^2 dx + \frac{1}{2} \int_\Omega V|u_S^*|^2 dx - \int_\Omega fu_S^* dx \\ &= \frac{1}{2} \int_\Omega |\nabla u_S^*|^2 dx + \frac{1}{2} \int_\Omega V|u_S^*|^2 dx + \int_\Omega (\Delta u_S^* - Vu_S^*) u_S^* dx \\ &= \frac{1}{2} \int_\Omega |\nabla u_S^*|^2 dx + \int_\Omega (\Delta u_S^*) u_S^* dx - \frac{1}{2} \int_\Omega V|u_S^*|^2 dx. \end{aligned}$$

Furthermore, applying Green's formula to the true solution $u_S^*$ yields:

$$\boldsymbol{E}_S(u_S^*) = \frac{1}{2}\int_\Omega |\nabla u_S^*|^2 dx + \int_\Omega (\Delta u_S^*)u_S^* dx - \frac{1}{2}\int_\Omega V|u_S^*|^2 dx$$

$$= \int_{\partial\Omega} \frac{\partial u_S^*}{\partial n} u_S^* dx - \frac{1}{2}\int_\Omega |\nabla u_S^*|^2 dx - \frac{1}{2}\int_\Omega V|u_S^*|^2 dx$$

$$= -\frac{1}{2}\int_\Omega |\nabla u_S^*|^2 dx - \frac{1}{2}\int_\Omega V|u_S^*|^2 dx,$$

where the last identity above follows from the second equality in D.1. Now for any $u \in H_1(\Omega)$, applying Green's formula to $u$ and the true solution $u_S^*$ implies:

$$\boldsymbol{E}_S(u) - \boldsymbol{E}_S(u_S^*) = \frac{1}{2}\int_\Omega |\nabla u|^2 dx + \frac{1}{2}\int_\Omega V|u|^2 dx - \int_\Omega fu dx + \frac{1}{2}\int_\Omega |\nabla u_S^*|^2 dx + \frac{1}{2}\int_\Omega V|u_S^*|^2 dx$$

$$= \frac{1}{2}\int_\Omega |\nabla u|^2 dx + \frac{1}{2}\int_\Omega V|u|^2 dx + \int_\Omega (\Delta u_S^* - Vu_S^*)u dx + \frac{1}{2}\int_\Omega |\nabla u_S^*|^2 dx + \frac{1}{2}\int_\Omega V|u_S^*|^2 dx$$

$$= \frac{1}{2}\int_\Omega |\nabla u|^2 dx + \int_\Omega (\Delta u_S^*)u dx + \frac{1}{2}\int_\Omega |\nabla u_S^*|^2 dx + \frac{1}{2}\int_\Omega V(u_S^* - u)^2 dx$$

$$= \frac{1}{2}\int_\Omega |\nabla u|^2 dx + \int_{\partial\Omega} \frac{\partial u_S^*}{\partial n} u dx - \int_\Omega \nabla u_S^* \cdot \nabla u dx + \frac{1}{2}\int_\Omega |\nabla u_S^*|^2 dx + \frac{1}{2}\int_\Omega V(u_S^* - u)^2 dx$$

$$= \frac{1}{2}\int_\Omega |\nabla u - \nabla u_S^*|^2 dx + \frac{1}{2}\int_\Omega V(u_S^* - u)^2\ dx$$

where the last identity above again follows from the second equality in D.1. This completes our proof of identity D.4. Using the assumptions on the potential function $V$ then implies:

$$\boldsymbol{E}_S(u) - \boldsymbol{E}_S(u_S^*) \leq \frac{\max(1, V_{\max})}{2}\Big[\int_\Omega |\nabla u - \nabla u_S^*|^2 dx + \int_\Omega (u_S^* - u)^2\ dx\Big]$$

$$= \frac{\max(1, V_{\max})}{2}\|u - u_S^*\|_{H_1(\Omega)}^2,$$

$$\boldsymbol{E}_S(u) - \boldsymbol{E}_S(u_S^*) \geq \frac{\max(1, V_{\min})}{2}\Big[\int_\Omega |\nabla u - \nabla u_S^*|^2 dx + \int_\Omega (u_S^* - u)^2\ dx\Big]$$

$$= \frac{\max(1, V_{\min})}{2}\|u - u_S^*\|_{H_1(\Omega)}^2.$$

This completes our proof of D.1. $\square$

### Regularity Results of the PINN Objective Function

**Theorem D.2.** *We consider the static Schrödinger equation on the unit hypercube on $\mathbb{R}^d$ with the Neumann boundary condition:*

$$-\Delta u + Vu = f \text{ on } \Omega,$$
$$u = 0 \text{ on } \partial\Omega. \tag{D.5}$$

*where $f \in L^2(\Omega)$ and $V \in L^\infty(\Omega)$ with $V - \frac{1}{2}\Delta V > C_{\min}, 0 < V(x) \leq V_{\max}$ and $\Delta V(x) \leq V_{\max}$. Then there exists a unique solution $u_S^*$ to the following minimization problem [3]:*

$$u_S^* = \arg\min_{u \in H^1(\Omega)} \boldsymbol{E}_S^{PINN}(u) := \arg\min_{u \in H^1(\Omega)} \Big\{ \int_\Omega |\Delta u - Vu + f|^2 dx\Big\}. \tag{D.6}$$

*Then for any $u \in H_1(\Omega)$, we have:*

$$\min\{1, C_{\min}\}\|u - u_S^*\|_{H_2(\Omega)}^2 \leq \boldsymbol{E}(u) - \boldsymbol{E}(u_S^*) \leq 2(1 + V_{\max})\|u - u_S^*\|_{H_2(\Omega)}^2. \tag{D.7}$$

*Proof.* Let $\tilde{u} = u - u^*$, we have

$$\boldsymbol{E}(u) - \boldsymbol{E}(u_S^*) = \int_\Omega (\Delta\tilde{u})^2 + V^2\tilde{u}^2 - 2V\tilde{u}\Delta\tilde{u} dx = \int_\Omega (\Delta\tilde{u})^2 + V^2\tilde{u}^2 + 2V\|\nabla\tilde{u}\|^2 + 2\tilde{u}\nabla V \cdot \nabla\tilde{u} dx$$

$$= \int (V - \frac{1}{2}\Delta V)\tilde{u}^2 + V\|\nabla\tilde{u}\|^2 + (\Delta\tilde{u})^2 \tag{D.8}$$

For we have assumed $V \in L^\infty(\Omega)$ with $V - \frac{1}{2}\Delta V > C_{\min}, 0 < V(x) \le V_{\max}$ and $\Delta V(x) \le V_{\max}$, thus we have

$$\min\{1, C_{\min}\}\|u - u_S^*\|_{H_2(\Omega)}^2 \le \boldsymbol{E}(u) - \boldsymbol{E}(u_S^*) \le 2(1 + V_{\max})\|u - u_S^*\|_{H_2(\Omega)}^2. \tag{D.9}$$

$\square$

## D.2 Auxiliary definitions and lemmata On Generalization Error

To bound the generalization error, we use the localized Rademencher complexity [2]. Recall that the Rademacher complexity of a function class $\boldsymbol{G}$ is defined by

$$R_n(\boldsymbol{G}) = \mathbb{E}_Z \mathbb{E}_\sigma \left[ \sup_{g \in \boldsymbol{G}} \left| \frac{1}{n} \sum_{j=1}^n \sigma_j g(Z_j) \right| \, \Big| \, Z_1, \cdots, Z_n \right],$$

where $Z_i$ are i.i.d samples according to the data distributions and $\sigma_j$ are i.i.d Rademencher random variables which take the value 1 with probaility $\frac{1}{2}$ and value -1 with probaility $\frac{1}{2}$.

The following important symmetrization lemma makes the connection between the uniform law of large numbers and the Rademacher complexity.

**Lemma D.1** (Symmetrization Lemma)**.** *Let $\boldsymbol{F}$ be a set of functions. Then*

$$\mathbb{E} \sup_{u \in \boldsymbol{F}} \left| \frac{1}{n} \sum_{j=1}^n u(X_j) - \mathbb{E}_{X \sim \boldsymbol{P}_\Omega} u(X) \right| \le 2R_n(\boldsymbol{F}).$$

**Lemma D.2** (Ledoux-Talagrand contraction [32, Theorem 4.12])**.** *Assume that $\phi : \mathbb{R} \to \mathbb{R}$ is L-Lipschitz with $\phi(0) = 0$. Let $\{\sigma_i\}_{i=1}^n$ be independent Rademacher random variables. Then for any $T \subset \mathbb{R}^n$*

$$\mathbb{E}_\sigma \sup_{(t_1, \cdots, t_n) \in T} \left| \sum_{i=1}^n \sigma_i \phi(t_i) \right| \le 2L \cdot \mathbb{E}_\sigma \sup_{(t_1, \cdots, t_n) \in T} \left| \sum_{i=1}^n \sigma_i t_i \right|.$$

Let $(E, \rho)$ be a metric space with metric $\rho$. A $\delta$-*cover* of a set $A \subset E$ with respect to $\rho$ is a collection of points $\{x_1, \cdots, x_n\} \subset A$ such that for every $x \in A$, there exists $i \in \{1, \cdots, n\}$ such that $\rho(x, x_i) \le \delta$. The $\delta$-covering number $\boldsymbol{N}(\delta, A, \rho)$ is the cardinality of the smallest $\delta$-cover of the set $A$ with respect to the metric $\rho$. Equivalently, the $\delta$-covering number $\boldsymbol{N}(\delta, A, \rho)$ is the minimal number of balls $B_\rho(x, \delta)$ of radius $\delta$ needed to cover the set $A$.

**Theorem D.3** (Dudley's Integral theorem)**.** *Let $\boldsymbol{F}$ be a function class such that $\sup_{f \in \boldsymbol{F}} \|f\|_\infty \le M$. Then the Rademacher complexity $R_n(\boldsymbol{F})$ satisfies that*

$$R_n(\boldsymbol{F}) \le \inf_{0 \le \delta \le M} \left\{ 4\delta + \frac{12}{\sqrt{n}} \int_\delta^M \sqrt{\log \boldsymbol{N}(\epsilon, \boldsymbol{F}, \| \cdot \|_\infty)} \, d\epsilon \right\}.$$

**Lemma D.3** (Talagrand Concentration Inequality)**.** *For a function class $\mathcal{F}$ defined on a probability measure $\mu$, if for all $f \in \mathcal{F}$, we have $\|f\|_\infty \le \beta, \mathbb{E}_\mu[f] = 0, \mathbb{E}_\mu[f^2] \le \sigma^2$. Then for any $t > 0$, we can have the following concentration results.*

$$\mathbb{P}_{z_1, \cdots, z_n \sim \mu} \left[ \sup_{f \in \mathcal{F}} \frac{1}{n} \sum_{i=1}^n f(z_i) \ge 2 \sup_{f \in \mathcal{F}} \mathbb{E}_{z_1', \cdots, z_n' \sim \mu} \frac{1}{n} \sum_{i=1}^n f(z_i') + \sqrt{\frac{2t\sigma^2}{n}} + \frac{2t\beta}{n} \right] \le e^{-t}$$

**Lemma D.4** (Peeling lemma [2])**.** *For a function class $\mathcal{F}$ defined on a probability measure $\mu$, if for all $f \in \mathcal{F}$, we have $\|f\|_\infty \le \beta, \mathbb{E}_\mu[f] = 0$. We further have a sub-root function $\phi(r)$ satisfies*

$$R_n(\{f \in \mathcal{F} | Pf \le r\}) \le \phi(r) \ (\forall r > 0). \tag{D.10}$$

*Then we have*

$$\mathbb{E}_{\sigma_i, z_n} \left[ \sup_{g \in \mathcal{G}} \frac{\frac{1}{n} \sum_{i=1}^n \sigma_i g(z_i)}{Pg + r} \right] \le \frac{4\phi(r)}{r}$$

*Proof.* Denote $G(r)$ to be the localized set with radius $r$. Then we have

$$\mathbb{E}_{\sigma_i, z_i}\left[\frac{\frac{1}{n}\sum_{i=1}^n \sigma_i g(z_i)}{\mathbf{P}g + r}\right] \leq \sup_{g \in \mathcal{G}(r)} \frac{\frac{1}{n}\sum_{i=1}^n \sigma_i g(z_i)}{r}$$

$$+ \sum_{j=0}^{\infty} \sup_{g \in \mathcal{G}(r4^{j+1})\backslash\mathcal{G}(r4^j)} \frac{\frac{1}{n}\sum_{i=1}^n \sigma_i g(z_i)}{r4^j + r}$$

$$\leq \frac{R_n(G_r)}{r} + \sum_{j=0}^{\infty} \frac{R_n(G_{r4^{j+1}+r})}{r4^j + r} \leq \frac{\phi(r)}{r} + \sum_{j=0}^{\infty} \frac{\phi(r4^{j+1}+r)}{r4^j + r}$$

$$\leq \frac{\phi(r)}{r} + \sum_{j=0}^{\infty} \frac{2^{j+1}\phi(r)}{r4^j + r} \leq \frac{4\phi(r)}{r}$$

$\square$

### D.2.1 Local Rademacher Complexity of Truncated Fourier Basis

**Definition D.1.** *(Fourier Series) Given a domain $\Omega \subseteq [0,1]^d$. For any $z \in \mathbb{N}^d$, we consider the corresponding Fourier basis function $\phi_z(x) := e^{2\pi i \langle z, x\rangle}$ $(x \in \Omega)$. With respect to the Fourier basis, any function $f \in L^2(\Omega)$ can be decomposed as the following sum:*

$$f(x) := \sum_{z \in \mathbb{N}^d} f_z \phi_z(x). \tag{D.11}$$

*where for any $z \in \mathbb{N}^d$, the Fourier coefficient $f_z = \int_\Omega f(x)\overline{\phi_z(x)}dx$.*

**Definition D.2.** *(Truncated Fourier Series) For a fixed positive integer $\xi \in \mathbb{Z}^+$, we define the space $F_\xi(\Omega)$ of truncated Fourier series as follows:*

$$F_\xi(\Omega) := \left\{ f = \sum_{z \in \mathbb{N}^d} f_z \phi_z \,\Big|\, f_z = 0, \forall \|z\|_\infty > \xi \right\}. \tag{D.12}$$

*Equivalently, we can decompose any $f \in F_\xi(\Omega)$ as $f := \sum_{\|z\|_\infty \leq \xi} f_z \phi_z$.*

**Lemma D.5.** *(Local Rademacher Complexity of Localized Truncated Fourier Series) For a fixed $\xi \in \mathbb{Z}^+$, we consider a localized class of functions $\boldsymbol{F}_{\rho,\xi}(\Omega) = \left\{ f \in F_\xi(\Omega) \,\Big|\, \|f\|_{H^1(\Omega)}^2 \leq \rho \right\}$, where $\rho > 0$ is fixed. Then we have the following upper bound on the local Rademacher complexity:*

$$R_n(\boldsymbol{F}_{\rho,\xi}(\Omega)) = \mathbb{E}_X\left[\mathbb{E}_\sigma\left[\sup_{f \in \boldsymbol{F}_{\rho,\xi}(\Omega)} \frac{1}{n}\sum_{i=1}^n \sigma_i f(X_i) \,\Big|\, X_1, \cdots, X_n\right]\right] \lesssim \sqrt{\frac{\rho}{n}} \xi^{\frac{d-2}{2}}. \tag{D.13}$$

*Proof.* Take an arbitrary function $f \in \boldsymbol{F}_{\rho,\xi}(\Omega)$. Let $f = \sum_{\|z\|_\infty \leq \xi} f_z \phi_z$ be the Fourier basis expansion of $f$. On the one hand, substituting the Fourier expansion into the norm restriction and using orthogonality of the Fourier basis imply:

$$\rho \geq \|f\|_{H^1(\Omega)}^2 = \|\sum_{\|z\|_\infty \leq \xi} f_z \phi_z\|_{H^1(\Omega)}^2 = \|\sum_{\|z\|_\infty \leq \xi} f_z \phi_z\|_{L^2(\Omega)}^2 + \|\sum_{\|z\|_\infty \leq \xi} f_z \nabla\phi_z\|_{L^2(\Omega)}^2$$

$$= \sum_{\|z\|_\infty \leq \xi} |f_z|^2 \|\phi_z\|_{L^2(\Omega)}^2 + \sum_{\|z\|_\infty \leq \xi} |f_z|^2 \|\nabla\phi_z\|_{L^2(\Omega)}^2$$

$$= |\Omega|\left(\sum_{\|z\|_\infty \leq \xi} |f_z|^2 + 4\pi^2 \sum_{\|z\|_\infty \leq \xi} |f_z|^2 \|z\|_2^2\right), \Rightarrow \sum_{\|z\|_\infty \leq \xi} |f_z|^2 \|z\|_2^2 \lesssim \rho.$$

On the other hand, substituting the Fourier expansion into the average sum $\frac{1}{n}\sum_{i=1}^{n}\sigma_i f(X_i)$ and using Cauchy-Schwarz inequality let us upper bound as follows:

$$\frac{1}{n}\sum_{i=1}^{n}\sigma_i f(X_i) = \frac{1}{n}\sum_{i=1}^{n}\sigma_i \sum_{\|z\|_\infty \le \xi} f_z \phi_z(X_i) = \frac{1}{n}\sum_{\|z\|_\infty \le \xi}\sum_{i=1}^{n}\sigma_i f_z \phi_z(X_i)$$

$$\le \frac{1}{n}\Big(\sum_{\|z\|_\infty \le \xi}|f_z|^2\|z\|_2^2\Big)^{\frac{1}{2}}\Big(\sum_{\|z\|_\infty \le \xi}\Big|\sum_{i=1}^{n}\frac{\sigma_i}{\|z\|_2}\phi_z(X_i)\Big|^2\Big)^{\frac{1}{2}}$$

$$\lesssim \frac{\sqrt{\rho}}{n}\Big(\sum_{\|z\|_\infty \le \xi}\Big|\sum_{i=1}^{n}\frac{\sigma_i}{\|z\|_2}\phi_z(X_i)\Big|^2\Big)^{\frac{1}{2}}.$$

where we have used the constraint $\sum_{\|z\|_\infty \le \xi}|f_z|^2\|z\|_2^2 \lesssim \rho$ in the last step above. Moreover, by taking expectation with respect to the i.i.d Rademacher random variables $\sigma_i$ $(1 \le i \le n)$ and the uniformly sampled data points $\{X_i\}_{i=1}^{n}$ on both sides and applying Jensen's inequality, we can deduce that:

$$\mathbb{E}_X\mathbb{E}_\sigma\Big[\frac{1}{n}\sum_{i=1}^{n}\sigma_i f(X_i)\Big] \lesssim \frac{\sqrt{\rho}}{n}\mathbb{E}_{X,\sigma}\Big[\Big(\sum_{\|z\|_\infty \le \xi}\Big|\sum_{i=1}^{n}\frac{\sigma_i}{\|z\|_2}\phi_z(X_i)\Big|^2\Big)^{\frac{1}{2}}\Big]$$

$$\le \frac{\sqrt{\rho}}{n}\Big(\mathbb{E}_{X,\sigma}\Big[\sum_{\|z\|_\infty \le \xi}\Big|\sum_{i=1}^{n}\frac{\sigma_i}{\|z\|_2}\phi_z(X_i)\Big|^2\Big]\Big)^{\frac{1}{2}}.$$

Using independence between the random variables $\sigma_i$ $(1 \le i \le n)$, we can further simplify the expectation inside the square root above as below:

$$\mathbb{E}_{X,\sigma}\Big[\sum_{\|z\|_\infty \le \xi}\Big|\sum_{i=1}^{n}\frac{\sigma_i}{\|z\|_2}\phi_z(X_i)\Big|^2\Big] = \sum_{\|z\|_\infty \le \xi}\mathbb{E}_{X,\sigma}\Big[\Big|\sum_{i=1}^{n}\frac{\sigma_i}{\|z\|_2}\phi_z(X_i)\Big|^2\Big]$$

$$= \sum_{\|z\|_\infty \le \xi}\sum_{i=1}^{n}\mathbb{E}_{X,\sigma}\Big[\frac{\sigma_i^2}{\|z\|_2^2}\big|\phi_z(X_i)\big|^2\Big]$$

$$= \sum_{\|z\|_\infty \le \xi}\sum_{i=1}^{n}\frac{|\Omega|}{\|z\|_2^2} \lesssim n\sum_{\|z\|_\infty \le \xi}\frac{1}{\|z\|_2^2} \lesssim n\frac{\xi^d}{\xi^2} = n\xi^{d-2}.$$

Combining the two bounds above yields the desired upper bound:

$$\mathbb{E}_X\Big[\mathbb{E}_\sigma\Big[\sup_{f \in \boldsymbol{F}_{\rho,\xi}(\Omega)}\frac{1}{n}\sum_{i=1}^{n}\sigma_i f(X_i)\,\Big|\,X_1,\cdots,X_n\Big]\Big] \lesssim \frac{\sqrt{\rho}}{n}\sqrt{n\xi^{d-2}} = \sqrt{\frac{\rho}{n}}\xi^{\frac{d-2}{2}}.$$

$\square$

**Lemma D.6.** *(Local Rademacher Complexity of Localized Truncated Fourier Series' Gradient) For a fixed $\xi \in \mathbb{Z}^+$, we consider a localized class of functions $\boldsymbol{G}_{\rho,\xi}(\Omega) = \{\|\nabla f\|_2 \mid f \in F_{\rho,\xi}(\Omega)\}$, where $\rho > 0$ is fixed. Then for any sample $\{X_i\}_{i=1}^{n} \subset \Omega$, we have the following upper bound on the local Rademacher complexity:*

$$R_n(\boldsymbol{G}_{\rho,\xi}(\Omega)) = \mathbb{E}_X\Big[\mathbb{E}_\sigma\Big[\sup_{f \in \boldsymbol{F}_{\rho,\xi}(\Omega)}\frac{1}{n}\sum_{i=1}^{n}\sigma_i\|\nabla f(X_i)\|_2\,\Big|\,X_1,\cdots,X_n\Big]\Big] \lesssim \sqrt{\frac{\rho}{n}}\xi^{\frac{d}{2}}. \quad \text{(D.14)}$$

*Proof.* Take an arbitrary function $f \in \boldsymbol{F}_{\rho,\xi}(\Omega)$. Let $f = \sum_{\|z\|_\infty \le \xi} f_z\phi_z$ be the Fourier basis expansion of $f$. Similarly, the norm restriction condition $\|f\|_{H^1(\Omega)}^2 \le \rho$ can be reduced to the following condition about fourier coefficients:

$$\sum_{\|z\|_\infty \le \xi}|f_z|^2\|z\|_2^2 \lesssim \rho.$$

Moreover, substituting the Fourier expansion into the average sum $\frac{1}{n}\sum_{i=1}^{n}\sigma_i\nabla f(X_i)$ and using Cauchy-Schwarz inequality let us upper bound as follows:

$$\frac{1}{n}\sum_{i=1}^{n}\sigma_i\|\nabla f(X_i)\|_2 = \frac{1}{n}\sum_{i=1}^{n}\sigma_i\|\sum_{\|z\|_\infty\leq\xi}f_z\nabla\phi_z(X_i)\|_2 \leq \frac{1}{n}\sum_{\|z\|_\infty\leq\xi}\sum_{i=1}^{n}\sigma_i\|f_z\nabla\phi_z(X_i)\|_2$$

$$\leq \frac{1}{n}\Big(\sum_{\|z\|_\infty\leq\xi}|f_z|^2\|z\|_2^2\Big)^{\frac{1}{2}}\Big(\sum_{\|z\|_\infty\leq\xi}\Big|\sum_{i=1}^{n}\frac{\sigma_i}{\|z\|_2}\|\nabla\phi_z(X_i)\|_2\Big|^2\Big)^{\frac{1}{2}}$$

$$\lesssim \frac{\sqrt{\rho}}{n}\Big(\sum_{\|z\|_\infty\leq\xi}\Big|\sum_{i=1}^{n}\frac{\sigma_i}{\|z\|_2}\|\nabla\phi_z(X_i)\|_2\Big|^2\Big)^{\frac{1}{2}}.$$

where we have used the constraint $\sum_{\|z\|_\infty\leq\xi}|f_z|^2\|z\|_2^2 \lesssim \rho$ in the last step above. Moreover, by taking expectation with respect to the i.i.d Rademacher random variables $\sigma_i$ ($1\leq i\leq n$) and the uniformly sampled data points $\{X_i\}_{i=1}^{n}$ on both sides and applying Jensen's inequality, we can deduce that:

$$\mathbb{E}_X\mathbb{E}_\sigma\Big[\frac{1}{n}\sum_{i=1}^{n}\sigma_i\|\nabla f(X_i)\|_2\Big] \lesssim \frac{\sqrt{\rho}}{n}\mathbb{E}_{X,\sigma}\Big[\Big(\sum_{\|z\|_\infty\leq\xi}\Big|\sum_{i=1}^{n}\frac{\sigma_i}{\|z\|_2}\|\nabla\phi_z(X_i)\|_2\Big|^2\Big)^{\frac{1}{2}}\Big]$$

$$\leq \frac{\sqrt{\rho}}{n}\Big(\mathbb{E}_{X,\sigma}\Big[\sum_{\|z\|_\infty\leq\xi}\Big|\sum_{i=1}^{n}\frac{\sigma_i}{\|z\|_2}\|\nabla\phi_z(X_i)\|_2\Big|^2\Big]\Big)^{\frac{1}{2}}.$$

Using independence between the random variables $\sigma_i$ ($1\leq i\leq n$), we can further simplify the expectation inside the square root above as below:

$$\mathbb{E}_{X,\sigma}\Big[\sum_{\|z\|_\infty\leq\xi}\Big|\sum_{i=1}^{n}\frac{\sigma_i}{\|z\|_2}\|\nabla\phi_z(X_i)\|_2\Big|^2\Big] = \sum_{\|z\|_\infty\leq\xi}\mathbb{E}_{X,\sigma}\Big[\Big|\sum_{i=1}^{n}\frac{\sigma_i}{\|z\|_2}\|\nabla\phi_z(X_i)\|_2\Big|^2\Big]$$

$$= \sum_{\|z\|_\infty\leq\xi}\sum_{i=1}^{n}\mathbb{E}_{X,\sigma}\Big[\frac{\sigma_i^2}{\|z\|_2^2}\|\nabla\phi_z(X_i)\|_2^2\Big]$$

$$= \sum_{\|z\|_\infty\leq\xi}\sum_{i=1}^{n}|\Omega|\frac{4\pi^2\|z\|_2^2}{\|z\|_2^2} \lesssim n\sum_{\|z\|_\infty\leq\xi}1 \lesssim n\xi^d.$$

Combining the two bounds above yields the desired upper bound:

$$\mathbb{E}_X\Big[\mathbb{E}_\sigma\Big[\sup_{f\in\boldsymbol{F}_{\rho,\xi}(\Omega)}\frac{1}{n}\sum_{i=1}^{n}\sigma_i\|\nabla f(X_i)\|_2 \ \Big|\ X_1,\cdots,X_n\Big]\Big] \lesssim \frac{\sqrt{\rho}}{n}\sqrt{n\xi^d} = \sqrt{\frac{\rho}{n}}\xi^{\frac{d}{2}}.$$

$$\square$$

### D.2.2 Local Rademacher Complexity of the Deep Neural Network Model

In this section we aim to bound the local Rademacher Complexity of a Deep Neural Network. We first bound the covering number of the function space composed by the gradient of all possible neural networks and then apply a Duley Integral to achieve the final bound.

**Definition D.3.** *Let $\eta_l$ denote the $l$-ReLU activiation function. Here we use $\eta_3 := \max\{0,x\}^3$[12] as the activation function to ensure smoothness. We can define the space consisting of all neural network models with depth $L$, width $W$, sparsity constraint $S$ and norm constraint $B$ as follows:*

$$\Phi(L,W,S,B) := \Big\{(\mathcal{W}^{(L)}\eta_3(\cdot)+b^{(L)})\cdots(\mathcal{W}^{(1)}x+b^{(1)}) \mid \mathcal{W}^{(L)}\in\mathbb{R}^{1\times W}, b^{(L)}\in\mathbb{R},\quad \text{(D.15)}$$

$$\mathcal{W}^{(1)}\in\mathbb{R}^{W\times d}, b^{(1)}\in\mathbb{R}^W, \mathcal{W}^{(l)}\in\mathbb{R}^{W\times W}, b^{(l)}\in\mathbb{R}^W(1<l<L), \quad\quad\quad\quad \text{(D.16)}$$

$$\sum_{l=1}^{L}(\|\mathcal{W}^{(l)}\|_0+\|b^{(l)}\|_0)\leq S, \max_l\|\mathcal{W}^{(l)}\|_{\infty,\infty}\vee\|b^{(l)}\|_\infty\leq B\Big\}. \quad\quad\quad\quad \text{(D.17)}$$

where $\| \cdot \|_0$ measures the number of nonzero entries in a matrix and $\| \cdot \|_{\infty,\infty}$ measures the maximum of the absolute values of the entries in a matrix.

For any $d \in \mathbb{Z}^+$, we refer to an arbitrary element in $\Phi(L, W, S, B)$ as a ReLU3 Deep Neural Network. Then for any index $1 \le k \le L$, we use $F_k$ to denote the $k-$ReLU3 Deep Neural Network composed by the first $k$ layers, i.e:

$$F_k(x) := (\mathcal{W}_F^{(k)} \eta_3(\cdot) + b_F^{(k)}) \cdots (\mathcal{W}_F^{(1)} x + b_F^{(1)}).$$

Also, we use $\Phi_k(L, W, S, B)$ to denote the space consisting of all $F_k$. In particular, when $k = L$, we have:

$$F(x) := F_L(x) = (\mathcal{W}_F^{(L)} \eta_3(\cdot) + b_F^{(L)}) \cdots (\mathcal{W}_F^{(1)} x + b_F^{(1)}), \text{ and } \Phi_L(L, W, S, B) = \Phi(L, W, S, B).$$

Furthermore, given that the domain $\Omega \subset [0, 1]^d$ is bounded, we have $\sup_{x \in \Omega} \|x\|_\infty = 1$

**Lemma D.7.** *(Upper bound on $\infty$-norm of functions in DNN space) For any $1 \le k \le L$, the following inequality holds:*

$$\sup_{x \in \Omega, \ F_k \in \Phi_k(L,W,S,B)} \|F_k(x)\|_\infty \le W^{\frac{3^{k-1}-1}{2}} (B \vee d)^{\frac{5 \cdot 3^{k-1}-1}{2}} 2^{\frac{3^{k-1}}{2}-k+1}.$$

*Proof.* We use induction to prove this claim.

Base cases: When $k = 1$, we have that for any $x \in \Omega$ and any $F_1 \in \Phi_1(L, W, S, B)$, the following holds:

$$\begin{aligned}
\|F_1(x)\|_\infty = \|\mathcal{W}_F^{(1)} x + b_F^{(1)}\|_\infty &\le \|\mathcal{W}_F^{(1)}\|_\infty \|x\|_\infty + \|b_F^{(1)}\|_\infty \\
&\le d\|\mathcal{W}_F^{(1)}\|_{\infty,\infty} + B \le dB + B \le 2(B \vee d)^2.
\end{aligned} \tag{D.18}$$

When $k = 2$, we have that for any $x \in \Omega$ and any $F_2 \in \Phi_2(L, W, S, B)$, the following holds:

$$\|F_2(x)\|_\infty = \|\mathcal{W}_F^{(2)} \eta_3(F_1(x)) + b_F^{(2)}\|_\infty \le \|\mathcal{W}_F^{(2)}\|_\infty \|\eta_3(F_1(x))\|_\infty + \|b_F^{(2)}\|_\infty \le W\|\mathcal{W}_F^{(2)}\|_{\infty,\infty} \|F_1(x)\|_\infty^3 + B.$$

By applying the bound proved in the case when $k = 1$, we have:

$$\begin{aligned}
\|F_2(x)\|_\infty &\le WB(dB + B)^3 + B = WB^4(d + 1)^3 + B \\
&= WB^4(d^3 + 3d^2 + 3d + 1) + B \le 8W(B \vee d)^7.
\end{aligned}$$

where the last inequality follows from the assumption that $W \ge 2$.

Inductive Step: Now we assume that the claim has been proved for $k-1$, where $3 \le k \le L$. Similarly, for any $x \in \Omega$ and any $F_k \in \Phi_k(L, W, S, B)$, we have:

$$\begin{aligned}
\|F_k(x)\|_\infty = \|\mathcal{W}_F^{(k)} \eta_3(F_{k-1}(x)) + b_F^{(k)}\|_\infty &\le \|\mathcal{W}_F^{(k)}\|_\infty \|\eta_3(F_{k-1}(x))\|_\infty + \|b_F^{(k)}\|_\infty \\
&\le W\|\mathcal{W}_F^{(k)}\|_{\infty,\infty} \|F_{k-1}(x)\|_\infty^3 + B \le WB\|F_{k-1}(x)\|_\infty^3 + B.
\end{aligned}$$

Using inductive hypothesis, we can further deduce that:

$$\begin{aligned}
\|F_k(x)\|_\infty &\le WB \times W^{\frac{3^{k-1}-3}{2}} (B \vee d)^{\frac{5 \cdot 3^{k-1}-3}{2}} 2^{\frac{3^k-3}{2}-3k+6} + B \\
&\le W^{\frac{3^{k-1}-1}{2}} (B \vee d)^{\frac{5 \cdot 3^{k-1}-1}{2}} 2^{\frac{3^k-3}{2}-3k+6} + B \vee d \\
&\le W^{\frac{3^{k-1}-1}{2}} (B \vee d)^{\frac{5 \cdot 3^{k-1}-1}{2}} [2^{\frac{3^k-3}{2}-3k+6} + 1] \\
&\le W^{\frac{3^{k-1}-1}{2}} (B \vee d)^{\frac{5 \cdot 3^{k-1}-1}{2}} 2^{\frac{3^k-3}{2}-k+2} \ (k \ge 3) \\
&= W^{\frac{3^{k-1}-1}{2}} (B \vee d)^{\frac{5 \cdot 3^{k-1}-1}{2}} 2^{\frac{3^k-1}{2}-k+1}.
\end{aligned}$$

Thus, the inequality also holds for $k$. By induction, the claim is proved. $\square$

We also need to show that the ReLu3 activation function is a Lipschitzness functions over a bounded domain.

**Lemma D.8.** *For any $k \in \mathbb{Z}^+$, consider the $k-$ReLU activation function $\eta_k$ defined on some bounded domain $\mathcal{D} \subset \mathbb{R}^d$ (i.e, $\sup_{x \in \mathcal{D}} \|x\|_\infty \le C$ for some $C > 0$). Then we have that for any $x, y \in \mathcal{D}$, the following inequalities hold:*

$$\begin{aligned}
\|\eta_3(x) - \eta_3(y)\|_\infty &\le 3C^2\|x - y\|_\infty, \\
\|\eta_2(x) - \eta_2(y)\|_\infty &\le 2C\|x - y\|_\infty.
\end{aligned}$$

*Proof.* This is because $|\nabla\eta_3(x)| = |3\max\{x,0\}^2| \le 3C^2$ and $|\nabla\eta_2(x)| = |2\max\{x,0\}| \le 2C$. $\qquad\square$

**Lemma D.9.** *(Relation between the covering number of DNN space and parameter space) For any* $1 \le k \le L$, *suppose that a pair of different two networks* $F_k, G_k \in \Phi_k(L,W,S,B)$ *are given by:*

$$F_k(x) := (\mathcal{W}_F^{(k)}\eta_3(\cdot) + b_F^{(k)})\cdots(\mathcal{W}_F^{(1)}x + b_F^{(1)}),$$

$$G_k(x) := (\mathcal{W}_G^{(k)}\eta_3(\cdot) + b_G^{(k)})\cdots(\mathcal{W}_G^{(1)}x + b_G^{(1)}).$$

*Furthermore, assume that the* $\|\ \|_\infty$ *norm of the distance between the parameter spaces are upper bounded by* $\delta$, *i.e*

$$\|W_F^{(l)} - W_G^{(l)}\|_{\infty,\infty} \le \delta, \ \|b_F^{(l)} - b_G^{(l)}\|_\infty \le \delta. \ (\forall \ 1 \le l \le k)$$

*Then we have that:*

$$\sup_{x\in\Omega, F_k, G_k \in \Phi_k(L,W,S,B)} \|F_k(x) - G_k(x)\|_\infty \le \delta W^{\frac{3^{k-1}-1}{2}}(B\vee d)^{\frac{5\cdot3^{k-1}-1}{2}}2^{\frac{3^k-1}{2}-k+2}3^{k-1}.$$

$$(D.19)$$

*Proof.* Let's prove the claim by using induction on $k$.

Base Case: When $k = 1$, we have that for any $x \in \Omega$ and any $F_1, G_1 \in \Phi_1(L,W,S,B)$, the following holds:

$$\begin{aligned}\|F_1(x) - G_1(x)\|_\infty &= \|\mathcal{W}_F^{(1)}x + b_F^{(1)} - \mathcal{W}_G^{(1)}x - b_G^{(1)}\|_\infty \\ &\le \|\mathcal{W}_F^{(1)} - \mathcal{W}_G^{(1)}\|_\infty\|x\|_\infty + \|b_F^{(1)} - b_G^{(1)}\|_\infty \qquad (D.20) \\ &\le \delta d + \delta = \delta(d+1) \le 2\delta(B\vee d) \le 4\delta(B\vee d)^2.\end{aligned}$$

When $k = 2$, we have that for any $x \in \Omega$ and any $F_2, G_2 \in \Phi_2(L,W,S,B)$, the following inequality holds:

$$\begin{aligned}\|F_2(x) - G_2(x)\|_\infty &= \|\mathcal{W}_F^{(2)}\eta_3(F_1(x)) + b_F^{(2)} - \mathcal{W}_G^{(2)}\eta_3(G_1(x)) - b_G^{(2)}\|_\infty \\ &\le \|\mathcal{W}_F^{(2)}\eta_3(F_1(x)) - \mathcal{W}_G^{(2)}\eta_3(G_1(x))\|_\infty + \|b_F^{(2)} - b_G^{(2)}\|_\infty \\ &\le \|\mathcal{W}_F^{(2)}\eta_3(F_1(x)) - \mathcal{W}_G^{(2)}\eta_3(F_1(x))\|_\infty + \|\mathcal{W}_G^{(2)}\eta_3(F_1(x)) - \mathcal{W}_G^{(2)}\eta_3(G_1(x))\|_\infty + \delta.\end{aligned}$$

By applying the upper bound proved in equation D.18, we can upper bound the first part $\|\mathcal{W}_F^{(2)}\eta_3(F_1(x)) - \mathcal{W}_G^{(2)}\eta_3(F_1(x))\|_\infty$ by:

$$\begin{aligned}\|\mathcal{W}_F^{(2)}\eta_3(F_1(x)) - \mathcal{W}_G^{(2)}\eta_3(F_1(x))\|_\infty &\le \|\mathcal{W}_F^{(2)} - \mathcal{W}_G^{(2)}\|_\infty\|\eta_3(F_1(x))\|_\infty \\ &\le W\delta\|F_1(x)\|_\infty^3 \le \delta W[2W(B\vee d)]^3.\end{aligned}$$

By applying the Lipschitz condition proved in Lemma D.8 and the bound proved in equation D.20, we can further upper bound the second part $\|\mathcal{W}_G^{(2)}\eta_3(F_1(x)) - \mathcal{W}_G^{(2)}\eta_3(G_1(x))\|_\infty$ by:

$$\begin{aligned}\|\mathcal{W}_G^{(2)}\eta_3(F_1(x)) - \mathcal{W}_G^{(2)}\eta_3(G_1(x))\|_\infty &\le \|\mathcal{W}_G^{(2)}\|_\infty\|\eta_3(F_1(x)) - \eta_3(G_1(x))\|_\infty \\ &\le WB\times 3\sup_{F_1\in\Phi_1(L,W,S,B)}\|F_1(x)\|_\infty^2 \times \|F_1(x) - G_1(x)\|_\infty \\ &\le WB\times 3[2W(B\vee d)]^2 \times 2\delta W(B\vee d) \\ &\le 24\delta W^4(B\vee d)^4.\end{aligned}$$

Summing the two upper bounds above yields:

$$\|F_2(x) - G_2(x)\|_\infty \le 8\delta W^4(B\vee d)^3 + 24\delta W^4(B\vee d)^4 + \delta \le 48\delta W^4(B\vee d)^4.$$

where we again use the assumption $W \ge 2$ in the last step.

Inductive Step: Now we assume that the claim has been proved for $k - 1$, where $k \ge 3$. For any $x \in \Omega$ and $F_k \in \Phi_k(L,W,S,B)$, we have that:

$$\begin{aligned}\|F_k(x) - G_k(x)\|_\infty &= \|\mathcal{W}_F^{(k)}\eta_3(F_{k-1}(x)) + b_F^{(k)} - \mathcal{W}_G^{(k)}\eta_3(G_{k-1}(x)) - b_G^{(k)}\|_\infty \\ &\le \|\mathcal{W}_F^{(k)}\eta_3(F_{k-1}(x)) - \mathcal{W}_G^{(k)}\eta_3(G_{k-1}(x))\|_\infty + \|b_F^{(k)} - b_G^{(k)}\|_\infty \\ &\le \|\mathcal{W}_F^{(k)}\eta_3(F_{k-1}(x)) - \mathcal{W}_G^{(k)}\eta_3(G_{k-1}(x))\|_\infty + \delta.\end{aligned}$$

Applying triangle inequality helps us upper bound the first term above as follows:

$$\|\mathcal{W}_F^{(k)}\eta_3(F_{k-1}(x)) - \mathcal{W}_G^{(k)}\eta_3(G_{k-1}(x))\|_\infty$$
$$\leq \|\mathcal{W}_F^{(k)}\eta_3(F_{k-1}(x)) - \mathcal{W}_G^{(k)}\eta_3(F_{k-1}(x))\|_\infty + \|\mathcal{W}_G^{(k)}\eta_3(F_{k-1}(x)) - \mathcal{W}_G^{(k)}\eta_3(G_{k-1}(x))\|_\infty$$
$$\leq \|\mathcal{W}_F^{(k)} - \mathcal{W}_G^{(k)}\|_\infty\|\eta_3(F_{k-1}(x))\|_\infty + \|\mathcal{W}_G^{(k)}\|_\infty\|\eta_3(F_{k-1}(x)) - \eta_3(G_{k-1}(x))\|_\infty$$
$$\leq \delta W\|F_{k-1}(x)\|_\infty^3 + BW\|\eta_3(F_{k-1}(x)) - \eta_3(G_{k-1}(x))\|_\infty.$$

From Lemma D.7, we can upper bound the first term $\delta W\|F_{k-1}(x)\|_\infty^3$ by:

$$\delta W\|F_{k-1}(x)\|_\infty^3 \leq \delta W^{\frac{3^{k-1}-1}{2}}(B \vee d)^{\frac{5 \cdot 3^{k-1}-3}{2}}2^{\frac{3^k-3}{2}-3k+6}.$$

Moreover, applying Lemma D.8 and the inductive hypothesis let us upper bound the second term $BW\|\eta_3(F_{k-1}(x)) - \eta_3(G_{k-1}(x))\|_\infty$ as follows:

$$BW\|\eta_3(F_{k-1}(x)) - \eta_3(G_{k-1}(x))\|_\infty$$
$$\leq BW \times 3 \sup_{x \in \Omega,\ F_{k-1} \in \Phi_{k-1}(L,W,S,B)} \|F_{k-1}(x)\|_\infty^2 \times \|F_{k-1}(x) - G_{k-1}(x)\|_\infty$$
$$\leq 3BW \times W^{3^{k-2}-1}(B \vee d)^{5 \times 3^{k-2}-1}2^{3^{k-1}-1-2k+4}\|F_{k-1}(x) - G_{k-1}(x)\|_\infty$$
$$\leq 3BW \times W^{3^{k-2}-1}(B \vee d)^{5 \times 3^{k-2}-1}2^{3^{k-1}-1-2k+4} \times \delta W^{\frac{3^{k-2}-1}{2}}(B \vee d)^{\frac{5 \cdot 3^{k-2}-1}{2}}2^{\frac{3^{k-1}-1}{2}-k+3}3^{k-2}$$
$$\leq 3^{k-1}\delta W^{\frac{3^{k-1}-1}{2}}(B \vee d)^{\frac{5 \times 3^{k-1}-1}{2}}2^{\frac{3^k-1}{2}-3k+6}.$$

Combining the two upper bounds derived above yields:

$$\|F_k(x) - G_k(x)\|_\infty \leq \delta W^{\frac{3^{k-1}-1}{2}}(B \vee d)^{\frac{5 \cdot 3^{k-1}-3}{2}}2^{\frac{3^k-3}{2}-3k+6}$$
$$+ 3^{k-1}\delta W^{\frac{3^{k-1}-1}{2}}(B \vee d)^{\frac{5 \times 3^{k-1}-1}{2}}2^{\frac{3^k-1}{2}-3k+6} + \delta$$
$$\leq \delta 3^{k-1}W^{\frac{3^{k-1}-1}{2}}(B \vee d)^{\frac{5 \times 3^{k-1}-1}{2}}2^{\frac{3^k-1}{2}-k+2},$$

where the last inequality above follows from the fact that $k \geq 3$. Thus, the upper bound also holds for $k$. By induction, the claim is proved. $\qquad\square$

**Theorem D.4.** *(Bounding the DNN space covering number) Fix some sufficiently large $N \in \mathbb{Z}^+$. Consider a Deep Neural Network space $\Phi(L, W, S, B)$ with $L = O(1), W = O(N), S = O(N)$ and $B = O(N)$. Then the $\log$ value of the covering number of this DNN space with respect to the $\|\cdot\|_\infty$ norm, which is denoted by $\mathcal{N}(\delta, \Phi(L, W, S, B), \|\cdot\|_\infty)$, can be bounded by:*

$$\log\mathcal{N}(\delta, \Phi(L, W, S, B), \|\cdot\|_\infty) = O\left(S\Big[\log(\delta^{-1}) + 3^L\log(WB)\Big]\right) \qquad (D.21)$$

*Proof.* We firstly fix a sparsity pattern (i.e, the locations of the non-zero entries are fixed). By picking $k = L$ in Lemma D.9, we get the following upper bound on the covering number with respect to $\|\cdot\|_\infty$:

$$\left(\frac{\delta}{3^{L-1}W^{\frac{3^{L-1}-1}{2}}(B \vee d)^{\frac{5 \times 3^{L-1}-1}{2}}2^{\frac{3^L-1}{2}-L+2}}\right)^{-S}$$

Furthermore, note that the number of feasible configurations is upper bounded by:

$$\binom{(W+1)^L}{S} \leq (W+1)^{LS}$$

Multiplying the two bounds above yields:

$$\log\mathcal{N}(\delta, \Phi(L, W, S, B), \|\cdot\|_\infty) \leq \log\left[(W+1)^{LS}\{\frac{\delta}{3^{L-1}W^{\frac{3^{L-1}-1}{2}}(B \vee d)^{\frac{5 \times 3^{L-1}-1}{2}}2^{\frac{3^L-1}{2}-L+2}})^{-S}\right]$$
$$\leq S\log\left[\delta^{-1}(W+1)^L3^{L-1}W^{\frac{3^{L-1}-1}{2}}(B \vee d)^{\frac{5 \times 3^{L-1}-1}{2}}2^{\frac{3^L-1}{2}-L+2}\right]$$
$$\lesssim S\left[\log(\delta^{-1}) + L\log(3W) + 3^L\log(W(B \vee d)) + 3^L\log 2\right]$$

Note that here the dimension $d$ is some constant. Thus, by plugging in thee given magnitudes $L = O(1), W = O(N), S = O(N)$ and $B = O(N)$, we can further deduce that:

$$\log \mathcal{N}(\delta, \Phi(L, W, S, B), \|\cdot\|_\infty) \lesssim S\Big[\log(\delta^{-1}) + 3^L \log(WB)\Big]$$

This finishes our proof. $\qquad\qquad\qquad\qquad\qquad\qquad\qquad\qquad\qquad\qquad\qquad\qquad\qquad\quad$ $\square$

Now let's consider upper bounding the covering number of the $l_2$ norm of the sparse Deep Neural Networks' gradients. Note that for any $1 \le k \le L - 1$, any $k-$ReLU3 Deep Neural Network $F_k \in \Phi_k(L, W, S, B)$ is a map from $\mathbb{R}^d$ to $\mathbb{R}^W$. For any $1 \le l \le W$, we use $F_{k,l}(x)$ to denote the $l$-th component of the map $F_k$. This helps us write the map $F_k(x)$ and its Jacobian matrix $J[F_k](x)$ explicitly as:

$$F_k(x) = [F_{k,1}(x), F_{k,2}(x), \cdots, F_{k,W}(x)]^T \in \mathbb{R}^W$$

$$J[F_k](x) = \begin{bmatrix} \frac{\partial}{\partial x_1} F_{k,1}(x) & \frac{\partial}{\partial x_2} F_{k,1}(x) & \cdots & \frac{\partial}{\partial x_d} F_{k,1}(x) \\ \frac{\partial}{\partial x_1} F_{k,2}(x) & \frac{\partial}{\partial x_2} F_{k,2}(x) & \cdots & \frac{\partial}{\partial x_d} F_{k,2}(x) \\ \cdots & \cdots & \ddots & \\ \frac{\partial}{\partial x_1} F_{k,W}(x) & \frac{\partial}{\partial x_2} F_{k,W}(x) & \cdots & \frac{\partial}{\partial x_d} F_{k,W}(x) \end{bmatrix} \in \mathbb{R}^{W \times d}$$

In particular, when $k = L$, we have that any $F_L \in \Phi_L(L, W, S, B) = \Phi(L, W, S, B)$ is a map from $\mathbb{R}^d$ to $\mathbb{R}$. Thus, its Jacobian can be explicitly written as the following row vector:

$$J[F_L](x) = [\frac{\partial}{\partial x_1} F_L(x), \frac{\partial}{\partial x_2} F_L(x), \cdots \frac{\partial}{\partial x_d} F_L(x)] \in \mathbb{R}^{1 \times d}$$

**Lemma D.10.** *(Upper bound on $\infty$-norm of function gradients in DNN space) For any $1 \le k \le L$, the following inequality holds:*

$$\sup_{x \in \Omega, F_k \in \Phi_k(L, W, S, B)} \|J[F_k](x)\|_\infty \le 3^{k-1}[W(B \vee d)]^{\frac{3^k - 1}{2}} 2^{\frac{3^k - 1}{2} - k^2}$$

*Proof.* We use induction on $k$ to prove the claim.
Base case: $k = 1$. By the definition of Jacobian matrix, we have that for any $x \in \Omega$ and any $F_1 \in \Phi_1(L, W, S, B)$, the following holds:

$$\|J[F_1](x)\|_\infty = \|\mathcal{W}_F^{(1)}\|_\infty \le dB \le W(B \vee d)$$

Inductive Step: Assume that the claim has been proved for $k - 1$. Again, by the definition of Jacobian matrix, we can write the Jacobian matrix $J[F_k](x)$ as follows:

$$J[F_k](x) = \mathcal{W}_F^{(k)} J[\eta_3 \circ F_{k-1}](x)$$

$$\Rightarrow \|J[F_k](x)\|_\infty \le \|\mathcal{W}_F^{(k)}\|_\infty \|J[\eta_3 \circ F_{k-1}](x)\|_\infty \le WB\|J[\eta_3 \circ F_{k-1}](x)\|_\infty$$

Note that for any $2 \le k \le L$, the mapping $\eta_3 \circ F_{k-1}$ maps from $\mathbb{R}^d$ to $\mathbb{R}^W$. Hence, the Jacobian matrix $J[\eta_3 \circ F_{k-1}](x)$ is of shape $\mathbb{R}^{W \times d}$. Moreover, from Chain Rule we know that its $\infty$-norm can be written as:

$$\|J[\eta_3 \circ F_{k-1}](x)\|_\infty = \sup_{1 \le l \le W} (\sum_{j=1}^d |3\eta_2(F_{k-1,l}(x)) \frac{\partial F_{k-1,l}(x)}{\partial x_j}|)$$

Furthermore, for any $1 \le l \le W$, the expression on the RHS above can be upper bounded as below:

$$\sum_{j=1}^d |3\eta_2(F_{k-1,l}(x)) \frac{\partial F_{k-1,l}(x)}{\partial x_j}|$$

$$\le 3\|F_{k-1}(x)\|_\infty^2 (\sum_{j=1}^d |\frac{\partial}{\partial x_j} F_{k-1,l}(x)|) \le 3\|F_{k-1}(x)\|_\infty^2 \|J[F_{k-1}](x)\|_\infty$$

$$\le 3[W(B \vee 1)]^{3^{k-1}-1}(M \vee 1 + 1)^{3^{k-1}-1-2k+2}\|J[F_{k-1}](x)\|_\infty$$

$$\le 3[W(B \vee 1)]^{3^{k-1}-1}(M \vee 1 + 1)^{3^{k-1}-1-2k+2} \times 3^{k-2}[W(B \vee 1)]^{\frac{3^{k-1}-1}{2}}(M \vee 1 + 1)^{\frac{3^{k-1}-1}{2}-(k-1)^2}$$

$$= 3^{k-1}[W(B \vee 1)]^{\frac{3^k-3}{2}}(M \vee 1 + 1)^{\frac{3^k-1}{2}-k^2}$$

where we use the inductive hypothesis in the second last step. Taking supremum with respect to $l$ implies:

$$\|J[\eta_3 \circ F_{k-1}](x)\|_\infty = \sup_{1 \le l \le W} \left( \sum_{j=1}^{d} |3\eta_2(F_{k-1,l}(x)) \frac{\partial F_{k-1,l}(x)}{\partial x_j}| \right) \le 3^{k-1}[W(B \vee 1)]^{\frac{3^k-3}{2}}(M \vee 1 + 1)^{\frac{3^k-1}{2}-k^2}$$

Combining the two bounds derived above yields:

$$\|J[F_k](x)\|_\infty \le W(B \vee 1)\|J[\eta_3 \circ F_{k-1}](x)\|_\infty \le 3^{k-1}[W(B \vee 1)]^{\frac{3^k-1}{2}}(M \vee 1 + 1)^{\frac{3^k-1}{2}-k^2}$$

Therefore, the inequality also holds for $k$. By induction, the claim is proved. $\qquad\square$

**Lemma D.11.** *(Relation between the covering number of Jacobian/Gradient of elements in the DNN space and parameter space) For any $1 \le k \le L$, suppose that a pair of different two networks $F_k, G_k \in \mathcal{F}_{DNN}^k$ are given by:*

$$F_k(x) := (\mathcal{W}_F^{(k)}\eta_3(\cdot) + b_F^{(k)}) \cdots (\mathcal{W}_F^{(1)}x + b_F^{(1)})$$
$$G_k(x) := (\mathcal{W}_G^{(k)}\eta_3(\cdot) + b_G^{(k)}) \cdots (\mathcal{W}_G^{(1)}x + b_G^{(1)})$$

*Furthermore, assume that the $\| \ \|_\infty$ norm of the distance between the parameter spaces are upper bounded by $\delta$, i.e*

$$\|W_F^{(l)} - W_G^{(l)}\|_{\infty,\infty} \le \delta, \ \|b_F^{(l)} - b_G^{(l)}\|_\infty \le \delta \ (\forall 1 \le l \le k)$$

*Then we have that:*

$$\sup_{x \in \Omega, F_k, G_k \in \mathcal{F}_{DNN}^k} \|J[F_k](x) - J[G_k](x)\|_\infty \le \delta 3^{2k-2}[W(B \vee 1)]^{\frac{3^k-1}{2}}(M \vee 1 + 1)^{\frac{3^k-1}{2}-k^2+2k}$$

(D.22)

*Proof.* We use induction on $k$ to prove the claim.
Base case: When $k = 1$, we have that for any $x \in \Omega$ and any $F_1, G_1 \in \mathcal{F}_{DNN}^1$, the following holds:

$$\|J[F_1](x) - J[G_1](x)\|_\infty = \|\mathcal{W}_F^{(1)} - \mathcal{W}_G^{(1)}\|_\infty \le \delta W \le \delta W(B \vee 1)(M \vee 1 + 1)^2$$

Inductive Step: assume that the claim has been proved for $k - 1$. Then for any $x \in \Omega$ and $F_k, G_k \in \mathcal{F}_{DNN}^k$, difference between the two Jacobian matrices $J[F_k](x)$ and $J[G_k](x)$ can be written as:

$$\|J[F_k](x) - J[G_k](x)\|_\infty = \|\mathcal{W}_F^{(k)}J[\eta_3 \circ F_{k-1}](x) - \mathcal{W}_G^{(k)}J[\eta_3 \circ G_{k-1}](x)\|_\infty$$
$$\le \|\mathcal{W}_F^{(k)}J[\eta_3 \circ F_{k-1}](x) - \mathcal{W}_G^{(k)}J[\eta_3 \circ F_{k-1}](x)\|_\infty + \|\mathcal{W}_G^{(k)}J[\eta_3 \circ F_{k-1}](x) - \mathcal{W}_G^{(k)}J[\eta_3 \circ G_{k-1}](x)\|_\infty$$
$$\le \|\mathcal{W}_F^{(k)} - \mathcal{W}_G^{(k)}\|_\infty \|J[\eta_3 \circ F_{k-1}](x)\|_\infty + \|\mathcal{W}_G^{(k)}\|_\infty \|J[\eta_3 \circ F_{k-1}](x) - J[\eta_3 \circ G_{k-1}](x)\|_\infty$$
$$\le \delta W \|J[\eta_3 \circ F_{k-1}](x)\|_\infty + BW \|J[\eta_3 \circ F_{k-1}](x) - J[\eta_3 \circ G_{k-1}](x)\|_\infty$$

Using what we have calculated before, the first term can be upper bounded by:

$$\delta W \|J[\eta_3 \circ F_{k-1}](x)\|_\infty \le \delta W 3^{k-1}[W(B \vee 1)]^{\frac{3^k-3}{2}}(M \vee 1 + 1)^{\frac{3^k-1}{2}-k^2}$$

Moreover, using the definition of Jacobian matrix again, we can deduce that:

$$\|J[\eta_3 \circ F_{k-1}](x) - J[\eta_3 \circ G_{k-1}](x)\|_\infty$$
$$= \sup_{1 \le l \le W} \left( \sum_{j=1}^{d} |3\eta_2(F_{k-1,l}(x)) \frac{\partial F_{k-1,l}(x)}{\partial x_j} - 3\eta_2(G_{k-1,l}(x)) \frac{\partial G_{k-1,l}(x)}{\partial x_j}| \right)$$

For any index $1 \leq l \leq W$, the summation above can be upper bounded by:

$$\sum_{j=1}^{d} |3\eta_2(F_{k-1,l}(x))\frac{\partial F_{k-1,l}(x)}{\partial x_j} - 3\eta_2(G_{k-1,l}(x))\frac{\partial G_{k-1,l}(x)}{\partial x_j}|$$

$$\leq \sum_{j=1}^{d} |3\eta_2(F_{k-1,l}(x))\frac{\partial F_{k-1,l}(x)}{\partial x_j} - 3\eta_2(G_{k-1,l}(x))\frac{\partial F_{k-1,l}(x)}{\partial x_j}|$$

$$+ \sum_{j=1}^{d} |3\eta_2(G_{k-1,l}(x))\frac{\partial F_{k-1,l}(x)}{\partial x_j} - 3\eta_2(G_{k-1,l}(x))\frac{\partial G_{k-1,l}(x)}{\partial x_j}|$$

$$\leq \sum_{j=1}^{d} |3\eta_2(F_{k-1,l}(x)) - 3\eta_2(G_{k-1,l}(x))\|\frac{\partial F_{k-1,l}(x)}{\partial x_j}|$$

$$+ \sum_{j=1}^{d} |3\eta_2(G_{k-1,l}(x))\|\frac{\partial F_{k-1,l}(x)}{\partial x_j} - \frac{\partial G_{k-1,l}(x)}{\partial x_j}|$$

$$\leq 6(\sup_{x \in \Omega,\ F_{k-1} \in \mathcal{F}_{\text{DNN}}^{k-1}} \|F_{k-1}(x)\|_\infty)\|F_{k-1}(x) - G_{k-1}(x)\|_\infty \sum_{j=1}^{d} |\frac{\partial F_{k-1,l}(x)}{\partial x_j}|$$

$$+ 3(\sup_{x \in \Omega,\ G_{k-1} \in \mathcal{F}_{\text{DNN}}^{k-1}} \|G_{k-1}(x)\|_\infty)^2 \sum_{j=1}^{d} |\frac{\partial F_{k-1,l}(x)}{\partial x_j} - \frac{\partial G_{k-1,l}(x)}{\partial x_j}| =: S_1 + S_2$$

Now let's consider using what we have calculated above to upper bound the two summations. On the one hand, the first summation $S_1$ can be upper bounded as follows:

$$S_1 \leq 6(\sup_{x \in \Omega,\ F_{k-1} \in \mathcal{F}_{\text{DNN}}^{k-1}} \|F_{k-1}(x)\|_\infty)\|F_{k-1}(x) - G_{k-1}(x)\|_\infty \|J[F_{k-1}](x)\|_\infty$$

$$\leq 6[W(B \vee 1)]^{\frac{3^{k-1}-1}{2}}(M \vee 1 + 1)^{\frac{3^{k-1}-1}{2}-k+2} \times \delta 3^{k-2}[W(B \vee 1)]^{\frac{3^{k-1}-1}{2}}(M \vee 1 + 1)^{\frac{3^{k-1}-1}{2}-k+3}$$

$$\times 3^{k-2}[W(B \vee 1)]^{\frac{3^{k-1}-1}{2}}(M \vee 1 + 1)^{\frac{3^{k-1}-1}{2}-k^2+2k-1}$$

$$= 2\delta \times 3^{2k-3}[W(B \vee 1)]^{\frac{3^k-3}{2}}(M \vee 1 + 1)^{\frac{3^k-3}{2}-k^2+4}$$

On the other hand, applying the inductive hypothesis helps us upper bound the second summation as follows:

$$S_2 \leq 3(\sup_{x \in \Omega,\ G_{k-1} \in \mathcal{F}_{\text{DNN}}^{k-1}} \|G_{k-1}(x)\|_\infty)^2 \|J[F_{k-1}](x) - J[G_{k-1}](x)\|_\infty$$

$$\leq 3[W(B \vee 1)]^{3^{k-1}-1}(M \vee 1 + 1)^{3^{k-1}-1-2k+4} \times \delta 3^{2k-4}[W(B \vee 1)]^{\frac{3^{k-1}-1}{2}}(M \vee 1 + 1)^{\frac{3^{k-1}-1}{2}-k^2+4k-3}$$

$$= \delta \times 3^{2k-3}[W(B \vee 1)]^{\frac{3^k-3}{2}}(M \vee 1 + 1)^{\frac{3^k-1}{2}-k^2+2k}$$

Combining all bounds proved above yields:

$$\|J[F_k](x) - J[G_k](x)\|_\infty \leq \delta W 3^{k-1}[W(B \vee 1)]^{\frac{3^k-3}{2}}(M \vee 1 + 1)^{\frac{3^k-1}{2}-k^2} + WB(S_1 + S_2)$$

$$\leq \delta W 3^{k-1}[W(B \vee 1)]^{\frac{3^k-3}{2}}(M \vee 1 + 1)^{\frac{3^k-1}{2}-k^2} + 2\delta \times 3^{2k-3}[W(B \vee 1)]^{\frac{3^k-1}{2}}(M \vee 1 + 1)^{\frac{3^k-3}{2}-k^2+4}$$

$$+ \delta \times 3^{2k-3}[W(B \vee 1)]^{\frac{3^k-1}{2}}(M \vee 1 + 1)^{\frac{3^k-1}{2}-k^2+2k}$$

$$\leq \delta 3^{2k-2}[W(B \vee 1)]^{\frac{3^k-1}{2}}(M \vee 1 + 1)^{\frac{3^k-1}{2}-k^2+2k}\ (k \geq 2)$$

By induction, this completes our proof of the claim. $\qquad\square$

**Lemma D.12.** *Given any two row vectors $\boldsymbol{u}, \boldsymbol{v} \in \mathbb{R}^{1 \times d}$, we have:*

$$\left|\|\boldsymbol{u}\|_2 - \|\boldsymbol{v}\|_2\right| \leq \sqrt{d}\|\boldsymbol{u} - \boldsymbol{v}\|_\infty$$

*Proof.* Assume that the two vectors $\boldsymbol{u}, \boldsymbol{v} \in \mathbb{R}^d$ can be explicitly written as $\boldsymbol{u} = [u_1, u_2, \cdots, u_d]$ and $v = [v_1, v_2, \cdots, v_d]$, respectively. By applying Cauchy-Schwarz inequality, we have:

$$
\left| \|\boldsymbol{u}\|_2 - \|\boldsymbol{v}\|_2 \right|^2 = \left| \sqrt{\sum_{i=1}^{d} u_i^2} - \sqrt{\sum_{i=1}^{d} v_i^2} \right|^2
$$

$$
= \sum_{i=1}^{d} u_i^2 + \sum_{i=1}^{d} v_i^2 - 2\sqrt{\sum_{i=1}^{d} u_i^2} \sqrt{\sum_{i=1}^{d} v_i^2}
$$

$$
\leq \sum_{i=1}^{d} u_i^2 + \sum_{i=1}^{d} v_i^2 - 2\sum_{i=1}^{d} u_i v_i = \sum_{i=1}^{d} |u_i - v_i|^2
$$

$$
\leq d \max_{1 \leq i \leq d} |u_i - v_i|^2 = d\|\boldsymbol{u} - \boldsymbol{v}\|_\infty^2
$$

Taking the square root on both sides yields the desired inequality. $\qquad\square$

**Theorem D.5.** *(Bounding the DNN gradient space covering number) Fix some sufficiently large* $N \in \mathbb{Z}^+$. *Consider a Deep Neural Network space* $\Phi(L, W, S, B)$ *with* $L = O(1), W = O(N), S = O(N)$ *and* $B = O(N)$. *Then the* $\log$ *value of the covering number of this DNN space with respect to the* $\|\cdot\|_\infty$ *norm, which is denoted by* $\mathcal{N}(\delta, \nabla\Phi(L, W, S, B), \|\cdot\|_\infty)$, *can be bounded by:*

$$
\log\mathcal{N}(\delta, \nabla\Phi(L, W, S, B), \|\cdot\|_\infty) = O\left(S\left[\log(\delta^{-1}) + 3^L \log(WB)\right]\right) \qquad \text{(D.23)}
$$

*Proof.* We firstly fix a sparsity pattern (i.e, the locations of the non-zero entries are fixed). By picking $k = L$ in Lemma D.9, we get the following upper bound on the covering number with respect to $\|\cdot\|_\infty$:

$$
\left(\frac{\delta}{3^{L-1}[W(B \vee d)]^{\frac{3^L-1}{2}} 2^{\frac{3^L-1}{2} - L + 2}}\right)^{-S}
$$

Furthermore, note that the number of feasible configurations is upper bounded by:

$$
\binom{(W+1)^L}{S} \leq (W+1)^{LS}
$$

Multiplying the two bounds above yields:

$$
\log\mathcal{N}(\delta, \Phi(L, W, S, B), \|\cdot\|_\infty) \leq \log\left[(W+1)^{LS}\left\{\frac{\delta}{3^{L-1}[W(B \vee d)]^{\frac{3^L-1}{2}} 2^{\frac{3^L-1}{2} - L + 2}}\right\}^{-S}\right]
$$

$$
\leq S\log\left[\delta^{-1}(W+1)^L 3^{L-1}[W(B \vee d)]^{\frac{3^L-1}{2}} 2^{\frac{3^L-1}{2} - L + 2}\right]
$$

$$
\lesssim S\left[\log(\delta^{-1}) + L\log(3W) + 3^L \log(W(B \vee d)) + 3^L \log 2\right]
$$

Note that here the dimension $d$ is some constant. Thus, by plugging in thee given magnitudes $L = O(1), W = O(N), S = O(N)$ and $B = O(N)$, we can further deduce that:

$$
\log\mathcal{N}(\delta, \Phi(L, W, S, B), \|\cdot\|_\infty) \lesssim S\left[\log(\delta^{-1}) + 3^L \log(WB)\right]
$$

This finishes our proof. $\qquad\square$

**Lemma D.13.** *(Relation between the covering number of the DNN Laplacian space and parameter space) For any* $1 \leq k \leq L$, *suppose that a pair of different two networks* $F_k, G_k \in \Phi_k(L, W, S, B)$ *are given by:*

$$
F_k(x) := (\mathcal{W}_F^{(k)}\eta_3(\cdot) + b_F^{(k)}) \cdots (\mathcal{W}_F^{(1)}x + b_F^{(1)}),
$$

$$
G_k(x) := (\mathcal{W}_G^{(k)}\eta_3(\cdot) + b_G^{(k)}) \cdots (\mathcal{W}_G^{(1)}x + b_G^{(1)}).
$$

*Furthermore, assume that the $\|\ \|_\infty$ norm of the distance between the parameter spaces is uniformly upper bounded by $\delta$, i.e*

$$\|W_F^{(l)} - W_G^{(l)}\|_{\infty,\infty} \le \delta, \ \|b_F^{(l)} - b_G^{(l)}\|_\infty \le \delta, \ (\forall\, 1 \le l \le k). \tag{D.24}$$

*Then we have:*

$$\sup_{x \in \Omega} \|\Delta[F_k](x) - \Delta[G_k](x)\|_\infty = O\left(\delta W^{\frac{3^{k-1}-1}{2}}(B \vee d)^{\frac{5 \cdot 3^{k-1}-1}{2}}\right). \tag{D.25}$$

*Proof.* We use induction on $k$ to prove the claim.

Base case: $k = 1$. Note that any $F_1 \in \Phi_1(L, W, S, B)$ is a linear transform, so the Laplacian $\Delta[F_1](x)$ must be the zero vector for any $x \in \Omega$. Hence, for any $x \in \Omega$ and any $F_1, G_1 \in \Phi_1(L, W, S, B)$, we have:

$$\|\Delta[F_1](x) - \Delta[G_1](x)\|_\infty = 0 \lesssim \delta(B \vee d)^2.$$

Inductive Step: assume that the claim has been proved for $k - 1$, where $2 \le k \le L$. Then for any $x \in \Omega$ and $F_k, G_k \in \Phi_k(L, W, S, B)$ satisfying constraint D.24, applying linearity of the Laplacian operator indicates:

$$\begin{aligned}
\|\Delta[F_k](x) - \Delta[G_k](x)\|_\infty &= \|\mathcal{W}_F^{(k)}\Delta[\eta_3 \circ F_{k-1}](x) - \mathcal{W}_G^{(k)}\Delta[\eta_3 \circ G_{k-1}](x)\|_\infty \\
&= \left\|\left(\mathcal{W}_F^{(k)} - \mathcal{W}_G^{(k)}\right)\Delta[\eta_3 \circ F_{k-1}](x)\right\|_\infty \\
&\quad + \left\|\mathcal{W}_G^{(k)}\left(\Delta[\eta_3 \circ F_{k-1}](x) - \Delta[\eta_3 \circ G_{k-1}](x)\right)\right\|_\infty \\
&\le \|\mathcal{W}_F^{(k)} - \mathcal{W}_G^{(k)}\|_\infty\|\Delta[\eta_3 \circ F_{k-1}](x)\|_\infty \\
&\quad + \|\mathcal{W}_G^{(k)}\|_\infty\|\Delta[\eta_3 \circ F_{k-1}](x) - \Delta[\eta_3 \circ G_{k-1}](x)\|_\infty.
\end{aligned}$$

For the first term $\|\mathcal{W}_F^{(k)} - \mathcal{W}_G^{(k)}\|_\infty\|\Delta[\eta_3 \circ F_{k-1}](x)\|_\infty$, applying the bound in equation **??** and equation D.24 yields:

$$\begin{aligned}
\|\mathcal{W}_F^{(k)} - \mathcal{W}_G^{(k)}\|_\infty\|\Delta[\eta_3 \circ F_{k-1}](x)\|_\infty &\lesssim \delta W \times W^{\frac{3^{k-1}-3}{2}}(B \vee d)^{\frac{5 \cdot 3^{k-1}-3}{2}} \\
&= \delta W^{\frac{3^{k-1}-1}{2}}(B \vee d)^{\frac{5 \cdot 3^{k-1}-3}{2}}.
\end{aligned} \tag{D.26}$$

For the second term $\|\mathcal{W}_G^{(k)}\|_\infty\|\Delta[\eta_3 \circ F_{k-1}](x) - \Delta[\eta_3 \circ G_{k-1}](x)\|_\infty$, we need to upper bound the norm $\|\Delta[\eta_3 \circ F_{k-1}](x) - \Delta[\eta_3 \circ G_{k-1}](x)\|_\infty$ at first. Note that for any $1 \le l \le W$, we can use equation **??** to write the $l$-th component of $\Delta[\eta_3 \circ F_{k-1}](x) - \Delta[\eta_3 \circ G_{k-1}](x)$ as:

$$\begin{aligned}
\left(\Delta[\eta_3 \circ F_{k-1}](x) - \Delta[\eta_3 \circ G_{k-1}](x)\right)_l &= \sum_{j=1}^{d}\frac{\partial^2}{\partial x_j^2}\eta_3[F_{k-1,l}(x)] - \sum_{j=1}^{d}\frac{\partial^2}{\partial x_j^2}\eta_3[G_{k-1,l}(x)] \\
&= 6\eta_1[F_{k-1,l}(x)]\sum_{j=1}^{d}\left(\frac{\partial}{\partial x_j}F_{k-1,l}(x)\right)^2 - 6\eta_1[G_{k-1,l}(x)]\sum_{j=1}^{d}\left(\frac{\partial}{\partial x_j}G_{k-1,l}(x)\right)^2 \\
&\quad + 3\eta_2[F_{k-1,l}(x)]\sum_{j=1}^{d}\frac{\partial^2}{\partial x_j^2}F_{k-1,l}(x) - 3\eta_2[G_{k-1,l}(x)]\sum_{j=1}^{d}\frac{\partial^2}{\partial x_j^2}G_{k-1,l}(x) \\
&= 6\eta_1[F_{k-1,l}(x)]\sum_{j=1}^{d}\left(\frac{\partial}{\partial x_j}F_{k-1,l}(x)\right)^2 - 6\eta_1[G_{k-1,l}(x)]\sum_{j=1}^{d}\left(\frac{\partial}{\partial x_j}F_{k-1,l}(x)\right)^2 \\
&\quad + 6\eta_1[G_{k-1,l}(x)]\sum_{j=1}^{d}\left(\frac{\partial}{\partial x_j}F_{k-1,l}(x)\right)^2 - 6\eta_1[G_{k-1,l}(x)]\sum_{j=1}^{d}\left(\frac{\partial}{\partial x_j}G_{k-1,l}(x)\right)^2 \\
&\quad + 3\eta_2[F_{k-1,l}(x)]\sum_{j=1}^{d}\frac{\partial^2}{\partial x_j^2}F_{k-1,l}(x) - 3\eta_2[G_{k-1,l}(x)]\sum_{j=1}^{d}\frac{\partial^2}{\partial x_j^2}F_{k-1,l}(x) \\
&\quad + 3\eta_2[G_{k-1,l}(x)]\sum_{j=1}^{d}\frac{\partial^2}{\partial x_j^2}F_{k-1,l}(x) - 3\eta_2[G_{k-1,l}(x)]\sum_{j=1}^{d}\frac{\partial^2}{\partial x_j^2}G_{k-1,l}(x).
\end{aligned}$$

We denote the four summations above by $V_1, V_2, V_3$ and $V_4$, respectively:

$$V_1 := 6\eta_1[F_{k-1,l}(x)] \sum_{j=1}^{d} \left(\frac{\partial}{\partial x_j} F_{k-1,l}(x)\right)^2 - 6\eta_1[G_{k-1,l}(x)] \sum_{j=1}^{d} \left(\frac{\partial}{\partial x_j} F_{k-1,l}(x)\right)^2,$$

$$V_2 := 6\eta_1[G_{k-1,l}(x)] \sum_{j=1}^{d} \left(\frac{\partial}{\partial x_j} F_{k-1,l}(x)\right)^2 - 6\eta_1[G_{k-1,l}(x)] \sum_{j=1}^{d} \left(\frac{\partial}{\partial x_j} G_{k-1,l}(x)\right)^2,$$

$$V_3 := 3\eta_2[F_{k-1,l}(x)] \sum_{j=1}^{d} \frac{\partial^2}{\partial x_j^2} F_{k-1,l}(x) - 3\eta_2[G_{k-1,l}(x)] \sum_{j=1}^{d} \frac{\partial^2}{\partial x_j^2} F_{k-1,l}(x),$$

$$V_4 := 3\eta_2[G_{k-1,l}(x)] \sum_{j=1}^{d} \frac{\partial^2}{\partial x_j^2} F_{k-1,l}(x) - 3\eta_2[G_{k-1,l}(x)] \sum_{j=1}^{d} \frac{\partial^2}{\partial x_j^2} G_{k-1,l}(x).$$

By applying Lemma **??**, Lemma D.8, Lemma D.9 and Lemma **??**, we can upper bound $V_1$ by:

$$V_1 = 6\Big(\eta_1[F_{k-1,l}(x)] - \eta_1[G_{k-1,l}(x)]\Big) \sum_{j=1}^{d} \left(\frac{\partial}{\partial x_j} F_{k-1,l}(x)\right)^2$$

$$\leq 6|F_{k-1,l}(x) - G_{k-1,l}(x)| \left(\sum_{j=1}^{d} \left|\frac{\partial}{\partial x_j} F_{k-1,l}(x)\right|\right)^2 \leq 6\|F_{k-1}(x) - G_{k-1}(x)\|_\infty \|J[F_{k-1}](x)\|_\infty^2$$

$$\lesssim \delta W^{\frac{3^{k-2}-1}{2}}(B \vee d)^{\frac{5 \cdot 3^{k-2}-1}{2}} 2^{\frac{3^{k-1}-1}{2}-k+2} 3^{k-2} \times W^{3^{k-2}-1}(B \vee d)^{5 \cdot 3^{k-2}-1} 2^{3^{k-1}-1-2k+4} 3^{2k-4}$$

$$\lesssim \delta W^{\frac{3^{k-1}-3}{2}}(B \vee d)^{\frac{5 \cdot 3^{k-1}-3}{2}}.$$

where the last step above follows from $k \leq L$ and $L = O(1)$.

Furthermore, note that for any $1 \leq j \leq d$, we can upper bound the difference $\left(\frac{\partial}{\partial x_j} F_{k-1,l}(x)\right)^2 - \left(\frac{\partial}{\partial x_j} G_{k-1,l}(x)\right)^2$ as follows:

$$\left(\frac{\partial}{\partial x_j} F_{k-1,l}(x)\right)^2 - \left(\frac{\partial}{\partial x_j} G_{k-1,l}(x)\right)^2 \leq \left|\left(\frac{\partial}{\partial x_j} F_{k-1,l}(x)\right)^2 - \left(\frac{\partial}{\partial x_j} G_{k-1,l}(x)\right)^2\right|$$

$$= \left|\frac{\partial}{\partial x_j} F_{k-1,l}(x) + \frac{\partial}{\partial x_j} G_{k-1,l}(x)\right| \left|\frac{\partial}{\partial x_j} F_{k-1,l}(x) - \frac{\partial}{\partial x_j} G_{k-1,l}(x)\right| \tag{D.27}$$

$$\leq \left(\left|\frac{\partial}{\partial x_j} F_{k-1,l}(x)\right| + \left|\frac{\partial}{\partial x_j} G_{k-1,l}(x)\right|\right) \left|\frac{\partial}{\partial x_j} F_{k-1,l}(x) - \frac{\partial}{\partial x_j} G_{k-1,l}(x)\right|.$$

Note that $\eta_1(G_{k-1,l}(x)) \geq 0$. Combining the non-negativity with equation D.27, Lemma **??**, Lemma **??** and Lemma **??** helps us upper bound $V_2$ by:

$$V_2 = 6\eta_1[G_{k-1,l}(x)] \sum_{j=1}^{d} \left[\left(\frac{\partial}{\partial x_j} F_{k-1,l}(x)\right)^2 - \left(\frac{\partial}{\partial x_j} G_{k-1,l}(x)\right)^2\right]$$

$$\leq 6\|G_{k-1}(x)\|_\infty \sum_{j=1}^{d} \left(\left|\frac{\partial}{\partial x_j} F_{k-1,l}(x)\right| + \left|\frac{\partial}{\partial x_j} G_{k-1,l}(x)\right|\right) \left|\frac{\partial}{\partial x_j} F_{k-1,l}(x) - \frac{\partial}{\partial x_j} G_{k-1,l}(x)\right|$$

$$\leq 6\|G_{k-1}(x)\|_\infty \left(\sum_{j=1}^{d} \left|\frac{\partial}{\partial x_j} F_{k-1,l}(x)\right| + \sum_{j=1}^{d} \left|\frac{\partial}{\partial x_j} G_{k-1,l}(x)\right|\right) \left(\sum_{j=1}^{d} \left|\frac{\partial}{\partial x_j} F_{k-1,l}(x) - \frac{\partial}{\partial x_j} G_{k-1,l}(x)\right|\right)$$

$$\leq 6\|G_{k-1}(x)\|_\infty \Big(\|J[F_{k-1}](x)\|_\infty + \|J[G_{k-1}](x)\|_\infty\Big) \Big\|J[F_{k-1}](x) - J[G_{k-1}](x)\Big\|_\infty$$

$$\leq 6W^{\frac{3^{k-2}-1}{2}}(B \vee d)^{\frac{5 \cdot 3^{k-2}-1}{2}} 2^{\frac{3^{k-1}-1}{2}-k+2} \times 2W^{\frac{3^{k-2}-1}{2}}(B \vee d)^{\frac{5 \cdot 3^{k-2}-1}{2}} 2^{\frac{3^{k-1}-1}{2}-k+2} 3^{k-2}$$

$$\times \delta W^{\frac{3^{k-2}-1}{2}}(B \vee d)^{\frac{5 \cdot 3^{k-2}-1}{2}} 2^{\frac{3^{k-1}-1}{2}-k+1} 3^{2k-4} \lesssim \delta W^{\frac{3^{k-1}-3}{2}}(B \vee d)^{\frac{5 \cdot 3^{k-1}-3}{2}}.$$

where the last step above follows from $k \leq L$ and $L = O(1)$.

Moreover, using Lemma **??**, Lemma D.8 and Lemma **??** helps us upper bound $V_3$ by:

$$V_3 = \Big(3\eta_2[F_{k-1,l}(x)] - 3\eta_2[G_{k-1,l}(x)]\Big)\sum_{j=1}^{d}\frac{\partial^2}{\partial x_j^2}F_{k-1,l}(x)$$

$$\leq \Big|3\eta_2[F_{k-1,l}(x)] - 3\eta_2[G_{k-1,l}(x)]\Big|\Big|\sum_{j=1}^{d}\frac{\partial^2}{\partial x_j^2}F_{k-1,l}(x)\Big|$$

$$\leq 6\Big(\sup_{x\in\Omega,\ F_{k-1}\in\Phi_{k-1}(L,W,S,B)}\|F_{k-1}(x)\|_\infty\Big)\|F_{k-1}(x) - G_{k-1}(x)\|_\infty\|\Delta[F_{k-1}](x)\|_\infty$$

$$\lesssim 6W^{\frac{3^{k-2}-1}{2}}(B\vee d)^{\frac{5\cdot3^{k-2}-1}{2}}2^{\frac{3^{k-1}-1}{2}-k+2}\times\delta W^{\frac{3^{k-2}-1}{2}}(B\vee d)^{\frac{5\cdot3^{k-2}-1}{2}}2^{\frac{3^{k-1}-1}{2}-k+2}3^{k-2}$$

$$\times W^{\frac{3^{k-2}-1}{2}}(B\vee d)^{\frac{5\cdot3^{k-2}-1}{2}}\lesssim\delta W^{\frac{3^{k-1}-3}{2}}(B\vee d)^{\frac{5\cdot3^{k-1}-3}{2}}$$

where the last step above follows from $k \leq L$ and $L = O(1)$.

Finally, applying Lemma **??** and inductive hypothesis helps us upper bound $V_4$ by:

$$V_4 = 3\eta_2[G_{k-1,l}(x)]\Big(\sum_{j=1}^{d}\frac{\partial^2}{\partial x_j^2}F_{k-1,l}(x) - \sum_{j=1}^{d}\frac{\partial^2}{\partial x_j^2}G_{k-1,l}(x)\Big)$$

$$\leq 3\|G_{k-1}(x)\|_\infty^2\|\Delta[F_{k-1}](x) - \Delta[G_{k-1}](x)\|_\infty$$

$$\lesssim 3W^{3^{k-2}-1}(B\vee d)^{5\cdot3^{k-2}-1}2^{3^{k-1}-1-2k+4}\times\delta W^{\frac{3^{k-2}-1}{2}}(B\vee d)^{\frac{5\cdot3^{k-2}-1}{2}}$$

$$\lesssim\delta W^{\frac{3^{k-1}-3}{2}}(B\vee d)^{\frac{5\cdot3^{k-1}-3}{2}}$$

where the last step above follows from $k \leq L$ and $L = O(1)$.

Combining the four bounds on $V_1, V_2, V_3$ and $V_4$ implies:

$$\Big(\Delta[\eta_3\circ F_{k-1}](x) - \Delta[\eta_3\circ G_{k-1}](x)\Big)_l = \sum_{i=1}^{4}V_i \lesssim \delta W^{\frac{3^{k-1}-3}{2}}(B\vee d)^{\frac{5\cdot3^{k-1}-3}{2}}$$

Taking supremum with respect to $1 \leq l \leq W$ gives us an upper bound on the second term $\|\mathcal{W}_G^{(k)}\|_\infty\|\Delta[\eta_3\circ F_{k-1}](x) - \Delta[\eta_3\circ G_{k-1}](x)\|_\infty$:

$$\|\mathcal{W}_G^{(k)}\|_\infty\|\Delta[\eta_3\circ F_{k-1}](x) - \Delta[\eta_3\circ G_{k-1}](x)\|_\infty \lesssim WB\times\delta W^{\frac{3^{k-1}-3}{2}}(B\vee d)^{\frac{5\cdot3^{k-1}-3}{2}}$$

$$= \delta W^{\frac{3^{k-1}-1}{2}}(B\vee d)^{\frac{5\cdot3^{k-1}-1}{2}}$$

$$\tag{D.28}$$

Combining the two bounds derived in equation D.26 and equation D.28 then implies:

$$\|\Delta[F_k](x) - \Delta[G_k](x)\|_\infty \lesssim \delta W^{\frac{3^{k-1}-1}{2}}(B\vee d)^{\frac{5\cdot3^{k-1}-3}{2}} + \delta W^{\frac{3^{k-1}-1}{2}}(B\vee d)^{\frac{5\cdot3^{k-1}-1}{2}}$$

$$\lesssim \delta W^{\frac{3^{k-1}-1}{2}}(B\vee d)^{\frac{5\cdot3^{k-1}-1}{2}}$$

Taking supremum with respect to $x \in \Omega$ on the LHS implies that the given upper bound also holds for $k$. By induction, the claim is proved. $\square$

Given a DNN space $\Phi(L,W,S,B)$, we define a corresponding DNN Laplacian space $\Delta\Phi(L,W,S,B)$ as:

$$\Delta\Phi(L,W,S,B) := \{\Delta F \mid F \in \Phi(L,W,S,B)\}.\tag{D.29}$$

**Theorem D.6.** *(Bounding the DNN Laplacian space covering number) Fix some sufficiently large $N \in \mathbb{Z}^+$. Consider a Deep Neural Network space $\Phi(L,W,S,B)$ with $L = O(1), W = O(N), S = O(N)$ and $B = O(N)$. Then the* log *value of the covering number of the DNN Laplacian space with respect to the $\|\cdot\|_\infty$ norm $\|F(x)\|_\infty := \sup_{x\in\Omega}|F(x)|$, which is denoted by $\mathcal{N}(\delta,\Delta\Phi(L,W,S,B),\|\cdot\|_\infty)$, can be upper bounded by:*

$$\log\mathcal{N}(\delta,\Delta\Phi(L,W,S,B),\|\cdot\|_\infty) = O\Big(S\Big[\log(\delta^{-1}) + 3^L\log(WB)\Big]\Big)\tag{D.30}$$

*Proof.* We firstly fix a sparsity pattern (i.e, the locations of the non-zero entries are fixed). Applying Lemma D.13 yields that there exists some constant $C = O(1)$, such that the covering number with respect to $\|\cdot\|_\infty$ can be upper bounded by:

$$\left(\frac{\delta}{CW^{\frac{3^{L-1}-1}{2}}(B \vee d)^{\frac{5\cdot3^{L-1}-1}{2}}}\right)^{-S}$$

Furthermore, note that the number of feasible configurations is upper bounded by:

$$\binom{(W+1)^L}{S} \leq (W+1)^{LS}$$

Multiplying the two bounds above yields:

$$\log\mathcal{N}(\delta, \Phi(L,W,S,B), \|\cdot\|_\infty) \leq \log\left[(W+1)^{LS}\left(\frac{\delta}{CW^{\frac{3^{L-1}-1}{2}}(B\vee d)^{\frac{5\cdot3^{L-1}-1}{2}}}\right)^{-S}\right]$$

$$\leq S\log\left[\delta^{-1}(W+1)^L W^{\frac{3^{L-1}-1}{2}}(B\vee d)^{\frac{5\cdot3^{L-1}-1}{2}}\right]$$

$$\lesssim S\left[\log(\delta^{-1}) + L\log(W) + 3^L\log(W(B\vee d))\right]$$

Note that here the dimension $d$ is some constant. Thus, by plugging in thee given magnitudes $L = O(1), W = O(N), S = O(N)$ and $B = O(N)$, we can further deduce that:

$$\log\mathcal{N}(\delta, \Delta\Phi(L,W,S,B), \|\cdot\|_\infty) \lesssim S\left[\log(\delta^{-1}) + 3^L\log(WB)\right]$$

This finishes our proof. $\qquad\square$

**Lemma D.14** (Local Rademencher Complexity Bound for Deep Ritz Method). *At the the same time, for any $\rho > 0$, we assume the Rademencher complexity of a localized function space $\boldsymbol{S}_\rho(\Omega) :=$ $\left\{h := |\Omega|\cdot\left[\frac{1}{2}\left(|\nabla u|^2 - |\nabla u^*|^2\right) + \frac{1}{2}V(|u|^2 - |u^*|^2) - f(u - u^*)\right] \ \Big| \ \|u - u^*\|_{H_1}^2 \leq \rho, u \in$ $\Phi(L,W,S,B)\right\}$ can be upper bounded by a sub-root function*

$$\phi(r) := O\left(\sqrt{\frac{S3^L r}{n}\log(LBWn)}\right)$$

i.e. *we have*

$$\phi(4\rho) \leq 2\phi(\rho) \text{ and } R_n(\boldsymbol{S}_\rho(\Omega)) \leq \phi(\rho) \tag{D.31}$$

*holds for all constant $\rho > 0$*

*Proof.* We first apply the Talagrand Contraction Lemma D.2 to upper bound the local Rademacher complexity $R_n(\boldsymbol{S}_\rho(\Omega))$ as

$$R_n(\boldsymbol{S}_\rho(\Omega)) = \mathbb{E}_x\mathbb{E}_\sigma\left[\sup_{f \in \boldsymbol{F}}\frac{1}{n}\sum_{i=1}^n \sigma_i\left[\frac{1}{2}\left(|\nabla u|^2 - |\nabla u^*|^2\right) + \frac{1}{2}V(|u|^2 - |u^*|^2) - f(u - u^*)\right]\right]$$

$$\leq 2L\mathbb{E}_x\mathbb{E}_\sigma\left[\sup_{u \in S}\frac{1}{n}\sum_{i=1}^n \sigma_i\left(u(x_i) - u^*(x_i)\right)\right] + 2L\mathbb{E}_x\mathbb{E}_{\sigma'}\left[\sup_{u \in S}\frac{1}{n}\sum_{i=1}^n \sigma_i'\left(\|\nabla u(x_i) - \nabla u^*(x_i)\|_2\right)\right]$$

$$\leq 2L\mathbb{E}_x\mathbb{E}_\sigma\left[\sup_{u \in S}\frac{1}{n}\sum_{i=1}^n \sigma_i\left(u(x_i) - u_\xi^*(x_i)\right)\right] + 2L\mathbb{E}_x\mathbb{E}_{\sigma'}\left[\sup_{u \in S}\frac{1}{n}\sum_{i=1}^n \sigma_i'\left(\|\nabla u(x_i) - \nabla u_\xi^*(x_i)\|_2\right)\right].$$

We denote the localization set $L_\rho := \{u : u \in \phi(L,W,S,B), \|u - u^*\|_{H_1}^2 \leq \rho\}$, then we can bound the local Radmencher complexity using duley integral

$$\mathbb{E}_\eta R_n \boldsymbol{S}_\rho(\Omega) \leq C_L \left[ \mathbb{E}_\eta R_n \{u - u_* : u \in L_\rho\} + \mathbb{E}_\eta R_n \{\|\nabla u - \nabla u^*\| : u \in L_\rho\} \right]$$

$$\leq C_L \mathbb{E}_\eta R_n \{u - u_* : u \in \phi(L, W, S, B), \|u - u^*\|_n^2 \leq 2\rho\}$$

$$+ C_L \mathbb{E}_\eta R_n \{\|\nabla u - \nabla u^*\| : u \in \phi(L, W, S, B), \|\nabla u - \nabla u^*\|_n^2 \leq 2\rho\}$$

$$\lesssim C_L \inf_{0 < \alpha < \sqrt{2\rho_0}} \left\{ \alpha + \frac{1}{\sqrt{n}} \int_\alpha^{2r_0} \sqrt{\log \mathcal{N}(\delta, \Phi(L, W, S, B), \|\cdot\|_n)} d\delta \right\}$$

$$+ C_L \inf_{0 < \alpha < \sqrt{2\rho_0}} \left\{ \alpha + \frac{1}{\sqrt{n}} \int_\alpha^{2r_0} \sqrt{\log \mathcal{N}(\delta, \Phi(L, W, S, B), \|\nabla \cdot\|_n)} d\delta \right\}$$

$$\lesssim C_L \left[ \frac{1}{n} + \frac{1}{\sqrt{n}} \int_{\frac{1}{n}}^{c\sqrt{\frac{r}{\alpha}}} \sqrt{2SL \log(LBW\epsilon^{-1})} + \frac{1}{\sqrt{n}} \int_{\frac{1}{n}}^{c\sqrt{\frac{r}{\alpha}}} \sqrt{2S3^L \log(LBW\epsilon^{-1})} \right]$$

$$\lesssim \sqrt{\frac{S3^L r}{n} \log(LBWn)}$$

$\square$

**Lemma D.15** (Local Rademencher Complexity Bound for Physics Informed Neural Network). *At the the same time, for any $\rho > 0$, we assume the Rademencher complexity of a localized function space $\boldsymbol{S}_\rho(\Omega) := \left\{ h := |\Omega| \cdot \left[ (\Delta u - Vu + f)^2 - (\Delta u^* - Vu^* + f)^2 \right] \mid \|u - u^*\|_{H_2}^2 \leq \rho, u \in \Phi(L, W, S, B) \right\}$ can be upper bounded by a sub-root function*

$$\phi(r) := O\left( \sqrt{\frac{S3^L r}{n} \log(LBWn)} \right)$$

i.e. *we have*

$$\phi(4\rho) \leq 2\phi(\rho) \text{ and } R_n(\boldsymbol{S}_\rho(\Omega)) \leq \phi(\rho) \tag{D.32}$$

*holds for all constant $\rho > 0$*

*Proof.* We first apply the Talagrand Contraction Lemma D.2 to upper bound the local Rademacher complexity $R_n(\boldsymbol{S}_\rho(\Omega))$ as

$$R_n(\boldsymbol{S}_\rho(\Omega)) = \mathbb{E}_x \mathbb{E}_\sigma \left[ \sup_{f \in \boldsymbol{F}} \frac{1}{n} \sum_{i=1}^n \sigma_i \left[ (\Delta u - Vu + f)^2 - (\Delta u^* - Vu^* + f)^2 \right] \right]$$

$$\leq 2L \mathbb{E}_x \mathbb{E}_\sigma \left[ \sup_{u \in S} \frac{1}{n} \sum_{i=1}^n \sigma_i \left( u(x_i) - u^*(x_i) \right) \right] + 2L \mathbb{E}_x \mathbb{E}_{\sigma'} \left[ \sup_{u \in S} \frac{1}{n} \sum_{i=1}^n \sigma_i' \left( \|\Delta u(x_i) - \Delta u^*(x_i)\|_2 \right) \right]$$

$$\leq 2L \mathbb{E}_x \mathbb{E}_\sigma \left[ \sup_{u \in S} \frac{1}{n} \sum_{i=1}^n \sigma_i \left( u(x_i) - u_\xi^*(x_i) \right) \right] + 2L \mathbb{E}_x \mathbb{E}_{\sigma'} \left[ \sup_{u \in S} \frac{1}{n} \sum_{i=1}^n \sigma_i' \left( \|\Delta u(x_i) - \Delta u_\xi^*(x_i)\|_2 \right) \right].$$

We denote the localization set $L_\rho := \{u : u \in \phi(L, W, S, B), \|u - u^*\|_{H_1}^2 \leq \rho\}$, then we can bound the local Radmencher complexity using duley integral

$$\mathbb{E}_\eta R_n \boldsymbol{S}_\rho(\Omega) \leq C_L \left[\mathbb{E}_\eta R_n \{u - u_* : u \in L_\rho\} + \mathbb{E}_\eta R_n \{|\Delta u - \Delta u^*| : u \in L_\rho\}\right]$$

$$\leq C_L \mathbb{E}_\eta R_n \{u - u_* : u \in \phi(L, W, S, B), \|u - u^*\|_n^2 \leq 2\rho\}$$

$$+ C_L \mathbb{E}_\eta R_n \{|\Delta u - \Delta u^*| : u \in \phi(L, W, S, B), |\Delta u - \Delta u^*|_n^2 \leq 2\rho\}$$

$$\lesssim C_L \inf_{0 < \alpha < \sqrt{2\rho_0}} \left\{\alpha + \frac{1}{\sqrt{n}} \int_\alpha^{2r_0} \sqrt{\log \mathcal{N}(\delta, \Phi(L, W, S, B), \|\cdot\|_n)} d\delta\right\}$$

$$+ C_L \inf_{0 < \alpha < \sqrt{2\rho_0}} \left\{\alpha + \frac{1}{\sqrt{n}} \int_\alpha^{2r_0} \sqrt{\log \mathcal{N}(\delta, \Phi(L, W, S, B), \|\Delta \cdot\|_n)} d\delta\right\}$$

$$\lesssim C_L \left[\frac{1}{n} + \frac{1}{\sqrt{n}} \int_{\frac{1}{n}}^{c\sqrt{\frac{r}{\alpha}}} \sqrt{2SL \log(LBW\epsilon^{-1})} + \frac{1}{\sqrt{n}} \int_{\frac{1}{n}}^{c\sqrt{\frac{r}{\alpha}}} \sqrt{2SL \log(LBW\epsilon^{-1})}\right]$$

$$\lesssim \sqrt{\frac{S3^L r}{n} \log(LBWn)}$$

$\square$

## D.3 Auxiliary definitions and lemmata On Approximation Error

### D.3.1 Approximation using Truncated Fourier Basis

**Lemma D.16.** *Given $\alpha > 0$ and a fixed integer $\xi \in \mathbb{Z}^+$. For any function $f \in H^\alpha(\Omega)$, we let $f_\xi = \sum_{\|z\|_\infty \leq \xi} f_z \phi_z$ be the best approximation of $f$ in the space $F_\xi(\Omega)$. Then for any $0 < \beta \leq \alpha$, we have the following inequality:*

$$\|f - f_\xi\|_{H^\beta(\Omega)}^2 \leq \xi^{-2(\alpha-\beta)} \|f\|_{H^\alpha}^2.$$

*Proof.* For $f \in H^\alpha(\Omega)$, we know the Fourier coefficient satisfies

$$\sum_{\|z\|_\infty \geq \xi} |f_z|^2 \|z\|_2^{2\alpha} \lesssim \|f\|_{H^\alpha}^2.$$

We directly construct $f_\xi = \sum_{\|z\|_\infty \leq \xi} f_z \phi_z$ to be the truncated Fourier series of the function $f$, then we have

$$\|f - f_\xi\|_{H^\beta(\Omega)}^2 \lesssim \sum_{\|z\|_\infty \geq \xi} |f_z|^2 \|z\|_2^{2\beta} \leq \xi^{-2(\alpha-\beta)} \sum_{\|z\|_\infty \geq \xi} |f_z|^2 \|z\|_2^{2\alpha} \leq \xi^{-2(\alpha-\beta)} \|f\|_{H^\alpha}^2.$$

$\square$

### D.3.2 Approximation using Neural Network

In this section, we aim to provide approximation bound for deep neural network. Our proof of the approximation upper bound is based on the observation that the B-spline approximation[9, 53] can be formulated as a ReLU3 neural network efficiently[59, 17, 11, 25]. Although the proof of the approximation of the neural network to the Sobolev spaces is a standard approach, we still demonstrate the proof sketch here.

**Definition D.4.** *(Univariate and Multivariate B-splines) Fix an arbitrary integer $l \in \mathbb{Z}^+$. Consider a corresponding uniform partition $\pi_l$ of $[0, 1]$:*

$$\pi_l : 0 = t_0^{(l)} < t_1^{(l)} < \cdots < t_{l-1}^{(l)} < t_l^{(l)} = 1,$$

*where $t_i^{(l)} = \frac{i}{l}$ ($\forall 0 \leq i \leq l$). Now for any $k \in \mathbb{Z}^+$, we can define an extended partition $\pi_{l,k}$ as:*

$$\pi_{l,k} : t_{-k+1}^{(l)} = \cdots t_{-1}^{(l)} = 0 = t_0^{(l)} < t_1^{(l)} < \cdots < t_{l-1}^{(l)} < t_l^{(l)} = 1 = t_{l+1}^{(l)} = \cdots = t_{l+k-1}^{(l)}$$

*Based on the extended partition $\pi_{l,k}$, the univariate B-splines of order $k$ with respect to partition $\pi_l$ are defined by:*

$$N_{l,i}^{(k)}(x) := (-1)^k (t_{i+k}^{(l)} - t_i^{(l)}) \cdot \left[t_i^{(l)}, \cdots, t_{i+k}^{(l)}\right] \max\{(x - t), 0\}^{k-1}, \ x \in [0, 1], \ i \in I_{l,k} \quad \text{(D.33)}$$

*where $I_{l,k} = \{-k+1, -k+2, \cdots, l-1\}$ and $\left[t_i^{(l)}, \cdots, t_{i+k}^{(l)}\right]$ denotes the divided difference operator.*

*Equivalently, for any $x \in [0,1]$, we can rewrite the univariate B-splines $N_{l,i}^{(k)}(x)$ in an explicit form:*

$$N_{l,i}^{(k)}(x) = \begin{cases} \frac{l^{k-1}}{(k-1)!} \sum_{j=0}^{k}(-1)^j \binom{k}{j} \max\left\{x - \frac{i+j}{l}, 0\right\}^{k-1}, & (0 \leq i \leq l-k+1) \\ \sum_{j=0}^{k-1} a_{ij} \max\left\{x - \frac{j}{l}, 0\right\}^{k-1} + \sum_{n=1}^{k-2} b_{in}x^n + b_{i0}, & (-k+1 \leq i \leq 0) \\ \sum_{j=l-k+1}^{l} c_{ij} \max\left\{x - \frac{j}{l}, 0\right\}^{k-1}, & (l-k+1 \leq i \leq l-1) \end{cases} \quad \text{(D.34)}$$

*where $\{a_{ij} \mid -k+1 \leq i \leq 0, \ 0 \leq j \leq k-1\}$, $\{b_{in} \mid -k+1 \leq i \leq 0, \ 1 \leq n \leq k-2\}$ and $\{c_{ij} \mid l-k+1 \leq i \leq l-1, \ l-k+1 \leq j \leq l-1\}$ are some fixed constants.*

*For any index vector $\boldsymbol{i} = (i_1, i_2, \cdots, i_d) \in I_{l,k}^d$, we can define a corresponding multivariate B-spline as a product of univariate B-splines:*

$$N_{l,\boldsymbol{i}}^{(k)}(\boldsymbol{x}) := \Pi_{j=1}^{d} N_{l,i_j}^{(k)}(x_j). \quad \text{(D.35)}$$

**Definition D.5.** *(Interpolation Operator[53]) Take some domain $\Omega \subset [0,1]^d$ and two arbitrary integers $k, l \in \mathbb{Z}^+$. Consider the extended partition $\pi_{l,k}$ and the corresponding set of multivariate B-splines $\{N_{l,\boldsymbol{i}}^{(k)}(x)\}_{\boldsymbol{i} \in I_{l,k}^d}$ defined in Definition D.4. For any $\boldsymbol{i} \in I_{l,k}^d$, we define the domain $\Omega_{\boldsymbol{i}} := \{\boldsymbol{x} \in \Omega : x_j \in [t_{i_j}, t_{i_j+k}], \ 1 \leq j \leq d\}$. There exists a set of linear functionals $\{\lambda_{\boldsymbol{i}}\}_{\boldsymbol{i} \in I_{k,l}^d}$, where $\lambda_{\boldsymbol{i}} : L^1(\Omega) \to \mathbb{R}$ ($\forall \boldsymbol{i} \in I_{k,l}^d$), such that for any $\boldsymbol{i} \in I_{k,l}^d$ and $p \in [1, \infty]$, we have:*

$$\lambda_{\boldsymbol{i}}(N_{l,\boldsymbol{j}}^{(k)}) = \delta_{\boldsymbol{i},\boldsymbol{j}} \text{ and } |\lambda_{\boldsymbol{i}}(f)| \leq 9^{d(k-1)}(2k+1)^d \left(\frac{k}{l}\right)^{-\frac{d}{p}} \|f\|_{L^p(\Omega_{\boldsymbol{i}})}, \ \forall f \in L^p(\Omega). \quad \text{(D.36)}$$

*The corresponding interpolation operator $Q_{k,l}$ is defined as:*

$$Q_{k,l}f := \sum_{\boldsymbol{i} \in I_{k,l}^d} \lambda_{\boldsymbol{i}}(f) N_{l,\boldsymbol{i}}^{(k)}, \ \forall f \in L^1(\Omega).$$

**Theorem D.7.** *[[53]] Fix $f \in W^s(\Omega)$ with $\Omega \subseteq [0,1]^d, s \in \mathbb{Z}^+$ and $p \in [1, \infty)$. Then for any $k, l, r \in \mathbb{Z}^+$ with $k \geq s$ and $0 \leq r \leq s$, we have that there exists some constant $C = C(k, s, r, p, d)$, such that:*

$$\|f - Q_{k,l}f\|_{H^r(\Omega)} \leq C\left(\frac{1}{l}\right)^{s-r} \|f\|_{H^s(\Omega)}.$$

**Theorem D.8.** *(Approximation result of Deep Neural Network) Fix some dimension $d \in \mathbb{Z}^+$, some domain $\Omega \subseteq [0,1]^d$. Pick some $N \in \mathbb{Z}^+$ that is sufficiently large. Then for any $s, r \in \mathbb{Z}^+$ with $0 \leq r \leq s$ and any function $u^* \in H^s(\Omega)$, there exists some sparse Deep Neural Network $u_{DNN} \in \Phi(L, W, S, B)$ with $L = O(1), W = O(N), S = O(N), B = O(N)$, such that:*

$$\|u_{DNN} - u^*\|_{H^r(\Omega)} \lesssim N^{-\frac{s-r}{d}} \|u^*\|_{H^s(\Omega)}. \quad \text{(D.37)}$$

*Proof.* We firstly show that the given function $u^*$ can be approximated well by some linear combination of multivariate splines, which is denoted by $u_{\text{sp}}$. Note that $N$ is assumed to be sufficiently large. Hence, we may pick $l = \lceil N^{\frac{1}{d}} \rceil = \Theta(N^{\frac{1}{d}}) \in \mathbb{Z}^+$ to be the partition size of the B-splines. Moreover, by picking $k = 4$ and $p = 2$ in Theorem D.7, we have that the linear combination $u_{\text{sp}} := Q_{4,l}u^* = \sum_{\boldsymbol{i} \in I_{4,l}^d} \lambda_{\boldsymbol{i}}(u^*) N_{l,\boldsymbol{i}}^{(4)}$ satisfies:

$$\|u^* - u_{\text{sp}}\|_{H^r(\Omega)} = \|u^* - Q_{4,l}u^*\|_{H^r(\Omega)} \leq C\left(\frac{1}{l}\right)^{s-r} \|u^*\|_{H^s(\Omega)} = CN^{-\frac{s-r}{d}} \|u^*\|_{H^s(\Omega)}.$$

We will then show that the linear combination $u_{\text{sp}} = \sum_{\boldsymbol{i} \in I_{4,l}^d} \lambda_{\boldsymbol{i}}(f) N_{l,\boldsymbol{i}}^{(4)}$ can be implemented by some Deep Neural Network $u_{\text{DNN}} \in \Phi(L, W, S, B)$ with $L = O(1), W = O(N), S = O(N)$ and $B = O(\log N)$. Firstly, note that for $x \geq 0$, both $x$ and $x^2$ can be expressed in terms of the ReLU3 activation function $\eta_3$ with no error:

$$x = -\frac{1}{12}[\eta_3(x+3) - 5\eta_3(x+2) + 7\eta_3(x+1) - 3\eta_3(x) + 6]$$

$$x^2 = -\frac{1}{6}[\eta_3(x+2) - 4\eta_3(x+1) + 3\eta_3(x) - 4]$$

Applying the explicit formula listed in equation D.34 implies that for any $-3 \leq i \leq l-1$, the univariate B-spline function $N_{l,i}^{(4)}(x)$ $(x \in [0,1])$ can be implemented by some ReLU3 Deep Neural Network $v_{\text{DNN}}$ with both scalar input and scalar output. We have that for $v_{\text{DNN}}$, the depth $L_v$ is 2 and the maximum width $W_v$ is upper bounded by 11.

Secondly, for any $x, y \geq 0$, we have that the product operation $x \cdot y$ can be expressed in terms of the ReLU3 activation function $\eta_3$ with no error:

$$
\begin{aligned}
x \cdot y &= \frac{1}{2}[(x+y)^2 - x^2 - y^2] \\
&= -\frac{1}{12}\Big[\eta_3(x+y+2) - 4\eta_3(x+y+1) + 3\eta_3(x+y) \\
&\quad - \eta_3(x+2) + 4\eta_3(x+1) - 3\eta_3(x) - \eta_3(y+2) + 4\eta_3(y+1) - 3\eta_3(y) + 4\Big]
\end{aligned}
$$

In [53], it has been proved that the B-splines are always non-negative, i.e $N_{l,i}^{(4)}(x) \geq 0$, $\forall x \in [0,1]$. Therefore, by multiplying the non-negative univariate B-splines, we can implement any multivariate B-spline $N_{l,\boldsymbol{i}}^{(4)} = \Pi_{j=1}^d N_{l,i_j}^{(4)}(x_j)$ with some ReLU3 Deep Neural Network $p_{\text{DNN}}$. We have that for $p_{\text{DNN}}$, the depth $L_p = \lceil \log_2 d \rceil + 2$ and the maximum width $W_p = \max\{11d, \frac{9}{2}d\}$.

Hence, we can further claim that $u^* = \sum_{\boldsymbol{i} \in I_{4,l}^d} \lambda_{\boldsymbol{i}}(u^*) N_{l,\boldsymbol{i}}^{(4)}$, which is a linear combination of the multivariate B-splines $N_{l,\boldsymbol{i}}^{(4)}$, can be implemented by some ReLU3 Deep Neural Network $u_{\text{DNN}}$. It remains to check that $u_{\text{DNN}} \in \Phi(L, W, S, B)$ with $L = O(1), W = O(N), S = O(N)$ and $B = O(N)$. Note that we can ensure that the hidden layers of $u_{\text{DNN}}$ are of the same dimension $W$ by adding inactive neurons.

For the depth $L$ of $u_{\text{DNN}}$, we have that $L$ is equal to $L_p + 1$, where $L_p$ denotes the depth of the ReLU3 Deep Neural Network $p_{\text{DNN}}$. Thus, we have $L = L_p + 1 = \lceil \log_2 d \rceil + 3$, which implies that $L = O(1)$.

For the width $W$ of $u_{\text{DNN}}$, we have that $W \leq |I_{k,l}^d| W_p$, where $W_p$ denotes the width of the ReLU3 Deep Neural Network $p_{\text{DNN}}$. This implies:

$$
W \leq |I_{k,l}^d| \times 11d = 11d(l+k)^d = 11d(l+4)^d = O(l^d) \Rightarrow W = O(N)
$$

For the sparsity constraint $S$ of $u_{\text{DNN}}$, we have that starting from the third layer, the number of active neurons decreases by a factor of 2 when the index of the layer increases by 1. This yields the following upper bound on $S$:

$$
S \leq 2(W + W + \sum_{j=0}^{L-2} \frac{W}{2^j}) \leq 8W \Rightarrow S = O(W) = O(N)
$$

For the norm constraint $B$ of $u_{\text{DNN}}$, we have the following upper bound on $B$ from equation D.34 and equation D.36:

$$
B = O(\max\{l^{k-1}, \sup_{\boldsymbol{i} \in I_{k,l}^d} \lambda_{\boldsymbol{i}}(u^*)\}) = O(\max\{l^3, l^d\}) = O(N)
$$

Now we have shown that parameters $L, W, S, B$ of the Deep Neural Network $u_{\text{DNN}}$ are of the desired magnitude, which completes our proof. $\square$

## D.4  Final Upper Bound

In this subsection, we provide the proof of upper bounds for PINN and DRM. For both estimator, we first provide a meta-theorem to illustrate the approximation and generalization decomposition with a $O(1/n)$ fast rate generalization bound[2, 66]. Then we use truncated fourier basis estimator and neural network estimator as example to obtain the final rate.

### D.4.1  Deep Ritz Methods

**Theorem D.9** (Meta-theorem for Upper Bounds of Deep Ritz Methods). *Let $u^* \in H^s(\Omega)$ denote the true solution to the PDE model with Dirichlet boundary condition:*

$$
\begin{aligned}
-\Delta u + Vu &= f \text{ on } \Omega, \\
u &= 0 \text{ on } \partial\Omega,
\end{aligned}
\tag{D.38}
$$

where $f \in L^2(\Omega)$ and $V \in L^\infty(\Omega)$ with $0 < V_{\min} \leq V(x) \leq V_{\max} > 0$. For a fixed function space $\boldsymbol{F}(\Omega)$, consider the empirical loss induced by the Deep Ritz Method:

$$\boldsymbol{E}_n(u) = \frac{1}{n} \sum_{j=1}^{n} \left[ |\Omega| \cdot \left( \frac{1}{2} |\nabla u(X_j)|^2 + \frac{1}{2} V(X_j) |u(X_j)|^2 - f(X_j) u(X_j) \right) \right], \qquad \text{(D.39)}$$

where $\{X_j\}_{j=1}^n$ are datapoints uniformly sampled from the domain $\Omega$. Then the Deep Ritz estimator associated with function space $\boldsymbol{F}(\Omega)$ is defined as the minimizer of $\boldsymbol{E}_n(u)$ over the function space $\boldsymbol{F}(\Omega)$:

$$\hat{u}_{DRM} = \min_{u \in \boldsymbol{F}(\Omega)} \boldsymbol{E}_n(u)$$

Moreover, we assume that there exists some constant $C > 0$ such that all function $u$ in the function space $\boldsymbol{F}(\Omega)$, the real solution $u^*$ and $f, V$ satisfy the following two conditions.

- *The gradients and function value are uniformly bounded*

$$\max \left\{ \sup_{u \in \boldsymbol{F}(\Omega)} \|u\|_{L^\infty(\Omega)}, \sup_{u \in \boldsymbol{F}(\Omega)} \|\nabla u\|_{L^\infty(\Omega)}, \|u^*\|_{L^\infty(\Omega)}, \|\nabla u^*\|_{L^\infty(\Omega)}, V_{max}, \|f\|_{L^\infty(\Omega)} \right\} \leq C.$$
$$\text{(D.40)}$$

- *All the functions in the function space $\boldsymbol{F}(\Omega)$ satisfies the boundary condition*

$$u = 0 \text{ on } \partial\Omega.$$

At the the same time, for any $\rho > 0$, we assume the Rademencher complexity of a localized function space $\boldsymbol{S}_\rho(\Omega) := \left\{ h := |\Omega| \cdot \left[ \frac{1}{2} \left( |\nabla u|^2 - |\nabla u^*|^2 \right) + \frac{1}{2} V(|u|^2 - |u^*|^2) - f(u - u^*) \right] \mid \|u - u^*\|_{H_1}^2 \leq \rho \right\}$ can be upper bounded by a sub-root function $\phi = \phi(\rho) : [0, \infty) \to [0, \infty)$, i.e.

$$\phi(4\rho) \leq 2\phi(\rho) \text{ and } R_n(\boldsymbol{S}_\rho(\Omega)) \leq \phi(\rho) \; (\forall \, \rho > 0). \qquad \text{(D.41)}$$

For all constant $t > 0$. We denote $r^*$ to be the solution of the fix point equation of local Rademacher complexity $r = \phi(r)$. There exists a constant $C_p$ such that for probability $1 - C_p \exp(-t)$, we have the following upper bound for the Deep Ritz Estimator

$$\|\hat{u}_{DRM} - u^*\|_{H_1}^2 \lesssim \inf_{u_{\boldsymbol{F}} \in \boldsymbol{F}(\Omega)} \left( \boldsymbol{E}(u_{\boldsymbol{F}}) - \boldsymbol{E}(u^\star) \right) + \max \left\{ r^*, \frac{t}{n} \right\}.$$

*Proof.* To upper bound the excess risk $\Delta \boldsymbol{E}^{(n)} := \boldsymbol{E}(\hat{u}_{DRM}) - \boldsymbol{E}(u^*)$, following[66, 38, 11], we decompose the excess risk into approximation error and generalization error with probability $1 - e^{-t}$:

$$\begin{aligned}
\Delta \boldsymbol{E}^{(n)}(\hat{u}_{DRM}) = \boldsymbol{E}(\hat{u}_{DRM}) - \boldsymbol{E}(u^\star) &= \left[ \boldsymbol{E}(\hat{u}_{DRM}) - \boldsymbol{E}_n(\hat{u}_{DRM}) \right] + \left[ \boldsymbol{E}_n(\hat{u}_{DRM}) - \boldsymbol{E}_n(u_{\boldsymbol{F}}) \right] \\
&\quad + \left[ \boldsymbol{E}_n(u_{\boldsymbol{F}}) - \boldsymbol{E}(u_{\boldsymbol{F}}) \right] + \left[ \boldsymbol{E}(u_{\boldsymbol{F}}) - \boldsymbol{E}(u^\star) \right] \\
&\leq \left[ \boldsymbol{E}(\hat{u}_{DRM}) - \boldsymbol{E}_n(\hat{u}_{DRM}) \right] + \left[ \boldsymbol{E}_n(u_{\boldsymbol{F}}) - \boldsymbol{E}(u_{\boldsymbol{F}}) \right] + \left[ \boldsymbol{E}(u_{\boldsymbol{F}}) - \boldsymbol{E}(u^\star) \right] \\
&\leq \left[ \boldsymbol{E}(\hat{u}_{DRM}) - \boldsymbol{E}(u^*) + \boldsymbol{E}_n(u^*) - \boldsymbol{E}_n(u) \right] \\
&\quad + \frac{3}{2} \left[ \boldsymbol{E}(u_{\boldsymbol{F}}) - \boldsymbol{E}(u^\star) \right] + \frac{t}{2n},
\end{aligned}$$
$$\text{(D.42)}$$

where the expectation is on all sampled data. The inequality of the third line is because the $u$ is the minimizer of the empirical loss $\boldsymbol{E}_n$ in the solution set $\boldsymbol{F}(\Omega)$, so we have $\boldsymbol{E}_n(u) \leq \boldsymbol{E}_n(u_{\boldsymbol{F}})$. The last inequality is based on the Bernstein inequality. The variance of $h = |\Omega| \cdot \left[ \frac{1}{2} \left( |\nabla u|^2 - |\nabla u^*|^2 \right) + \frac{1}{2} V(|u|^2 - |u^*|^2) - f(u - u^*) \right]$ can be bounded by $\left[ \boldsymbol{E}(u_{\boldsymbol{F}}) - \boldsymbol{E}(u^\star) \right]$ due to the strong convexity of the variation objective (D.44). According to the Bernstein inequality, we know with probability $1 - e^{-t}$ we have

$$\boldsymbol{E}_n(u_{\boldsymbol{F}}) - \boldsymbol{E}_n(u^*) - \boldsymbol{E}(u_{\boldsymbol{F}}) + \boldsymbol{E}(u^*) \leq \sqrt{\frac{t \left[ \boldsymbol{E}(u_{\boldsymbol{F}}) - \boldsymbol{E}(u^\star) \right]}{n}} \leq \frac{1}{2} \left[ \boldsymbol{E}(u_{\boldsymbol{F}}) - \boldsymbol{E}(u^\star) \right] + \frac{t}{2n}.$$

Note that D.42 holds for all function lies in the function space $\boldsymbol{F}$. Thus, we can take $u_{\boldsymbol{F}} := \arg\min_{u_0 \in \boldsymbol{F}(\Omega)} \left( \boldsymbol{E}(u_0) - \boldsymbol{E}(u^\star) \right)$ and finally get

$$\Delta \boldsymbol{E}^{(n)} \leq \underbrace{\boldsymbol{E}(\hat{u}_{\mathrm{DRM}}) - \boldsymbol{E}(u^*) + \boldsymbol{E}_n(u^*) - \boldsymbol{E}_n(u)}_{\Delta \boldsymbol{E}_{\mathrm{gen}}} + \frac{3}{2} \underbrace{\inf_{u_{\boldsymbol{F}} \in \boldsymbol{F}(\Omega)} \left( \boldsymbol{E}(u_{\boldsymbol{F}}) - \boldsymbol{E}(u^\star) \right)}_{\Delta \boldsymbol{E}_{\mathrm{app}}} + \frac{t}{2n}.$$

This inequality decompose the excess risk to the generalization error $\Delta \boldsymbol{E}_{\mathrm{gen}} := \mathbb{E}_{x \sim \mu}[\boldsymbol{E}(\hat{u}_{\mathrm{DRM}}) - \boldsymbol{E}(u^*) + \boldsymbol{E}_n(u^*) - \boldsymbol{E}_n(\hat{u}_{\mathrm{DRM}})]$ and the approximation error $\Delta \boldsymbol{E}_{\mathrm{app}} = \inf_{u_{\boldsymbol{F}} \in \boldsymbol{F}(\Omega)} \left( \boldsymbol{E}(u_{\boldsymbol{F}}) - \boldsymbol{E}(u^\star) \right)$.

From the lemmata proved in Section D.3, we already have an estimation of the approximation error's convergence rate. So now we'll focus on providing fast rate upper bounds of the generalization error for the two estimators using the localization technique[2, 66]. To achieve the fast generalization bound, we focus on the following normalized empirical process

$$\tilde{\boldsymbol{S}}_r(\Omega) := \left\{ \tilde{h}(x) := \frac{\mathbb{E}[h] - h(x)}{\mathbb{E}[h] + r} \mid h \in \boldsymbol{S}(\Omega) \right\} \ (r > 0).$$

First, we try to bound the expectation of the normalized empirical process. Applying the Symmetrization Lemma D.1, we can first bound the expectation as

$$\sup_{\tilde{h} \in \tilde{S}_r(\Omega)} \mathbb{E}_{x'} \left[ \frac{1}{n} \sum_{i=1}^n \tilde{h}(x_i') \right] \leq \mathbb{E}_{x'} \left[ \sup_{h \in S(\Omega)} \left| \frac{1}{n} \sum_{i=1}^n \frac{h(x_i') - \mathbb{E}[h]}{\mathbb{E}[h] + r} \right| \right] \leq 2R_n(\hat{\boldsymbol{S}}_r(\Omega)).$$

where the function class $\hat{\boldsymbol{S}}_r(\Omega)$ is defined as:

$$\hat{\boldsymbol{S}}_r(\Omega) := \left\{ \hat{h}(x) := \frac{h(x)}{\mathbb{E}[h] + r} \mid h \in \boldsymbol{S}(\Omega) \right\},$$

where $\boldsymbol{S}(\Omega) = \left\{ h := |\Omega| \cdot \left[ \frac{1}{2} \left( |\nabla u|^2 - |\nabla u^*|^2 \right) + \frac{1}{2} V(|u|^2 - |u^*|^2) - f(u - u^*) \right] \right\}$. Then Applying the Peeling Lemma to any function $h \in \boldsymbol{S}(\Omega)$ helps us upper bound the local Rademacher complexity $R_n(\hat{\boldsymbol{S}}_r(\Omega))$ with the function $\phi$ defined in equation D.41:

$$R_n(\hat{\boldsymbol{S}}_r(\Omega)) = \mathbb{E}_\sigma \left[ \mathbb{E}_x \left[ \sup_{h \in \boldsymbol{S}(\Omega)} \frac{\frac{1}{n} \sum_{i=1}^n \sigma_i h(x_i)}{\mathbb{E}[h] + r} \right] \right] \leq \frac{4\phi(r)}{r}.$$

Combining all inequalities derived above yields:

$$\sup_{\tilde{h} \in \tilde{S}_r(\Omega)} \mathbb{E}_{x'} \left[ \frac{1}{n} \sum_{i=1}^n \tilde{h}(x_i') \right] \leq 2R_n(\hat{\boldsymbol{S}}_r(\Omega)) \leq \frac{8\phi(r)}{r} \ (r > 0). \tag{D.43}$$

Secondly we'll apply the Talagrand concentration inequality, which requires us to verify the condition needed. We will first check that the expectation value $\mathbb{E}[h]$ is always non-negative for any $h \in \boldsymbol{S}(\Omega)$:

$$\mathbb{E}[h] = \frac{1}{|\Omega|} \int_\Omega |\Omega| \cdot (\frac{1}{2} |\nabla u(x)|^2 + \frac{1}{2} V(x)|u(x)|^2 - f(x)u(x))dx$$

$$- \frac{1}{|\Omega|} \int_\Omega |\Omega| \cdot (\frac{1}{2} |\nabla u^\star(x)|^2 + \frac{1}{2} V(x)|u^\star(x)|^2 - f(x)u^\star(x))dx$$

$$= \boldsymbol{E}(u) - \boldsymbol{E}(u^\star) \geq 0 \Rightarrow \mathbb{E}[h] \geq 0.$$

We will proceed to verify that any $\tilde{h} = \frac{\mathbb{E}[h] - h}{\mathbb{E}[h] + r} \in \tilde{\boldsymbol{S}}_r(\Omega)$ is of bounded inf-norm. We need to prove that any $h \in \boldsymbol{S}(\Omega)$ is of bounded inf-norm beforehand. Using boundedness condition listed in

equation D.40 implies:

$$\|h\|_\infty = |\Omega|\|\frac{1}{2}\Big(|\nabla u|^2 - |\nabla u^*|^2\Big) + \frac{1}{2}V(|u|^2 - |u^*|^2) - f(u - u^*)\|_\infty$$

$$\leq \frac{|\Omega|}{2}\Big(\|\nabla u\|_\infty^2 + \|\nabla u^*\|_\infty^2\Big) + \frac{|\Omega|}{2}V_{\max}\Big(\|u\|_\infty^2 + \|u^*\|_\infty^2\Big) + |\Omega|\|f\|_\infty\Big(\|u\|_\infty + \|u^*\|_\infty\Big)$$

$$\leq \frac{|\Omega|}{2} \times 2C^2 + \frac{|\Omega|}{2}V_{\max} \times 2C^2 + 2|\Omega|C^2 = |\Omega|(V_{\max} + 3)C^2$$

By taking $M := |\Omega|(V_{\max} + 3)C^2$, we then have $\|h\|_\infty \leq M$ for all $h \in \boldsymbol{S}(\Omega)$. Note that the denominator can be lower bounded by $|\mathbb{E}[h] + r| \geq r > 0$. Combining these two inequalities help us upper bound the inf-norm $\|\tilde{h}\|_\infty = \sup_{x\in\Omega} |\tilde{h}(x)|$ as follows:

$$\|\tilde{h}\|_\infty = \frac{\|\mathbb{E}[h] - h\|_\infty}{|\mathbb{E}[h] + r|} \leq \frac{2\|h\|_\infty}{r} \leq \frac{2M}{r} =: \beta.$$

We will then check the normalized functions $\frac{\mathbb{E}[h]-h(x)}{\mathbb{E}[h]+r}$ in $\tilde{S}_r(\Omega)$ have bounded second moment, which is satisfied because of the regularity results of the PDE. We aim to show that there exist some constants $\alpha, \alpha' > 0$, such that for any $h \in \boldsymbol{S}(\Omega)$, the following inequality holds:

$$\alpha\mathbb{E}[h^2] \leq \|u - u^*\|_{H^1(\Omega)}^2 \leq \alpha'\mathbb{E}[h]. \tag{D.44}$$

The RHS of the inequality follows from strong convexity of the DRM objective function proved in Theorem D.1:

$$\mathbb{E}[h] = \boldsymbol{E}(u) - \boldsymbol{E}(u^*) \geq \frac{\min\{1, V_{\min}\}}{4}\|u - u^*\|_{H^1(\Omega)}^2$$

The LHS of the inequality follows from boundedness condition listed in equation D.40 and the QM-AM inequality:

$$\mathbb{E}[h^2] = \int_\Omega \left[\frac{1}{2}\Big(|\nabla u|^2 - |\nabla u^*|^2\Big) + \frac{1}{2}V(|u|^2 - |u^*|^2) - f(u - u^*)\right]^2 dx$$

$$\leq \frac{3}{4}\int_\Omega \Big(|\nabla u|^2 - |\nabla u^*|^2\Big)^2 dx + \frac{3}{4}\int_\Omega V^2(|u|^2 - |u^*|^2)^2 dx + 3\int_\Omega f^2(u - u^*)^2 dx$$

$$\leq \frac{3}{4}\int_\Omega \Big||\nabla u| - |\nabla u^*|\Big|^2 (|\nabla u| + |\nabla u^*|)^2 dx + \frac{3}{4}V_{\max}^2\int_\Omega \Big||u| - |u^*|\Big|^2 (|u| + |u^*|)^2 dx$$

$$+ 3C^2\int_\Omega (u - u^*)^2 dx \leq 3C^2\int_\Omega |\nabla u - \nabla u^*|^2 dx + 3C^2(1 + V_{\max}^2)\int_\Omega |u - u^*|^2 dx$$

$$\leq 3C^2(1 + V_{\max}^2)\|u - u^*\|_{H^1(\Omega)}^2$$

By picking $\alpha' = \frac{4}{\min\{1, V_{\min}\}}$ and $\alpha = \frac{1}{3C^2(1+V_{\max}^2)}$, we have finished proving inequality D.44. Then we can can upper bound the expectation $\mathbb{E}[\tilde{h}^2]$ as:

$$\mathbb{E}[\tilde{h}^2] = \frac{\mathbb{E}[(h - \mathbb{E}[h])^2]}{|\mathbb{E}[h] + r|^2} = \frac{\mathbb{E}[h^2] - \mathbb{E}[h]^2}{|\mathbb{E}[h] + r|^2} \leq \frac{\mathbb{E}[h^2]}{|\mathbb{E}[h] + r|^2}.$$

Using the fact that $\mathbb{E}[h] \geq 0$ and inequality D.44, we can lower bound the denominator $|\mathbb{E}[h] + r|^2$ as follows:

$$|\mathbb{E}[h] + r|^2 \geq 2\mathbb{E}[h]r \geq \frac{2r\alpha}{\alpha'}\mathbb{E}[h^2].$$

Therefore, we can deduce that:

$$\mathbb{E}[\tilde{h}^2] \leq \frac{\mathbb{E}[h^2]}{|\mathbb{E}[h] + r|^2} \leq \frac{\mathbb{E}[h^2]}{\frac{2r\alpha}{\alpha'}\mathbb{E}[h^2]} = \frac{\alpha'}{2r\alpha} =: \sigma^2.$$

Hence, any function in the localized class $\tilde{\boldsymbol{S}}_r(\Omega)$ is of bounded second moment.

It's easy to check that for any $\tilde{h} \in \tilde{\boldsymbol{S}}_r(\Omega)$, we have

$$\mathbb{E}[\tilde{h}] = \frac{\mathbb{E}[h] - \mathbb{E}[h]}{\mathbb{E}[h] + r} = 0,$$

*i.e.* any function in the localized class $\tilde{\boldsymbol{S}}_r(\Omega)$ is of zero mean.

Now we have verified that any function $\tilde{h} \in \tilde{\boldsymbol{S}}_r(\Omega)$ satisfies all the required conditions. By taking $\mu$ to be the uniform distribution on the domain $\Omega$ and applying Talagrand's Concentration inequality given in Lemma D.3, we have:

$$\mathbb{P}_x \left[ \sup_{\tilde{h} \in \tilde{\boldsymbol{S}}_r(\Omega)} \frac{1}{n} \sum_{i=1}^{n} \tilde{h}(x_i) \geq 2 \sup_{\tilde{h} \in \tilde{\boldsymbol{S}}_r(\Omega)} \mathbb{E}_{x'} \left[ \frac{1}{n} \sum_{i=1}^{n} \tilde{h}(x_i') \right] + \sqrt{\frac{2t\sigma^2}{n}} + \frac{2t\beta}{n} \right] \leq e^{-t}.$$

By using the upper bound deduced above and plugging in the expressions of $\beta$ and $\sigma$, we can rewrite Talagrand's Concentration Inequality in the following way. With probability at least $1 - e^{-t}$, the inequality below holds:

$$\frac{1}{n} \sum_{i=1}^{n} \tilde{h}(x_i) \leq \sup_{\tilde{h} \in \tilde{\boldsymbol{S}}_r(\Omega)} \frac{1}{n} \sum_{i=1}^{n} \tilde{h}(x_i) \leq 2 \sup_{\tilde{h} \in \tilde{\boldsymbol{S}}_r(\Omega)} \mathbb{E}_{x'} \left[ \frac{1}{n} \sum_{i=1}^{n} \tilde{h}(x_i') \right] + \sqrt{\frac{2t\sigma^2}{n}} + \frac{2t\beta}{n}$$

$$\leq \frac{16\phi(r)}{r} + \sqrt{\frac{t\alpha'}{n\alpha r}} + \frac{4Mt}{nr} =: \psi(r).$$

Let's pick the threshold radius $r_0$ to be:

$$r_0 = \max\{2^{14}r^*, \frac{24Mt}{n}, \frac{36\alpha't}{\alpha n}\}. \tag{D.45}$$

Note that concavity of the function $\phi$ implies that $\phi(r) \leq r$ for any $r \geq r^*$. Combining this with the first inequality listed in D.41 yields:

$$\frac{16\phi(r)}{r} \leq \frac{2^{11}\phi(\frac{r_0}{2^{14}})}{2^{14}\frac{r_0}{2^{14}}} = \frac{1}{8} \times \frac{\phi(\frac{r_0}{2^{14}})}{\frac{r_0}{2^{14}}} \leq \frac{1}{8}.$$

On the other hand, applying equation D.45 yields:

$$\sqrt{\frac{\alpha't}{n\alpha r_0}} \leq \sqrt{\frac{\alpha't}{n\alpha} \frac{\alpha n}{36\alpha't}} = \frac{1}{6},$$

$$\frac{4Mt}{nr_0} \leq \frac{4Mt}{n} \times \frac{n}{24Mt} = \frac{1}{6}.$$

Summing the three inequalities above implies:

$$\psi(r_0) = \frac{16\phi(r_0)}{r_0} + \sqrt{\frac{t\alpha'}{n\alpha r_0}} + \frac{4Mt}{nr_0} \leq \frac{1}{8} + \frac{1}{6} + \frac{1}{6} < \frac{1}{2}.$$

By picking $r = r_0$, we can further deduce that for any function $u \in \boldsymbol{F}(\Omega)$, the following inequality holds with probability $1 - e^{-t}$:

$$\frac{\boldsymbol{E}(u) - \boldsymbol{E}(u^*) - \boldsymbol{E}_n(u) + \boldsymbol{E}_n(u^*)}{\boldsymbol{E}(u) - \boldsymbol{E}(u^*) + r_0} = \frac{1}{n} \sum_{i=1}^{n} \tilde{h}(x_i) \leq \psi(r_0) < \frac{1}{2}.$$

Multiplying the denominator on both sides indicates:

$$\Delta \boldsymbol{E}_{\text{gen}} = \boldsymbol{E}(u) - \boldsymbol{E}(u^*) - \boldsymbol{E}_n(u) + \boldsymbol{E}_n(u^*) \leq \frac{1}{2} \left[ \boldsymbol{E}(u) - \boldsymbol{E}(u^*) \right] + \frac{1}{2}r_0 = \frac{1}{2}\Delta \boldsymbol{E}^{(n)} + \frac{1}{2}r_0.$$

Substituting the upper bound above into the decomposition $\Delta \boldsymbol{E}^{(n)} \leq \Delta E_{\text{gen}} + \frac{3}{2}\Delta E_{\text{app}} + \frac{t}{2n}$ yields that with probability $1 - e^{-t}$, we have:

$$\Delta \boldsymbol{E}^{(n)} \leq \Delta E_{\text{gen}} + \frac{3}{2}\Delta E_{\text{app}} + \frac{t}{2n} \leq \frac{1}{2}\Delta \boldsymbol{E}^{(n)} + \frac{1}{2}r_0 + \frac{3}{2}\Delta E_{\text{app}} + \frac{t}{2n}.$$

Simplifying the inequality above yields that with probability $1 - e^{-t}$, we have:

$$\Delta \boldsymbol{E}^{(n)} \leq r_0 + 3\Delta \boldsymbol{E}_{\mathrm{app}} + \frac{t}{n} = 3 \inf_{u_{\boldsymbol{F}} \in \boldsymbol{F}(\Omega)} \left( \boldsymbol{E}(u_{\boldsymbol{F}}) - \boldsymbol{E}(u^{\star}) \right) + \max\{2^{14}r^*, 24M\frac{t}{n}, \frac{36\alpha'}{\alpha}\frac{t}{n}\} + \frac{t}{n}$$

$$\lesssim \inf_{u_{\boldsymbol{F}} \in \boldsymbol{F}(\Omega)} \left( \boldsymbol{E}(u_{\boldsymbol{F}}) - \boldsymbol{E}(u^{\star}) \right) + \max\left\{ r^*, \frac{t}{n} \right\}$$

Moreover, using strong convexity of the DRM objective function proved in Theorem D.1 implies:

$$\Delta \boldsymbol{E}^{(n)} = \boldsymbol{E}(\hat{u}_{\mathrm{DRM}}) - \boldsymbol{E}(u^*) \geq \frac{\min\{1, V_{\min}\}}{4} \|\hat{u}_{\mathrm{DRM}} - u^*\|_{H^1(\Omega)}^2$$

Combining the two bounds above yields that with probability $1 - e^{-t}$, we have:

$$\|\hat{u}_{\mathrm{DRM}} - u^*\|_{H^1(\Omega)}^2 \lesssim \inf_{u_{\boldsymbol{F}} \in \boldsymbol{F}(\Omega)} \left( \boldsymbol{E}(u_{\boldsymbol{F}}) - \boldsymbol{E}(u^{\star}) \right) + \max\left\{ r^*, \frac{t}{n} \right\}$$

$\square$

**Deep Neural Network Estimator.** For any $N \in \mathbb{Z}^+$, there exists some Deep Neural Network in $\Phi(L, W, S, B)$ with $L = O(1)$, $W = O(N)$, $S = O(N)$, $B = O(1)$, such that the approximation error $\Delta \boldsymbol{E}_{\mathrm{app}} = O(N^{-\frac{2(s-1)}{d}})$ and generalization error $\Delta \boldsymbol{E}_{\mathrm{gen}} = O(\frac{N \log N}{n})$. With optimal selection $N = n^{\frac{d}{d+2s-2}}$ to balance the bias and variance, we can achieve $n^{-\frac{2s-2}{d+2s-2}} \log n$ convergence rate for DRM estimator.

**Theorem D.10.** *(Final Upper Bound of DRM with Deep Neural Network Estimator) Consider the sparse Deep Neural Network function space $\Phi(L, W, S, B)$ with parameters $L = O(1)$, $W = O(n^{\frac{d}{d+2s-2}})$, $S = O(n^{\frac{d}{d+2s-2}})$, $B = O(1)$, then the Deep ritz estimator $\hat{u}_{DRM}^{DNN} = \min_{u \in \Phi(L, W, S, B)} \boldsymbol{E}_n(u)$ satisfies the following upper bound*

$$\|\hat{u}_{DRM}^{DNN} - u^*\|_{H_1}^2 \lesssim n^{-\frac{2s-2}{d+2s-2}} \log n.$$

*Proof.* On the one hand, by taking $s = 1$ and $p = 2$ in Theorem D.8 proved above, we have that there exists some Deep Neural Network $u_{\mathrm{DNN}} \in \Phi(L, W, S, B)$ with $L = O(1), W = O(N), S = O(N), B = O(1)$, such that.

$$\|u_{\mathrm{DNN}} - u^*\|_{H^1(\Omega)}^2 \leq N^{-\frac{2s-2}{d}} \|u^*\|_{H^s(\Omega)}.$$

Applying strong convexity of the DRM objective function proved in Section 2.1 further implies:

$$\Delta \boldsymbol{E}_{\mathrm{app}} \lesssim \|u_{\mathrm{DNN}} - u^*\|_{H^1(\Omega)}^2 \leq N^{-\frac{2s-2}{d}}.$$

On the other hand, from Lemma D.14 proved above, we know that the function $\phi(\rho)$ that upper bounds the local Rademacher complexity of the Deep Neural Networks $u_{\mathrm{DNN}}$ is dominated by the term $\sqrt{\frac{\rho 3^L S}{n} \log(W(B \vee 1)n)}$. By plugging in the magnitudes of $L, W, S, B$, we can determine the thresholding localization radius $\hat{r}$:

$$\sqrt{\frac{\rho 3^L S}{n} \log(W(B \vee 1)n)} \simeq \sqrt{\frac{\rho N}{n}(\log N + \log n)} \simeq \rho \Rightarrow \hat{r} \simeq \frac{N(\log N + \log n)}{n}.$$

Combining the two bounds above gives us:

$$\mathbb{E}_{x \sim \mu}[\Delta \boldsymbol{E}_n] \lesssim \Delta \boldsymbol{E}_{\mathrm{app}} + \hat{r} \lesssim N^{-\frac{2(s-1)}{d}} + \frac{N(\log N + \log n)}{n}.$$

By equating the two terms above, we can solve for the optimal $N$ that yields the desired bound:

$$N^{-\frac{2(s-1)}{d}} \simeq \frac{N}{n} \Rightarrow N \simeq n^{\frac{d}{d+2s-2}}.$$

Plugging in the optimal $N$ gives us the magnitudes of the four parameters $L = O(1)$, $W = O(n^{\frac{2s-2}{d+2s-2}})$, $S = O(n^{\frac{2s-2}{d+2s-2}})$, $B = O(1)$, as well as the final rate:

$$\mathbb{E}_{x \sim \mu}[\Delta \boldsymbol{E}_n] \lesssim N^{-\frac{2(s-1)}{d}} + \frac{N \log N}{n} \lesssim n^{-\frac{2(s-1)}{d+2(s-1)}} \log n.$$

$\square$

**Truncated Fourier Series Estimator.** For any $\xi \in \mathbb{Z}^+$, there exists some Truncated Fourier Series in $\boldsymbol{F}_\xi(\Omega)$ with approximation error $\Delta \boldsymbol{E}_{\text{app}} = O(\xi^{-2(s-1)})$ and generalization error $\Delta \boldsymbol{E}_{\text{gen}} = O(\frac{\xi^d}{n})$

**Theorem D.11.** *(Final Upper Bound of DRM with Truncated Fourier Series Estimator) Consider the Deep Ritz objective with a plug in Fourier Series estimator $\hat{u}_{DRM}^{Fourier} = \min_{u \in \boldsymbol{F}_\xi(\Omega)} \boldsymbol{E}_n(u)$ with $\xi = \Theta(n^{\frac{1}{d+2s-2}})$, then we have*

$$\|\hat{u}_{DRM}^{Fourier} - u^*\|_{H_1}^2 \lesssim n^{-\frac{2s-2}{d+2s-2}}$$

*Proof.* On the one hand, from Lemma D.5 and Lemma D.6 proved above, we know that the function $\phi(\rho)$ that upper bounds the local Rademacher complexity is dominated by the term $\sqrt{\frac{\rho}{n}} \xi^{\frac{d}{2}}$ for Truncated Fourier Series in $\boldsymbol{F}_\xi(\Omega)$. Thus, the thresholding localization radius $\hat{r}$ can be determined as follows:

$$\sqrt{\frac{\rho}{n}} \xi^{\frac{d}{2}} \simeq \rho \Rightarrow \hat{r} \simeq \frac{\xi^d}{n},$$

On the other hand, by taking $\alpha = s$ and $\beta = 1$ in Lemma D.16 and applying strong convexity of the DRM objective function proved in Section 2.1, we can upper bound the approximation error $\Delta \boldsymbol{E}_{\text{app}}$ as below:

$$\Delta \boldsymbol{E}_{\text{app}} \lesssim \xi^{-2(s-1)},$$

By equating the two terms above, we can solve for $\xi$ that yields the desired bound:

$$\frac{\xi^d}{n} \simeq \xi^{-2(s-1)} \Rightarrow \xi \simeq n^{\frac{1}{d+2s-2}},$$

Plugging in the expression of $\xi$ gives the final upper bound:

$$\mathbb{E}_{x \sim \mu}[\Delta \boldsymbol{E}_n] \lesssim \hat{r} + \Delta \boldsymbol{E}_{\text{app}} \lesssim \frac{\xi^d}{n} + \xi^{-2(s-1)} \simeq n^{-\frac{2s-2}{d+2s-2}}.$$

$\square$

### D.4.2 Physics Informed Neural Network

**Theorem D.12** (Meta-theorem for Upper Bounds of Physics Informed Neural Network). *Let $u^* \in H^s(\Omega)$ denote the true solution to the PDE model with Dirichlet boundary condition:*

$$\begin{aligned} -\Delta u + V u &= f \text{ on } \Omega, \\ u &= 0 \text{ on } \partial\Omega, \end{aligned} \tag{D.46}$$

*where $f \in L^2(\Omega)$ and $V \in L^\infty(\Omega)$ with $0 < V_{\min} \le V(x) \le V_{\max} > 0$. For a fixed function space $\boldsymbol{F}(\Omega)$, consider the empirical loss induced by the Physics Informed Neural Network:*

$$\boldsymbol{E}_n(u) = \frac{1}{n} \sum_{j=1}^n \left[ |\Omega| \cdot \Big( \Delta u(X_j) - V(X_j)u(X_j) + f(X_j) \Big)^2 \right], \tag{D.47}$$

*where $\{X_j\}_{j=1}^n$ are datapoints uniformly sampled from the domain $\Omega$. Then the Physics Informed Neural Network estimator associated with function space $\boldsymbol{F}(\Omega)$ is defined as the minimizer of $\boldsymbol{E}_n(u)$ over the function space $\boldsymbol{F}(\Omega)$:*

$$\hat{u}_{PINN} = \min_{u \in \boldsymbol{F}(\Omega)} \boldsymbol{E}_n(u)$$

*Moreover, we assume that there exists some constant $C > 0$ such that all function $u$ in the function space $\boldsymbol{F}(\Omega)$, the real solution $u^*$ and $f, V$ satisfy the following two conditions.*

- *The gradients and function value are uniformly bounded*

$$\max \Big\{ \sup_{u \in \boldsymbol{F}(\Omega)} \|u\|_{L^\infty(\Omega)}, \sup_{u \in \boldsymbol{F}(\Omega)} \|\nabla u\|_{L^\infty(\Omega)}, \sup_{u \in \boldsymbol{F}(\Omega)} \|\Delta u\|_{L^\infty(\Omega)}, $$
$$\|u^*\|_{L^\infty(\Omega)}, \|\nabla u^*\|_{L^\infty(\Omega)}, \|\Delta u^*\|_{L^\infty(\Omega)}, V_{max}, \|f\|_{L^\infty(\Omega)} \Big\} \le C. \tag{D.48}$$

- *All the functions in the function space $\boldsymbol{F}(\Omega)$ satisfies the boundary condition*

$$u = 0 \text{ on } \partial\Omega.$$

*At the the same time, for any $\rho > 0$, we assume the Rademencher complexity of a localized function space $\boldsymbol{T}_\rho(\Omega) := \left\{ h := |\Omega| \cdot \left[ (\Delta u - Vu + f)^2 - (\Delta u^* - Vu^* + f)^2 \right] \mid \|u - u^*\|_{H_2}^2 \le \rho \right\}$ can be upper bounded by a sub-root function $\phi = \phi(\rho) : [0, \infty) \to [0, \infty)$, i.e.*

$$\phi(4\rho) \le 2\phi(\rho) \text{ and } R_n(\boldsymbol{T}_\rho(\Omega)) \le \phi(\rho) \ (\forall \, \rho > 0). \tag{D.49}$$

*For all constant $t > 0$. We denote $r^*$ to be the solution of the fix point equation of local Rademacher complexity $r = \phi(r)$. There exists a constant $C_p$ such that for probability $1 - C_p \exp(-t)$, we have the following upper bound for the Physics Informed Neural Network Estimator*

$$\|\hat{u}_{PINN} - u^*\|_{H_2}^2 \lesssim \inf_{u_{\boldsymbol{F}} \in \boldsymbol{F}(\Omega)} \left( \boldsymbol{E}(u_{\boldsymbol{F}}) - \boldsymbol{E}(u^\star) \right) + \max\left\{ r^*, \frac{t}{n} \right\}.$$

*Proof.* To upper bound the excess risk $\Delta\boldsymbol{E}^{(n)}$, following[66, 38, 11], we decompose the excess risk into approximation error and generalization error with probability $1 - e^{-t}$:

$$
\begin{aligned}
\Delta\boldsymbol{E}^{(n)}(\hat{u}_{PINN}) = \boldsymbol{E}(\hat{u}_{PINN}) - \boldsymbol{E}(u^\star) &= \left[ \boldsymbol{E}(\hat{u}_{PINN}) - \boldsymbol{E}_n(\hat{u}_{PINN}) \right] + \left[ \boldsymbol{E}_n(\hat{u}_{PINN}) - \boldsymbol{E}_n(u_{\boldsymbol{F}}) \right] \\
&\quad + \left[ \boldsymbol{E}_n(u_{\boldsymbol{F}}) - \boldsymbol{E}(u_{\boldsymbol{F}}) \right] + \left[ \boldsymbol{E}(u_{\boldsymbol{F}}) - \boldsymbol{E}(u^\star) \right] \\
&\le \left[ \boldsymbol{E}(\hat{u}_{PINN}) - \boldsymbol{E}_n(\hat{u}_{PINN}) \right] + \left[ \boldsymbol{E}_n(u_{\boldsymbol{F}}) - \boldsymbol{E}(u_{\boldsymbol{F}}) \right] + \left[ \boldsymbol{E}(u_{\boldsymbol{F}}) - \boldsymbol{E}(u^\star) \right] \\
&\le \left[ \boldsymbol{E}(\hat{\boldsymbol{E}}(\hat{u}_{PINN}) - \boldsymbol{E}(u^*) + \boldsymbol{E}_n(u^*) - \boldsymbol{E}_n(\hat{u}_{PINN}) \right] \\
&\quad + \frac{3}{2}\left[ \boldsymbol{E}(u_{\boldsymbol{F}}) - \boldsymbol{E}(u^\star) \right] + \frac{t}{2n},
\end{aligned}
\tag{D.50}
$$

where the expectation is on all sampled data. The inequality of the third line is because the $u$ is the minimizer of the empirical loss $\boldsymbol{E}_n$ in the solution set $\boldsymbol{F}(\Omega)$, so we have $\boldsymbol{E}_n(u) \le \boldsymbol{E}_n(u_{\boldsymbol{F}})$. The last inequality is based on the Bernstein inequality. The variance of $h = |\Omega| \cdot \left[ (\Delta u - Vu + f)^2 - (\Delta u^* - Vu^* + f)^2 \right]$ can be bounded by $\left[ \boldsymbol{E}(u_{\boldsymbol{F}}) - \boldsymbol{E}(u^\star) \right]$ due to the strong convexity of the variation objective (D.52). According to the Brenstein inequality, we know with probability $1 - e^{-t}$ we have

$$\boldsymbol{E}_n(u_{\boldsymbol{F}}) - \boldsymbol{E}_n(u^*) - \boldsymbol{E}(u_{\boldsymbol{F}}) + \boldsymbol{E}(u^*) \le \sqrt{\frac{t\left[ \boldsymbol{E}(u_{\boldsymbol{F}}) - \boldsymbol{E}(u^\star) \right]}{n}} \le \frac{1}{2}\left[ \boldsymbol{E}(u_{\boldsymbol{F}}) - \boldsymbol{E}(u^\star) \right] + \frac{t}{2n}.$$

Note that E.5 holds for all function lies in the function space $\boldsymbol{F}$. Thus, we can take $u_{\boldsymbol{F}} := \arg\min_{u_0 \in \boldsymbol{F}(\Omega)} \left( \boldsymbol{E}(u_0) - \boldsymbol{E}(u^\star) \right)$ and finally get

$$\Delta\boldsymbol{E}^{(n)} \le \underbrace{\boldsymbol{E}(\hat{u}_{PINN}) - \boldsymbol{E}(u^*) + \boldsymbol{E}_n(u^*) - \boldsymbol{E}_n(\hat{u}_{PINN})}_{\Delta\boldsymbol{E}_{\text{gen}}} + \frac{3}{2}\underbrace{\inf_{u_{\boldsymbol{F}} \in \boldsymbol{F}(\Omega)} \left( \boldsymbol{E}(u_{\boldsymbol{F}}) - \boldsymbol{E}(u^\star) \right)}_{\Delta\boldsymbol{E}_{\text{app}}} + \frac{t}{2n}.$$

This inequality decompose the excess risk to the generalization error $\Delta\boldsymbol{E}_{\text{gen}} := \boldsymbol{E}(\hat{u}_{PINN}) - \boldsymbol{E}(u^*) + \boldsymbol{E}_n(u^*) - \boldsymbol{E}_n(\hat{u}_{PINN})$ and the approximation error $\Delta\boldsymbol{E}_{\text{app}} = \inf_{u_{\boldsymbol{F}} \in \boldsymbol{F}(\Omega)} \left( \boldsymbol{E}(u_{\boldsymbol{F}}) - \boldsymbol{E}(u^\star) \right)$.

From the lemmata proved in Section D.3, we already have an estimation of the approximation error's convergence rate. So now we'll focus on providing fast rate upper bounds of the generalization error for the two estimators using the localization techinque[2, 66]. To achieve the fast generalization bound, we focus on the following normalized empirical process

$$\tilde{\boldsymbol{T}}_r(\Omega) := \left\{ \tilde{h}(x) := \frac{\mathbb{E}[h] - h(x)}{\mathbb{E}[h] + r} \mid h \in \boldsymbol{T}(\Omega) \right\} (r > 0).$$

First, we try to bound the expectation of the normalized empirical process. Applying the Symmetrization Lemma D.1, we can first bound the expectation as

$$\sup_{\tilde{h} \in \tilde{T}_r(\Omega)} \mathbb{E}_{x'}\left[\frac{1}{n}\sum_{i=1}^{n}\tilde{h}(x_i')\right] \leq \mathbb{E}_{x'}\left[\sup_{h \in T(\Omega)}\left|\frac{1}{n}\sum_{i=1}^{n}\frac{h(x_i') - \mathbb{E}[h]}{\mathbb{E}[h] + r}\right|\right] \leq 2R_n(\hat{T}_r(\Omega)).$$

where the function class $\hat{S}_r(\Omega)$ is defined as:

$$\hat{T}_r(\Omega) := \left\{\hat{h}(x) := \frac{h(x)}{\mathbb{E}[h] + r} \mid h \in T(\Omega)\right\},$$

where $T(\Omega) = \left\{h := |\Omega| \cdot \left[(\Delta u - Vu + f)^2 - (\Delta u^* - Vu^* + f)^2\right]\right\}$. Then Applying the Peeling Lemma to any function $h \in T(\Omega)$ helps us upper bound the local Rademacher complexity $R_n(\hat{T}_r(\Omega))$ with the function $\phi$ defined in equation D.49:

$$R_n(\hat{T}_r(\Omega)) = \mathbb{E}_\sigma\left[\mathbb{E}_x\left[\sup_{h \in T(\Omega)}\frac{\frac{1}{n}\sum_{i=1}^{n}\sigma_i h(x_i)}{\mathbb{E}[h] + r}\right]\right] \leq \frac{4\phi(r)}{r}.$$

Combining all inequalities derived above yields:

$$\sup_{\tilde{h} \in \tilde{T}_r(\Omega)} \mathbb{E}_{x'}\left[\frac{1}{n}\sum_{i=1}^{n}\tilde{h}(x_i')\right] \leq 2R_n(\hat{T}_r(\Omega)) \leq \frac{8\phi(r)}{r} \ (r > 0). \tag{D.51}$$

Secondly we'll apply the Talagrand concentration inequality, which requires us to verify the condition needed. We will first check that the expectation value $\mathbb{E}[h]$ is always non-negative for any $h \in S(\Omega)$:

$$\begin{aligned}
\mathbb{E}[h] &= \frac{1}{|\Omega|}\int_\Omega |\Omega| \cdot \left(\frac{1}{2}|\nabla u(x)|^2 + \frac{1}{2}V(x)|u(x)|^2 - f(x)u(x)\right)dx \\
&\quad - \frac{1}{|\Omega|}\int_\Omega |\Omega| \cdot \left(\frac{1}{2}|\nabla u^\star(x)|^2 + \frac{1}{2}V(x)|u^\star(x)|^2 - f(x)u^\star(x)\right)dx \\
&= \boldsymbol{E}(u) - \boldsymbol{E}(u^\star) \geq 0 \Rightarrow \mathbb{E}[h] \geq 0.
\end{aligned}$$

We will proceed to verify that any $\tilde{h} = \frac{\mathbb{E}[h]-h}{\mathbb{E}[h]+r} \in \tilde{T}_r(\Omega)$ is of bounded inf-norm. We need to prove that any $h \in T(\Omega)$ is of bounded inf-norm beforehand. Using boundedness condition listed in equation D.48 implies:

$$\begin{aligned}
\|h\|_\infty &= |\Omega| \cdot \|(\Delta u - Vu + f)^2 - (\Delta u^* - Vu^* + f)^2\|_\infty = |\Omega| \cdot \|(\Delta u - Vu + f)^2\|_\infty \\
&\leq |\Omega| \cdot (\|\Delta u\|_\infty + V_{\max}\|u\|_\infty + \|f\|_\infty)^2 \leq |\Omega|(V_{\max} + 2)^2 C^2
\end{aligned}$$

By taking $M := |\Omega|(V_{\max} + 2)^2 C^2$, we then have $\|h\|_\infty \leq M$ for all $h \in T(\Omega)$. Note that the denominator can be lower bounded by $|\mathbb{E}[h] + r| \geq r > 0$. Combining these two inequalities help us upper bound the inf-norm $\|\tilde{h}\|_\infty = \sup_{x \in \Omega}|\tilde{h}(x)|$ as follows:

$$\|\tilde{h}\|_\infty = \frac{\|\mathbb{E}[h] - h\|_\infty}{|\mathbb{E}[h] + r|} \leq \frac{2\|h\|_\infty}{r} \leq \frac{2M}{r} =: \beta.$$

We will then check the normalized functions $\frac{\mathbb{E}[h]-h(x)}{\mathbb{E}[h]+r}$ in $\tilde{T}_r(\Omega)$ have bounded second moment, which is satisfied because of the regularity results of the PDE. We aim to show that there exist some constants $\alpha, \alpha' > 0$, such that for any $h \in T(\Omega)$, the following inequality holds:

$$\alpha\mathbb{E}[h^2] \leq \|u - u^*\|_{H_2(\Omega)}^2 \leq \alpha'\mathbb{E}[h]. \tag{D.52}$$

The RHS of the inequality follows from strong convexity of the PINN objective function proved in Theorem D.2:

$$\mathbb{E}[h] = \boldsymbol{E}(u) - \boldsymbol{E}(u^*) \geq \min\{1, V_{\min}\}\|u - u^*\|_{H_2(\Omega)}^2$$

The LHS of the inequality follows from boundedness condition listed in equation D.48 and the QM-AM inequality:

$$\mathbb{E}[h^2] = \int_\Omega \left[ (\Delta u - Vu + f)^2 - (\Delta u^* - Vu^* + f)^2 \right]^2 dx = \int_\Omega (\Delta u - Vu + f)^4 dx$$

$$\leq M^2 \int_\Omega (\Delta u - Vu - \Delta u^* + Vu^*)^2 dx \leq 2M^2 \int_\Omega [(\Delta u - \Delta u^*)^2 + V^2(u - u^*)^2] dx$$

$$\leq 2M^2 \max\{1, V_{\max}^2\} \|u - u^*\|_{H_2(\Omega)}^2$$

By picking $\alpha' = \min\{1, V_{\min}\}$ and $\alpha = \frac{1}{2M^2 \max\{1, V_{\max}^2\}}$, we have finished proving inequality D.52.

Then we can can upper bound the expectation $\mathbb{E}[\tilde{h}^2]$ as:

$$\mathbb{E}[\tilde{h}^2] = \frac{\mathbb{E}[(h - \mathbb{E}[h])^2]}{|\mathbb{E}[h] + r|^2} = \frac{\mathbb{E}[h^2] - \mathbb{E}[h]^2}{|\mathbb{E}[h] + r|^2} \leq \frac{\mathbb{E}[h^2]}{|\mathbb{E}[h] + r|^2}.$$

Using the fact that $\mathbb{E}[h] \geq 0$ and inequality D.52, we can lower bound the denominator $|\mathbb{E}[h] + r|^2$ as follows:

$$|\mathbb{E}[h] + r|^2 \geq 2\mathbb{E}[h]r \geq \frac{2r\alpha}{\alpha'} \mathbb{E}[h^2].$$

Therefore, we can deduce that:

$$\mathbb{E}[\tilde{h}^2] \leq \frac{\mathbb{E}[h^2]}{|\mathbb{E}[h] + r|^2} \leq \frac{\mathbb{E}[h^2]}{\frac{2r\alpha}{\alpha'} \mathbb{E}[h^2]} = \frac{\alpha'}{2r\alpha} =: \sigma^2.$$

Hence, any function in the localized class $\tilde{T}_r(\Omega)$ is of bounded second moment.

It's easy to check that for any $\tilde{h} \in \tilde{T}_r(\Omega)$, we have

$$\mathbb{E}[\tilde{h}] = \frac{\mathbb{E}[h] - \mathbb{E}[h]}{\mathbb{E}[h] + r} = 0,$$

*i.e.* any function in the localized class $\tilde{S}_r(\Omega)$ is of zero mean.

Now we have verified that any function $\tilde{h} \in \tilde{S}_r(\Omega)$ satisfies all the required conditions. By taking $\mu$ to be the uniform distribution on the domain $\Omega$ and applying Talagrand's Concentration inequality given in Lemma D.3, we have:

$$\mathbb{P}_x \left[ \sup_{\tilde{h} \in \tilde{T}_r(\Omega)} \frac{1}{n} \sum_{i=1}^n \tilde{h}(x_i) \geq 2 \sup_{\tilde{h} \in \tilde{T}_r(\Omega)} \mathbb{E}_{x'} \left[ \frac{1}{n} \sum_{i=1}^n \tilde{h}(x_i') \right] + \sqrt{\frac{2t\sigma^2}{n}} + \frac{2t\beta}{n} \right] \leq e^{-t}.$$

By using the upper bound deduced above and plugging in the expressions of $\beta$ and $\sigma$, we can rewrite Talagrand's Concentration Inequality in the following way. With probability at least $1 - e^{-t}$, the inequality below holds:

$$\frac{1}{n} \sum_{i=1}^n \tilde{h}(x_i) \leq \sup_{\tilde{h} \in \tilde{S}_r(\Omega)} \frac{1}{n} \sum_{i=1}^n \tilde{h}(x_i) \leq 2 \sup_{\tilde{h} \in \tilde{S}_r(\Omega)} \mathbb{E}_{x'} \left[ \frac{1}{n} \sum_{i=1}^n \tilde{h}(x_i') \right] + \sqrt{\frac{2t\sigma^2}{n}} + \frac{2t\beta}{n}$$

$$\leq \frac{16\phi(r)}{r} + \sqrt{\frac{t\alpha'}{n\alpha r}} + \frac{4Mt}{nr} =: \psi(r)$$

Let's pick the threshold radius $r_0$ to be:

$$r_0 = \max\{2^{14} r^*, \frac{24Mt}{n}, \frac{36\alpha't}{\alpha n}\}. \tag{D.53}$$

Note that concavity of the function $\phi$ implies that $\phi(r) \leq r$ for any $r \geq r^*$. Combining this with the first inequality listed in D.49 yields:

$$\frac{16\phi(r)}{r} \leq \frac{2^{11}\phi(\frac{r_0}{2^{14}})}{2^{14}\frac{r_0}{2^{14}}} = \frac{1}{8} \times \frac{\phi(\frac{r_0}{2^{14}})}{\frac{r_0}{2^{14}}} \leq \frac{1}{8}.$$

On the other hand, applying equation D.53 yields:

$$\sqrt{\frac{\alpha' t}{n\alpha r_0}} \leq \sqrt{\frac{\alpha' t}{n\alpha}\frac{\alpha n}{36\alpha' t}} = \frac{1}{6},$$

$$\frac{4Mt}{nr_0} \leq \frac{4Mt}{n} \times \frac{n}{24Mt} = \frac{1}{6}.$$

Summing the three inequalities above implies:

$$\psi(r_0) = \frac{16\phi(r_0)}{r_0} + \sqrt{\frac{t\alpha'}{n\alpha r_0}} + \frac{4Mt}{nr_0} \leq \frac{1}{8} + \frac{1}{6} + \frac{1}{6} < \frac{1}{2}.$$

By picking $r = r_0$, we can further deduce that for any function $u \in \boldsymbol{F}(\Omega)$, the following inequality holds with probability $1 - e^{-t}$:

$$\frac{\boldsymbol{E}(u) - \boldsymbol{E}(u^*) - \boldsymbol{E}_n(u) + \boldsymbol{E}_n(u^*)}{\boldsymbol{E}(u) - \boldsymbol{E}(u^*) + r_0} = \frac{1}{n}\sum_{i=1}^{n}\tilde{h}(x_i) \leq \psi(r_0) < \frac{1}{2}.$$

Multiplying the denominator on both sides indicates:

$$\Delta\boldsymbol{E}_{\text{gen}} = \boldsymbol{E}(u) - \boldsymbol{E}(u^*) - \boldsymbol{E}_n(u) + \boldsymbol{E}_n(u^*) \leq \frac{1}{2}\Big[\boldsymbol{E}(u) - \boldsymbol{E}(u^*)\Big] + \frac{1}{2}r_0 = \frac{1}{2}\Delta\boldsymbol{E}^{(n)} + \frac{1}{2}r_0.$$

Substituting the upper bound above into the decomposition $\Delta\boldsymbol{E}^{(n)} \leq \Delta E_{\text{gen}} + \frac{3}{2}\Delta E_{\text{app}} + \frac{t}{2n}$ yields that with probability $1 - e^{-t}$, we have:

$$\Delta\boldsymbol{E}^{(n)} \leq \Delta\boldsymbol{E}_{\text{gen}} + \frac{3}{2}\Delta\boldsymbol{E}_{\text{app}} + \frac{t}{2n} \leq \frac{1}{2}\Delta\boldsymbol{E}^{(n)} + \frac{1}{2}r_0 + \frac{3}{2}\Delta\boldsymbol{E}_{\text{app}} + \frac{t}{2n}.$$

Simplifying the inequality above yields that with probability $1 - e^{-t}$, we have:

$$\Delta\boldsymbol{E}^{(n)} \leq r_0 + 3\Delta\boldsymbol{E}_{\text{app}} + \frac{t}{n} = 3\inf_{u_{\boldsymbol{F}}\in\boldsymbol{F}(\Omega)}\Big(\boldsymbol{E}(u_{\boldsymbol{F}}) - \boldsymbol{E}(u^\star)\Big) + \max\{2^{14}r^*, 24M\frac{t}{n}, \frac{36\alpha'}{\alpha}\frac{t}{n}\} + \frac{t}{n}$$

$$\lesssim \inf_{u_{\boldsymbol{F}}\in\boldsymbol{F}(\Omega)}\Big(\boldsymbol{E}(u_{\boldsymbol{F}}) - \boldsymbol{E}(u^\star)\Big) + \max\Big\{r^*, \frac{t}{n}\Big\}$$

Moreover, using strong convexity of the PINN objective function proved in Theorem D.1 implies:

$$\Delta\boldsymbol{E}^{(n)} = \boldsymbol{E}(\hat{u}_{\text{PINN}}) - \boldsymbol{E}(u^*) \geq \min\{1, V_{\min}\}\|\hat{u}_{\text{PINN}} - u^*\|_{H^1(\Omega)}^2$$

Combining the two bounds above yields that with probability $1 - e^{-t}$, we have:

$$\|\hat{u}_{\text{PINN}} - u^*\|_{H^1(\Omega)}^2 \lesssim \inf_{u_{\boldsymbol{F}}\in\boldsymbol{F}(\Omega)}\Big(\boldsymbol{E}(u_{\boldsymbol{F}}) - \boldsymbol{E}(u^\star)\Big) + \max\Big\{r^*, \frac{t}{n}\Big\}$$

$\square$

**Deep Neural Network Estimator.** For any $N \in \mathbb{Z}^+$, there exists some Deep Neural Network in $\Phi(L, W, S, B)$ with $L = O(1)$, $W = O(N)$, $S = O(N)$, $B = O(1)$, such that the approximation error $\Delta\boldsymbol{E}_{\text{app}} = O(N^{-\frac{2(s-2)}{d}})$ and generalization error $\Delta\boldsymbol{E}_{\text{gen}} = O(\frac{N\log N}{n})$. With optimal selection $N = n^{\frac{d}{d+2s-2}}$ to balance the bias and variance, we can achieve $n^{-\frac{2s-2}{d+2s-2}}\log n$ convergence rate for PINN estimator.

**Theorem D.13.** *(Final Upper Bound of PINN with Deep Neural Network Estimator) Consider the sparse Deep Neural Network function space $\Phi(L, W, S, B)$ with parameters $L = O(1)$, $W = O(n^{\frac{d}{d+2s-4}})$, $S = O(n^{\frac{d}{d+2s-4}})$, $B = O(1)$, then the Physics Informed Neural Network estimator $\hat{u}_{PINN}^{DNN} = \min_{u\in\Phi(L,W,S,B)}\boldsymbol{E}_n(u)$ satisfies the following upper bound with high probability*

$$\|\hat{u}_{PINN}^{DNN} - u^*\|_{H_2}^2 \lesssim n^{-\frac{2s-4}{d+2s-4}}\log n.$$

*Proof.* On the one hand, by taking $s = 2$ and $p = 2$ in Theorem D.8 proved above, we have that there exists some Deep Neural Network $u_{\text{DNN}} \in \Phi(L, W, S, B)$ with $L = O(1), W = O(N), S = O(N), B = O(1)$, such that.

$$\|u_{\text{DNN}} - u^*\|^2_{H^2(\Omega)} \leq N^{-\frac{2s-4}{d}} \|u\|_{H^s(\Omega)}.$$

Applying strong convexity of the DRM objective function proved in Section 2.1 further implies:

$$\Delta \boldsymbol{E}_{\text{app}} \lesssim \|u_{\text{DNN}} - u^*\|^2_{H_2(\Omega)} \leq N^{-\frac{2s-4}{d}}.$$

On the other hand, from lemma D.15 proved above, we know that the function $\phi(\rho)$ that upper bounds the local Rademacher complexity of the Deep Neural Networks $u_{\text{DNN}}$ is dominated by the term $\sqrt{\frac{\rho 3^L S}{n} \log(W(B \vee 1)n)}$. By plugging in the magnitudes of $L, W, S, B$, we can determine the thresholding localization radius $\hat{r}$:

$$\sqrt{\frac{\rho 3^L S}{n} \log(W(B \vee 1)n)} \simeq \sqrt{\frac{\rho N}{n}(\log N + \log n)} \simeq \rho \Rightarrow \hat{r} \simeq \frac{N(\log N + \log n)}{n}.$$

Combining the two bounds above gives us:

$$\mathbb{E}_{x \sim \mu}[\Delta \boldsymbol{E}_n] \lesssim \Delta \boldsymbol{E}_{\text{app}} + \hat{r} \lesssim N^{-\frac{2(s-2)}{d}} + \frac{N(\log N + \log n)}{n}.$$

By equating the two terms above, we can solve for the optimal $N$ that yields the desired bound:

$$N^{-\frac{2(s-2)}{d}} \simeq \frac{N}{n} \Rightarrow N \simeq n^{\frac{d}{d+2s-4}}.$$

Plugging in the optimal $N$ gives us the magnitudes of the four parameters $L = O(1)$, $W = O(n^{\frac{d}{d+2s-4}})$, $S = O(n^{\frac{d}{d+2s-4}})$, $B = O(1)$, as well as the final rate:

$$\mathbb{E}_{x \sim \mu}[\Delta \boldsymbol{E}_n] \lesssim N^{-\frac{2(s-2)}{d}} + \frac{N \log N}{n} \lesssim n^{-\frac{2(s-2)}{d+2(s-2)}} \log n.$$

$\square$

**Truncated Fourier Series Estimator.** For any $\xi \in \mathbb{Z}^+$, there exists some Truncated Fourier Series in $\boldsymbol{F}_\xi(\Omega)$ with approximation error $\Delta \boldsymbol{E}_{\text{app}} = O(\xi^{-2(s-2)})$ and generalization error $\Delta \boldsymbol{E}_{\text{gen}} = O(\frac{\xi^d}{n})$

**Theorem D.14.** *(Final Upper Bound of PINN with Truncated Fourier Series Estimator)*

*Consider the PINN objective with a plug in Fourier Series estimator $\hat{u}_{PINN}^{Fourier} = \min_{u \in \boldsymbol{F}_\xi(\Omega)} \boldsymbol{E}_n(u)$ with $\xi = \Theta(n^{\frac{1}{d+2s-4}})$, then we have*

$$\|\hat{u}_{PINN}^{Fourier} - u^*\|^2_{H_2} \lesssim n^{-\frac{2s-4}{d+2s-4}}$$

*Proof.* On the one hand, from Lemma D.5 and Lemma D.6 proved above, we know that the function $\phi(\rho)$ that upper bounds the local Rademacher complexity is dominated by the term $\sqrt{\frac{\rho}{n}} \xi^{\frac{d}{2}}$ for Truncated Fourier Series in $\boldsymbol{F}_\xi(\Omega)$. Thus, the thresholding localization radius $\hat{r}$ can be determined as follows:

$$\sqrt{\frac{\rho}{n}} \xi^{\frac{d}{2}} \simeq \rho \Rightarrow \hat{r} \simeq \frac{\xi^d}{n},$$

On the other hand, by taking $\alpha = s$ and $\beta = 1$ in Lemma D.16 and applying strong convexity of the DRM objective function proved in Section 2.1, we can upper bound the approximation error $\Delta \boldsymbol{E}_{\text{app}}$ as below:

$$\Delta \boldsymbol{E}_{\text{app}} \lesssim \xi^{-2(s-2)},$$

By equating the two terms above, we can solve for $\xi$ that yields the desired bound:

$$\frac{\xi^d}{n} \simeq \xi^{-2(s-2)} \Rightarrow \xi \simeq n^{\frac{1}{d+2s-4}},$$

Plugging in the expression of $\xi$ gives the final upper bound:

$$\mathbb{E}_{x \sim \mu}[\Delta \boldsymbol{E}_n] \lesssim \hat{r} + \Delta \boldsymbol{E}_{\text{app}} \lesssim \frac{\xi^d}{n} + \xi^{-2(s-2)} \simeq n^{-\frac{2s-4}{d+2s-4}}.$$

$\square$

# E  Proof of Modified DRM

**Theorem E.1** (Meta-theorem for Upper Bounds of Modified Deep Ritz Method). *Let $u^* \in H^s(\Omega)$ denote the true solution to the PDE model with Dirichlet boundary condition:*

$$-\Delta u + Vu = f \ on \ \Omega,$$
$$u = 0 \ on \ \partial\Omega, \tag{E.1}$$

*where $f \in L^2(\Omega)$ and $V \in L^\infty(\Omega)$ with $0 < V_{\min} \le V(x) \le V_{\max} > 0$. For a fixed function space $\boldsymbol{F}(\Omega)$, consider the empirical loss induced by the Modified Deep Ritz Method ($N \ge n$):*

$$\boldsymbol{E}_{N,n}(u) = \frac{1}{N}\sum_{i=1}^{N}\left[|\Omega|\cdot\frac{1}{2}|\nabla u(X_i')|^2\right] + \frac{1}{n}\sum_{j=1}^{n}\left[|\Omega|\cdot\left(\frac{1}{2}V(X_j)|u(X_j)|^2 - f(X_j)u(X_j)\right)\right], \tag{E.2}$$

*where $\{X_i'\}_{i=1}^N$ and $\{X_j\}_{j=1}^n$ are datapoints uniformly and independently sampled from the domain $\Omega$. Then the Modified Deep Ritz estimator associated with function space $\boldsymbol{F}(\Omega)$ is defined as the minimizer of $\boldsymbol{E}_{N,n}(u)$ over the function space $\boldsymbol{F}(\Omega)$:*

$$\hat{u}_{MDRM} = \min_{u \in \boldsymbol{F}(\Omega)} \boldsymbol{E}_{N,n}(u)$$

*Moreover, we assume that there exists some constant $C > 0$ such that all function $u$ in the function space $\boldsymbol{F}(\Omega)$, the real solution $u^*$ and $f, V$ satisfy the following two conditions.*

- *The gradients and function value are uniformly bounded*

$$\max\left\{\sup_{u\in\boldsymbol{F}(\Omega)}\|u\|_{L^\infty(\Omega)}, \ \sup_{u\in\boldsymbol{F}(\Omega)}\|\nabla u\|_{L^\infty(\Omega)}, \|u^*\|_{L^\infty(\Omega)}, \|\nabla u^*\|_{L^\infty(\Omega)}, V_{max}, \|f\|_{L^\infty(\Omega)}\right\} \le C. \tag{E.3}$$

- *All the functions in the function space $\boldsymbol{F}(\Omega)$ satisfy the boundary condition*

$$u = 0 \ on \ \partial\Omega.$$

*At the the same time, for any $\rho > 0$, we assume the Rademacher complexity of a localized function space*

$$\boldsymbol{S}_\rho(\Omega) := \left\{h := |\Omega|\cdot\left[\frac{1}{2}\left(|\nabla u|^2 - |\nabla u^*|^2\right) + \frac{1}{2}V(|u|^2 - |u^*|^2) - f(u - u^*)\right] \ \Big| \ \|u-u^*\|_{H_1}^2 \le \rho\right\}$$

*can be upper bounded by a sub-root function $\phi = \phi(\rho) : [0,\infty) \to [0,\infty)$, i.e.*

$$\phi(4\rho) \le 2\phi(\rho) \ and \ R_{N,n}(\boldsymbol{S}_\rho(\Omega)) \le \phi(\rho) \ (\forall \ \rho > 0). \tag{E.4}$$

*For all constant $t > 0$. We denote $r^*$ to be the solution of the fix point equation of local Rademacher complexity $r = \phi(r)$. There exist a constant $C_p$ such that for probability $1 - C_p\exp(-t)$, we have the following upper bound for the Modified Deep Ritz Estimator*

$$\|\hat{u}_{MDRM} - u^*\|_{H_1}^2 \lesssim \inf_{u_{\boldsymbol{F}}\in\boldsymbol{F}(\Omega)}\left(\boldsymbol{E}(u_{\boldsymbol{F}}) - \boldsymbol{E}(u^\star)\right) + \max\left\{r^*, \frac{t}{n}\right\}.$$

*Proof.* To upper bound the excess risk $\Delta\boldsymbol{E}^{(N,n)} := \boldsymbol{E}(\hat{u}_{MDRM}) - \boldsymbol{E}(u^*)$, following[66, 38, 11], we decompose the excess risk into approximation error and generalization error with probability $1 - e^{-t}$:

$$\begin{aligned}
\Delta\boldsymbol{E}^{(N,n)} &= \left[\boldsymbol{E}(\hat{u}_{MDRM}) - \boldsymbol{E}(u^\star)\right] = \left[\boldsymbol{E}(\hat{u}_{MDRM}) - \boldsymbol{E}_{N,n}(\hat{u}_{MDRM})\right] + \left[\boldsymbol{E}_{N,n}(\hat{u}_{MDRM}) - \boldsymbol{E}_{N,n}(u_{\boldsymbol{F}})\right] \\
&\quad + \left[\boldsymbol{E}_{N,n}(u_{\boldsymbol{F}}) - \boldsymbol{E}(u_{\boldsymbol{F}})\right] + \left[\boldsymbol{E}(u_{\boldsymbol{F}}) - \boldsymbol{E}(u^\star)\right] \\
&\le \left[\boldsymbol{E}(\hat{u}_{MDRM}) - \boldsymbol{E}_{N,n}(\hat{u}_{MDRM})\right] + \left[\boldsymbol{E}_{N,n}(u_{\boldsymbol{F}}) - \boldsymbol{E}(u_{\boldsymbol{F}})\right] + \left[\boldsymbol{E}(u_{\boldsymbol{F}}) - \boldsymbol{E}(u^\star)\right] \\
&\lesssim \left[\boldsymbol{E}(\hat{u}_{MDRM}) - \boldsymbol{E}(u^*) + \boldsymbol{E}_{N,n}(u^*) - \boldsymbol{E}_{N,n}(\hat{u}_{MDRM})]\right] \\
&\quad + 2\left[\boldsymbol{E}(u_{\boldsymbol{F}}) - \boldsymbol{E}(u^\star)\right] + \frac{t}{\min\{N,n\}},
\end{aligned} \tag{E.5}$$

where the expectation is on all sampled data. The inequality of the third line is because the $u$ is the minimizer of the empirical loss $E_n$ in the solution set $F(\Omega)$, so we have $E_{N,n}(u) \leq E_{N,n}(u_F)$. The last inequality is based on the Bernstein inequality. The variance can be bounded by $\big[E(u_F) - E(u^\star)\big]$ due to the strong convexity of the variation objective. According to the Brenstein inequality, we know with probability $1 - e^{-t}$ we have

$$E_{N,n}(u_F) - E_{N,n}(u^*) - E(u_F) + E(u^*) \leq \sqrt{\frac{2t\big[E(u_F) - E(u^\star)\big]}{\min\{N,n\}}} \leq \big[E(u_F) - E(u^\star)\big] + \frac{4t}{\min\{N,n\}}.$$

Note that E.5 holds for all function lies in the function space $F$. Thus, we can take $u_F := \arg\min_{u_F \in F(\Omega)} \big(E(u_F) - E(u^\star)\big)$ and finally get

$$\Delta E^{N,n} \lesssim \underbrace{\mathbb{E}_{x,x'\sim\mu}[E(\hat{u}_{\text{MDRM}}) - E(u^*) + E_{N,n}(u^*) - E_{N,n}(\hat{u}_{\text{MDRM}})]}_{\Delta E_{\text{gen}}} + \underbrace{\inf_{u_F \in F(\Omega)}\big(E(u_F) - E(u^\star)\big)}_{\Delta E_{\text{app}}} + \frac{t}{n}.$$

This inequality decomposes the excess risk to the generalization error $\Delta E_{\text{gen}} := \mathbb{E}_{x\sim\mu}[E(\hat{u}_{\text{MDRM}}) - E(u^*) + E_{N,n}(u^*) - E_{N,n}(\hat{u}_{\text{MDRM}})]$ and the approximation error $\Delta E_{\text{app}} = \inf_{u_F \in F(\Omega)}\big(E(u_F) - E(u^\star)\big)$. From the lemmata proved in Section D.3, we already have an estimation of the approximation error's convergence rate. So now we'll focus on providing fast rate upper bounds of the generalization error for the two estimators using the localization techinque[2, 66]. To achieve the fast generalization bound, we focus on the following two normalized empirical processes:

$$\tilde{S}_{r,1}(\Omega) := \{\tilde{h}_1(x) := \frac{\mathbb{E}[h_1] - h_1(x)}{\mathbb{E}[h_1] + \mathbb{E}[h_2] + r} \mid h = h_1 + h_2 \in S(\Omega)\} \ (r > 0),$$

$$\tilde{S}_{r,2}(\Omega) := \{\tilde{h}_2(x) := \frac{\mathbb{E}[h_2] - h_2(x)}{\mathbb{E}[h_1] + \mathbb{E}[h_2] + r} \mid h = h_1 + h_2 \in S(\Omega)\} \ (r > 0).$$

First, we try to bound the expectation of the two normalized empirical processes. Applying the Symmetrization Lemma D.1, we can first bound the two expectations as:

$$\sup_{\tilde{h}_1 \in \tilde{S}_{r,1}(\Omega)} \mathbb{E}_{y'}\left[\frac{1}{N}\sum_{i=1}^{N}\tilde{h}_1(y_i')\right] \leq \mathbb{E}_{y'}\left[\sup_{h_1 \in S_1(\Omega)}\left|\frac{1}{N}\sum_{i=1}^{N}\frac{h_1(y_i') - \mathbb{E}[h_1]}{\mathbb{E}[h_1] + \mathbb{E}[h_2] + r}\right|\right] \leq 2R_N(\hat{S}_{r,1}(\Omega)),$$

$$\sup_{\tilde{h}_2 \in \tilde{S}_{r,2}(\Omega)} \mathbb{E}_{y}\left[\frac{1}{n}\sum_{j=1}^{n}\tilde{h}_2(y_j)\right] \leq \mathbb{E}_{y}\left[\sup_{h_2 \in S_2(\Omega)}\left|\frac{1}{n}\sum_{i=1}^{n}\frac{h_2(y_j) - \mathbb{E}[h_2]}{\mathbb{E}[h_1] + \mathbb{E}[h_2] + r}\right|\right] \leq 2R_n(\hat{S}_{r,2}(\Omega)).$$

where the function classes $\hat{S}_{r,k}(\Omega)$ $(1 \leq k \leq 2)$ are defined as:

$$\hat{S}_{r,1}(\Omega) := \{\hat{h}_1(x) := \frac{h_1(x)}{\mathbb{E}[h_1] + \mathbb{E}[h_2] + r} \mid h = h_1 + h_2 \in S(\Omega)\},$$

$$\hat{S}_{r,2}(\Omega) := \{\hat{h}_2(x) := \frac{h_2(x)}{\mathbb{E}[h_1] + \mathbb{E}[h_2] + r} \mid h = h_1 + h_2 \in S(\Omega)\},$$

Applying the Peeling Lemma to any function $h \in S(\Omega)$ helps us upper bound the sum of the two local Rademacher complexities $R_N(\hat{S}_{r,1}(\Omega)) + R_n(\hat{S}_{r,2}(\Omega))$ with the function $\phi$ defined in equation:

$$R_N(\hat{S}_{r,1}(\Omega)) + R_n(\hat{S}_{r,2}(\Omega)) = \mathbb{E}_\sigma\left[\mathbb{E}_y\left[\sup_{h \in S(\Omega)}\frac{\frac{1}{N}\sum_{i=1}^{N}\sigma_i h_1(y_i)}{\mathbb{E}[h] + r}\right]\right] + \mathbb{E}_\tau\left[\mathbb{E}_{y'}\left[\sup_{h \in S(\Omega)}\frac{\frac{1}{n}\sum_{j=1}^{n}\tau_j h_2(y_j')}{\mathbb{E}[h] + r}\right]\right]$$

$$= \mathbb{E}_\sigma\left[\mathbb{E}_{y,y'}\left[\sup_{h \in S(\Omega)}\frac{\frac{1}{N}\sum_{i=1}^{N}\sigma_i h_1(y_i)}{\mathbb{E}[h] + r} + \sup_{h \in S(\Omega)}\frac{\frac{1}{n}\sum_{j=1}^{n}\tau_j h_2(y_j')}{\mathbb{E}[h] + r}\right]\right]$$

$$= R_{N,n}(\hat{S}_r(\Omega)) \leq \frac{4\phi(r)}{r}.$$

Combining all inequalities derived above yields:

$$\sup_{\tilde{h}_1 \in \tilde{S}_{r,1}(\Omega)} \mathbb{E}_{y'} \left[ \frac{1}{N} \sum_{i=1}^{N} \tilde{h}_1(y_i') \right] + \sup_{\tilde{h}_2 \in \tilde{S}_{r,2}(\Omega)} \mathbb{E}_y \left[ \frac{1}{n} \sum_{j=1}^{n} \tilde{h}_2(y_j) \right] \tag{E.6}$$

$$\leq 2 R_N(\hat{\boldsymbol{S}}_{r,1}(\Omega)) + 2 R_n(\hat{\boldsymbol{S}}_{r,2}(\Omega)) = 2 R_{N,n}(\hat{\boldsymbol{S}}_r(\Omega)) \leq \frac{8\phi(r)}{r} \ (r > 0).$$

Secondly we'll apply the Talagrand concentration inequality to the two function classes $\tilde{S}_{r,1}(\Omega)$ and $\tilde{S}_{r,2}(\Omega)$, which requires us to verify the conditions needed. We will first check that the expectation value $\mathbb{E}[h]$ is always non-negative for any $h \in \boldsymbol{S}(\Omega)$:

$$\mathbb{E}[h] = \frac{1}{|\Omega|} \int_\Omega |\Omega| \cdot (\frac{1}{2}|\nabla u(x)|^2 + \frac{1}{2}V(x)|u(x)|^2 - f(x)u(x))dx$$

$$- \frac{1}{|\Omega|} \int_\Omega |\Omega| \cdot (\frac{1}{2}|\nabla u^\star(x)|^2 + \frac{1}{2}V(x)|u^\star(x)|^2 - f(x)u^\star(x))dx$$

$$= \boldsymbol{E}(u) - \boldsymbol{E}(u^\star) \geq 0 \Rightarrow \mathbb{E}[h] \geq 0.$$

Next, We will verify that $\tilde{S}_{r,1}(\Omega)$ satisfies all three requirements. At first, we will show that any $\tilde{h}_1 = \frac{\mathbb{E}[h_1] - h_1}{\mathbb{E}[h] + r} \in \tilde{S}_{r,1}(\Omega)$ is of bounded inf-norm. We need to prove that any $h_1 \in \boldsymbol{S}_1(\Omega)$ is of bounded inf-norm beforehand. Using boundedness condition listed in equation E.3 implies:

$$\|h_1\|_\infty = \|\frac{1}{2}\left(|\nabla u|^2 - |\nabla u^*|^2\right)\|_\infty \leq \frac{1}{2}\left(\|\nabla u\|_\infty^2 + \|\nabla u^*\|_\infty^2\right) \leq C^2.$$

By taking $M_1 := C^2$, we then have $\|h_1\|_\infty \leq M_1$ for all $h_1 \in \boldsymbol{S}_1(\Omega)$. Note that the denominator of $\tilde{h}_1$ can be lower bounded by $|\mathbb{E}[h] + r| \geq r > 0$. Combining these two inequalities help us upper bound the inf-norm $\|\tilde{h}_1\|_\infty = \sup_{x \in \Omega} |\tilde{h}_1(x)|$ as follows:

$$\|\tilde{h}_1\|_\infty = \frac{\|\mathbb{E}[h_1] - h_1\|_\infty}{|\mathbb{E}[h] + r|} \leq \frac{2\|h_1\|_\infty}{r} \leq \frac{2M_1}{r} =: \beta_1.$$

Also, it's easy to check that for any $\tilde{h}_1 \in \tilde{\boldsymbol{S}}_{r,1}(\Omega)$, we have

$$\mathbb{E}[\tilde{h}_1] = \frac{\mathbb{E}[h_1] - \mathbb{E}[h_1]}{\mathbb{E}[h] + r} = 0,$$

*i.e.* any function in the localized class $\tilde{\boldsymbol{S}}_{r,1}(\Omega)$ is of zero mean.

Moreover, we take $\sigma_1^2 = \sup_{\tilde{h}_1 \in \tilde{S}_{r,1}(\Omega)} \mathbb{E}[\tilde{h}_1^2]$ to be the upper bound on the second moment of functions in $\tilde{S}_{r,1}(\Omega)$. Now we have verified that any function $\tilde{h}_1 \in \tilde{S}_{r,1}(\Omega)$ satisfies all the required conditions. By taking $\mu$ to be the uniform distribution on the domain $\Omega$ and applying Talagrand's Concentration inequality given in Lemma D.3, we have:

$$\mathbb{P}_x \left[ \sup_{\tilde{h}_1 \in \tilde{\boldsymbol{S}}_{r,1}(\Omega)} \frac{1}{N} \sum_{i=1}^{N} \tilde{h}_1(x_i) \geq 2 \sup_{\tilde{h}_1 \in \tilde{\boldsymbol{S}}_{r,1}(\Omega)} \mathbb{E}_y \left[ \frac{1}{N} \sum_{i=1}^{N} \tilde{h}_1(y_i) \right] + \sqrt{\frac{2t\sigma_1^2}{N}} + \frac{2t\beta_1}{N} \right] \leq e^{-t}. \tag{E.7}$$

Moreover, We will verify that $\tilde{S}_{r,2}(\Omega)$ also satisfies all three requirements. At first, we will show that any $\tilde{h}_2 = \frac{\mathbb{E}[h_2] - h_2}{\mathbb{E}[h] + r} \in \tilde{\boldsymbol{S}}_{r,2}(\Omega)$ is of bounded inf-norm. We need to prove that any $h_2 \in \boldsymbol{S}_2(\Omega)$ is of bounded inf-norm beforehand. Using boundedness condition listed in equation E.3 implies:

$$\|h_2\|_\infty = \|\frac{1}{2}V(|u|^2 - |u^*|^2) - f(u - u^*)\|_\infty$$

$$\leq \frac{1}{2}V_{\max}\left(\|u\|_\infty^2 + \|u^*\|_\infty^2\right) + \|f\|_\infty\left(\|u\|_\infty + \|u^*\|_\infty\right)$$

$$\leq \frac{1}{2}V_{\max} \times 2C^2 + 2C^2 = (V_{\max} + 2)C^2.$$

By taking $M_2 := (V_{\max} + 2)C^2$, we then have $\|h_2\|_\infty \leq M_2$ for all $h_2 \in \boldsymbol{S}_2(\Omega)$. Note that the denominator of $\tilde{h}_2$ can be lower bounded by $|\mathbb{E}[h] + r| \geq r > 0$. Combining these two inequalities help us upper bound the inf-norm $\|\tilde{h}_2\|_\infty = \sup_{x \in \Omega} |\tilde{h}_2(x)|$ as follows:

$$\|\tilde{h}_2\|_\infty = \frac{\|\mathbb{E}[h_2] - h_2\|_\infty}{|\mathbb{E}[h] + r|} \leq \frac{2\|h_2\|_\infty}{r} \leq \frac{2M_2}{r} =: \beta_2.$$

Also, it's easy to check that for any $\tilde{h}_2 \in \tilde{\boldsymbol{S}}_{r,2}(\Omega)$, we have

$$\mathbb{E}[\tilde{h}_2] = \frac{\mathbb{E}[h_2] - \mathbb{E}[h_2]}{\mathbb{E}[h] + r} = 0,$$

*i.e.* any function in the localized class $\tilde{\boldsymbol{S}}_{r,2}(\Omega)$ is of zero mean.

Moreover, we take $\sigma_2^2 = \sup_{\tilde{h}_2 \in \tilde{\boldsymbol{S}}_{r,2}(\Omega)} \mathbb{E}[\tilde{h}_2^2]$ to be the upper bound on the second moment of functions in $\tilde{\boldsymbol{S}}_{r,2}(\Omega)$. Now we have verified that any function $\tilde{h}_2 \in \tilde{\boldsymbol{S}}_{r,2}(\Omega)$ satisfies all the required conditions. By taking $\mu$ to be the uniform distribution on the domain $\Omega$ and applying Talagrand's Concentration inequality given in Lemma D.3, we have:

$$\mathbb{P}_{x'}\left[\sup_{\tilde{h}_2 \in \tilde{\boldsymbol{S}}_{r,2}(\Omega)} \frac{1}{n} \sum_{j=1}^{n} \tilde{h}_2(x_j') \geq 2 \sup_{\tilde{h}_2 \in \tilde{\boldsymbol{S}}_{r,2}(\Omega)} \mathbb{E}_{y'}\left[\frac{1}{n} \sum_{j=1}^{n} \tilde{h}_2(y_j')\right] + \sqrt{\frac{2t\sigma_2^2}{n}} + \frac{2t\beta_2}{n}\right] \leq e^{-t}.$$
(E.8)

By applying a union bound to the two inequalities derived in E.7 and E.8, we can derive that with probability at least $1 - 2e^{-t}$, the inequality below holds:

$$\frac{1}{N} \sum_{i=1}^{N} \tilde{h}_1(x_i') + \frac{1}{n} \sum_{j=1}^{n} \tilde{h}(x_j) \leq \sup_{\tilde{h}_1 \in \tilde{\boldsymbol{S}}_{r,1}(\Omega)} \frac{1}{N} \sum_{i=1}^{N} \tilde{h}_1(x_i) + \sup_{\tilde{h}_2 \in \tilde{\boldsymbol{S}}_{r,2}(\Omega)} \frac{1}{n} \sum_{j=1}^{n} \tilde{h}_2(x_j')$$

$$\leq 2 \sup_{\tilde{h}_1 \in \tilde{\boldsymbol{S}}_{r,1}(\Omega)} \mathbb{E}_y\left[\frac{1}{N} \sum_{i=1}^{N} \tilde{h}_1(y_i)\right] + \sqrt{\frac{2t\sigma_1^2}{N}} + \frac{2t\beta_1}{N}$$

$$+ 2 \sup_{\tilde{h}_2 \in \tilde{\boldsymbol{S}}_{r,2}(\Omega)} \mathbb{E}_{y'}\left[\frac{1}{n} \sum_{j=1}^{n} \tilde{h}_2(y_j')\right] + \sqrt{\frac{2t\sigma_2^2}{n}} + \frac{2t\beta_2}{n}$$

$$\leq \frac{16\phi(r)}{r} + \sqrt{\frac{2t}{n}}(\sigma_1 + \sigma_2) + \frac{2t(\beta_1 + \beta_2)}{n}$$

By the definition of $\beta_1$ and $\beta_2$, we have that the term $\frac{2t(\beta_1+\beta_2)}{n}$ can be upper bounded by:

$$\frac{2t(\beta_1 + \beta_2)}{n} = \frac{4t(M_1 + M_2)}{nr} \leq \frac{4(V_{\max} + 3)C^2 t}{nr}$$

Now we will derive some upper bound on the sum $\sigma_1 + \sigma_2$. By definition we have that:

$$(\sigma_1 + \sigma_2)^2 \leq 2(\sigma_1^2 + \sigma_2^2) = 2\left[\sup_{\tilde{h}_1 \in \tilde{\boldsymbol{S}}_{r,1}(\Omega)} \mathbb{E}[\tilde{h}_1^2] + \sup_{\tilde{h}_1 \in \tilde{\boldsymbol{S}}_{r,1}(\Omega)} \mathbb{E}[\tilde{h}_1^2]\right]$$

$$= 2\left[\sup_{h \in \boldsymbol{S}(\Omega)} \frac{\mathbb{E}[h_1^2] - \mathbb{E}[h_1]^2}{|\mathbb{E}[h] + r|^2} + \sup_{h \in \boldsymbol{S}(\Omega)} \frac{\mathbb{E}[h_2^2] - \mathbb{E}[h_2]^2}{|\mathbb{E}[h] + r|^2}\right]$$

$$\leq 4 \sup_{h \in \boldsymbol{S}(\Omega)} \frac{\mathbb{E}[h_1^2] + \mathbb{E}[h_2^2]}{|\mathbb{E}[h] + r|^2}$$

Now it suffices to derive an upper bound of $\frac{\mathbb{E}[h_1^2] + \mathbb{E}[h_2^2]}{|\mathbb{E}[h] + r|^2}$ for any $h \in \boldsymbol{S}(\Omega)$. The existence of such an upper bound is guaranteed because of the regularity results of the PDE. We aim to show that there exist some constants $\alpha, \alpha' > 0$, such that for any $h \in \boldsymbol{S}(\Omega)$, the following inequality holds:

$$\alpha(\mathbb{E}[h_1^2] + \mathbb{E}[h_2^2]) \leq \|u - u^*\|_{H^1(\Omega)}^2 \leq \alpha' \mathbb{E}[h].$$
(E.9)

The RHS of the inequality follows from strong convexity of the DRM objective function proved in Section 2.1:

$$\mathbb{E}[h] = \boldsymbol{E}(u) - \boldsymbol{E}(u^*) \geq \frac{\min\{1, V_{\min}\}}{4} \|u - u^*\|^2_{H^1(\Omega)}$$

The LHS of the inequality follows from boundedness condition listed in equation E.3 and the QM-AM inequality:

$$
\begin{aligned}
\mathbb{E}[h_1^2] + \mathbb{E}[h_2^2] &= \int_\Omega \frac{1}{4}\left(|\nabla u|^2 - |\nabla u^*|^2\right)^2 dx + \int_\Omega \left[\frac{1}{2}V(|u|^2 - |u^*|^2) - f(u - u^*)\right]^2 dx \\
&\leq \frac{1}{4}\int_\Omega \left(|\nabla u|^2 - |\nabla u^*|^2\right)^2 dx + \frac{1}{2}\int_\Omega V^2(|u|^2 - |u^*|^2)^2 dx + 2\int_\Omega f^2(u - u^*)^2 dx \\
&\leq \frac{1}{4}\int_\Omega \left||\nabla u| - |\nabla u^*|\right|^2(|\nabla u| + |\nabla u^*|)^2 dx + \frac{1}{2}V_{\max}^2 \int_\Omega \left||u| - |u^*|\right|^2(|u| + |u^*|)^2 dx \\
&\quad + 2C^2 \int_\Omega (u - u^*)^2 dx \leq C^2 \int_\Omega |\nabla u - \nabla u^*|^2 dx + 2C^2(1 + V_{\max}^2)\int_\Omega |u - u^*|^2 dx \\
&\leq 2C^2(1 + V_{\max}^2)\|u - u^*\|^2_{H^1(\Omega)}.
\end{aligned}
$$

By picking $\alpha' = \frac{4}{\min\{1, V_{\min}\}}$ and $\alpha = \frac{1}{2C^2(1 + V_{\max}^2)}$, we have finished proving inequality E.9. Then we can can upper bound the term $\frac{\mathbb{E}[h_1^2] + \mathbb{E}[h_2^2]}{|\mathbb{E}[h] + r|^2}$ as:

$$\frac{\mathbb{E}[h_1^2] + \mathbb{E}[h_2^2]}{|\mathbb{E}[h] + r|^2} \leq \frac{\frac{\alpha'}{\alpha}\mathbb{E}[h]}{2r\mathbb{E}[h]} \leq \frac{\alpha'}{2\alpha r}.$$

Combining the bounds derived above helps us upper bound the term $\sqrt{\frac{2t}{n}}(\sigma_1 + \sigma_2)$ as below:

$$\sqrt{\frac{2t}{n}}(\sigma_1 + \sigma_2) \leq \sqrt{\frac{8t}{n}}\sqrt{\sup_{h \in \boldsymbol{S}(\Omega)} \frac{\mathbb{E}[h_1^2] + \mathbb{E}[h_2^2]}{|\mathbb{E}[h] + r|^2}} \leq \sqrt{\frac{4\alpha' t}{n\alpha r}}$$

Thus, using the two upper bounds on $\sqrt{\frac{2t}{n}}(\sigma_1 + \sigma_2)$ and $\frac{2t(\beta_1 + \beta_2)}{n}$, we have :

$$
\begin{aligned}
\frac{1}{N}\sum_{i=1}^N \tilde{h}_1(x_i') + \frac{1}{n}\sum_{j=1}^n \tilde{h}(x_j) &\leq \frac{16\phi(r)}{r} + \sqrt{\frac{2t}{n}}(\sigma_1 + \sigma_2) + \frac{2t(\beta_1 + \beta_2)}{n} \\
&\leq \frac{16\phi(r)}{r} + \sqrt{\frac{4\alpha' t}{n\alpha r}} + \frac{4(V_{\max} + 3)C^2 t}{nr} = \psi(r)
\end{aligned}
$$

Let's pick the threshold radius $r_0$ to be:

$$r_0 = \max\{2^{14} r^*, \frac{24Mt}{n}, \frac{144\alpha' t}{\alpha n}\}. \tag{E.10}$$

Note that concavity of the function $\phi$ implies that $\phi(r) \leq r$ for any $r \geq r^*$. Combining this with the first inequality listed in E.4 yields:

$$\frac{16\phi(r_0)}{r_0} \leq \frac{2^{11}\phi(\frac{r_0}{2^{14}})}{2^{14}\frac{r_0}{2^{14}}} = \frac{1}{8} \times \frac{\phi(\frac{r_0}{2^{14}})}{\frac{r)}{2^{14}}} \leq \frac{1}{8}.$$

On the other hand, applying equation E.10 yields:

$$
\begin{aligned}
\sqrt{\frac{4\alpha' t}{n\alpha r_0}} &\leq \sqrt{\frac{4\alpha' t}{n\alpha}\frac{\alpha n}{144\alpha' t}} = \frac{1}{6}, \\
\frac{4(V_{\max} + 3)C^2 t}{nr_0} &\leq \frac{4(V_{\max} + 3)C^2 t}{n} \times \frac{n}{24(V_{\max} + 3)C^2 t} = \frac{1}{6}.
\end{aligned}
$$

Summing the three inequalities above implies:

$$\psi(r_0) = \frac{16\phi(r_0)}{r_0} + \sqrt{\frac{4\alpha' t}{n\alpha r_0}} + \frac{4(V_{\max} + 3)C^2 t}{nr_0} \leq \frac{1}{8} + \frac{1}{6} + \frac{1}{6} < \frac{1}{2}.$$

By picking $r = r_0$, we can further deduce that for any function $u \in \boldsymbol{F}(\Omega)$, the following inequality holds with probability $1 - e^{-t}$:

$$\frac{\boldsymbol{E}(u) - \boldsymbol{E}(u^*) - \boldsymbol{E}_n(u) + \boldsymbol{E}_n(u^*)}{\boldsymbol{E}(u) - \boldsymbol{E}(u^*) + r_0} = \frac{1}{n}\sum_{i=1}^{n} \tilde{h}(x_i) \le \psi(r_0) < \frac{1}{2}.$$

Multiplying the denominator on both sides indicates:

$$\Delta \boldsymbol{E}_{\text{gen}} = \boldsymbol{E}(u) - \boldsymbol{E}(u^*) - \boldsymbol{E}_n(u) + \boldsymbol{E}_n(u^*) \le \frac{1}{2}\Big[\boldsymbol{E}(u) - \boldsymbol{E}(u^*)\Big] + \frac{1}{2}r_0 = \frac{1}{2}\Delta \boldsymbol{E}^{(n)} + \frac{1}{2}r_0.$$

Substituting the upper bound above into the decomposition $\Delta \boldsymbol{E}^{(n)} \le \Delta \boldsymbol{E}_{\text{gen}} + \frac{3}{2}\Delta \boldsymbol{E}_{\text{app}} + \frac{t}{2n}$ yields that with probability $1 - e^{-t}$, we have:

$$\Delta \boldsymbol{E}^{(n)} \le \Delta \boldsymbol{E}_{\text{gen}} + \frac{3}{2}\Delta \boldsymbol{E}_{\text{app}} + \frac{t}{2n} \le \frac{1}{2}\Delta \boldsymbol{E}^{(n)} + \frac{1}{2}r_0 + \frac{3}{2}\Delta \boldsymbol{E}_{\text{app}} + \frac{t}{2n}.$$

Simplifying the inequality above yields that with probability $1 - e^{-t}$, we have:

$$\Delta \boldsymbol{E}^{(n)} \le r_0 + 3\Delta \boldsymbol{E}_{\text{app}} + \frac{t}{n} = 3 \inf_{u_{\boldsymbol{F}} \in \boldsymbol{F}(\Omega)} \Big(\boldsymbol{E}(u_{\boldsymbol{F}}) - \boldsymbol{E}(u^\star)\Big) + \max\{2^{14}r^*, 24M\frac{t}{n}, \frac{36\alpha'}{\alpha}\frac{t}{n}\} + \frac{t}{n}$$

$$\lesssim \inf_{u_{\boldsymbol{F}} \in \boldsymbol{F}(\Omega)} \Big(\boldsymbol{E}(u_{\boldsymbol{F}}) - \boldsymbol{E}(u^\star)\Big) + \max\Big\{r^*, \frac{t}{n}\Big\}$$

Moreover, using strong convexity of the DRM objective function proved in Theorem D.1 implies:

$$\Delta \boldsymbol{E}^{(n)} = \boldsymbol{E}(\hat{u}_{\text{MDRM}}) - \boldsymbol{E}(u^*) \ge \{1, V_{\min}\}\|\hat{u}_{\text{MDRM}} - u^*\|^2_{H^1(\Omega)}$$

Combining the two bounds above yields that with probability $1 - e^{-t}$, we have:

$$\|\hat{u}_{\text{MDRM}} - u^*\|^2_{H^1(\Omega)} \lesssim \inf_{u_{\boldsymbol{F}} \in \boldsymbol{F}(\Omega)} \Big(\boldsymbol{E}(u_{\boldsymbol{F}}) - \boldsymbol{E}(u^\star)\Big) + \max\Big\{r^*, \frac{t}{n}\Big\}$$

$\square$

**Truncated Fourier Series Estimator.** Next we aim to show that the truncated Fourier series estimator can achieve the min-max optimal rate using the MDRM objective function. For any $\xi \in \mathbb{Z}^+$, there exists some Truncated Fourier Series in $\boldsymbol{F}_\xi(\Omega)$ with approximation error $\Delta \boldsymbol{E}_{\text{app}} = O(\xi^{-2(s-2)})$ and generalization error $\Delta \boldsymbol{E}_{\text{gen}} = O(\frac{\xi^d}{n})$

**Theorem E.2.** *(Final Upper Bound of PINN with Truncated Fourier Series Estimator) Consider the PINN objective with a plug in Fourier Series estimator $\hat{u}_{PINN}^{Fourier} = \min_{u \in \boldsymbol{F}_\xi(\Omega)} \boldsymbol{E}_n(u)$ with $\xi = \Theta(n^{\frac{1}{d+2s-4}})$, then we have*

$$\|\hat{u}_{PINN}^{Fourier} - u^*\|^2_{H_2} \lesssim n^{-\frac{2s-4}{d+2s-4}}$$

*Proof.* On the one hand, from Lemma D.5 and Lemma D.6 proved above, we know that the function $\phi(\rho)$ that upper bounds the local Rademacher complexity is dominated by the term $\sqrt{\frac{\rho}{n}}\xi^{\frac{d}{2}}$ for Truncated Fourier Series in $\boldsymbol{F}_\xi(\Omega)$. Thus, the thresholding localization radius $\hat{r}$ can be determined as follows:

$$\sqrt{\frac{\rho}{n}}\xi^{\frac{d-2}{2}} + \sqrt{\frac{\rho}{N}}\xi^{\frac{d}{2}} \simeq \rho \Rightarrow \hat{r} \simeq \frac{\xi^d}{n},$$

On the other hand, by taking $\alpha = s$ and $\beta = 1$ in Lemma D.16 and applying strong convexity of the DRM objective function proved in Section 2.1, we can upper bound the approximation error $\Delta \boldsymbol{E}_{\text{app}}$ as below:

$$\Delta \boldsymbol{E}_{\text{app}} \lesssim \xi^{-2(s-1)},$$

By equating the two terms above, we can solve for $\xi$ that yields the desired bound:

$$\frac{\xi^{d-2}}{n} + \frac{\xi^d}{N} \simeq \xi^{-2(s-1)} \Rightarrow \xi \simeq n^{\frac{1}{d+2s-4}},$$

Note that this is because $\frac{\xi^d}{N} < \frac{\xi^{d-2}}{n}$. Plugging in the expression of $\xi$ gives the final upper bound:

$$\mathbb{E}_{x \sim \mu}[\Delta \boldsymbol{E}_n] \lesssim \hat{r} + \Delta \boldsymbol{E}_{\text{app}} \lesssim \frac{\xi^{d-2}}{n} + \xi^{-2(s-2)} \simeq n^{-\frac{2s-2}{d+2s-4}}.$$

$\square$

# F Proof of the Lower Bounds

## F.1 Preliminaries on Tools for Lower Bounds

In this section, we repeat the standard tools we use to establish the lower bound. The main tool we use is the Fano's inequailty and the Varshamov-Gilber Lemma.

**Lemma F.1** (Fano's methods). *Assume that $V$ is a unifrom random variable over set $\mathcal{V}$, then for any markov chain $V \rightarrow X \rightarrow \hat{V}$, we always have*

$$\mathcal{P}(\hat{V} \neq V) \geq 1 - \frac{I(V;X) + \log 2}{\log(|\mathcal{V}|)}$$

**Lemma F.2** (Varshamov-Gillbert Lemma,[62] Theorem 2.9). *Let $D \geq 8$. There exists a subset $\mathcal{V} = \{\tau^{(0)}, \cdots, \tau^{(2^{D/8})}\}$ of $D-$dimensional hypercube $\mathcal{H}^D = \{0,1\}^D$ such that $\tau^{(0)} = (0,0,\cdots,0)$ and the $\ell_1$ distance between every two elements is larger than $\frac{D}{8}$*

$$\sum_{l=1}^{D} \|\tau^{(j)} - \tau^{(k)}\|_{\ell_1} \geq \frac{D}{8}, \textit{for all } 0 \leq j,k \leq 2^{D/8}$$

## F.2 Proof Of Lower Bound

In this section, we provide the proof of the lower bound for learning a PDE. Our proof uses standard Fano method to establish minimax lower bound but finally leads to a non-standard convergence rate. We state standard results for Fano methods in Appendix F.1. Following is the proof our main lower bound.

**Theorem F.1** (Lower bound). *We denote $u^*(f)$ to be the solution of the PDE 2.1 and we can aceess randomly sampled data $\{X_i, Y_i\}_{i=1,\cdots,n}$ as described in Section 2.2.*

**DRM Lower Bound.** *For all estimator $H : \left(\mathbb{R}^d\right)^{\otimes n} \times \mathbb{R}^{\otimes n} \rightarrow W_1^\infty(\Omega)$, we have*

$$\inf_H \sup_{u \in W_s^\infty(\Omega)} \mathbb{E}\|H(\{X_i, f_i\}_{i=1,\cdots,n}) - u^*(f)\|_{H_1}^2 \gtrsim n^{-\frac{2\alpha-2}{d+2\alpha-4}}. \tag{F.1}$$

**PINN Lower Bound.** *For all estimator $H : \left(\mathbb{R}^d\right)^{\otimes n} \times \mathbb{R}^{\otimes n} \rightarrow W_1^\infty(\Omega)$, we have*

$$\inf_H \sup_{u \in W_s^\infty(\Omega)} \mathbb{E}\|H(\{X_i, f_i\}_{i=1,\cdots,n}) - u^*(f)\|_{H_2}^2 \gtrsim n^{-\frac{2\alpha-4}{d+2\alpha-4}}. \tag{F.2}$$

*Proof.* We construct the following bump function to construct the multiple hypothesis test used for proving the lower bound. Consider a simple $C^\infty$ bump function supported on $[0,1]^d$

$$g(x) = \prod_{i=1}^{d} \xi(x_i), x = (x_1, \cdots, x_d),$$

where $\xi : \mathbb{R} \rightarrow \mathbb{R}$ be a non-zero funtion in $C^\infty(\mathbb{R})$ with support contained in $[0,1]$ and satisfies $\xi(x) \neq 0, \frac{d}{dx}\xi(x) \neq 0$. Then $\nabla g(x) \neq 0$ and the support of function $g$ is $[0,1]^d$.

Next, take $m = [n^{\frac{1}{2\alpha-4+d}}]$ and let's consider a regular gird $x^{(j)}, j \in [m]^d$. According to the Varshamov-Gilbert lemma, we can find $2^{m^d/8}$ $(0,1)$-sequence $\tau_1, \cdots, \tau_{2^{m^d/8}} \in \{0,1\}^{m^d}$ such that $\|\tau_k - \tau_{k'}\|_2^2 \geq \frac{m^d}{8}$ for all $0 < k \neq k' \leq 2^{m^d/8}$. Then we construct the multiple hypothesis as

$$u_k(x) = \sum_{j \in [m_1]^d} \tau_k(j) \frac{\omega}{m^{\alpha+\frac{d}{2}}} g(m(x - x^{(j)})), k = 1, 2, \cdots, 2^{m^d/8},$$

where $\omega$ is a constant to be determined later. It's easy to find out that $u_k \in \mathcal{C}^\alpha$.

Then we reduce solving PDE to the multiple hypothesis testing which considers all mappings from $n$ sampled data to the constructed hypothesis $\Psi : \left(\mathbb{R}^d\right)^{\otimes n} \times \mathbb{R}^{\otimes n} \rightarrow \mathcal{V} := \{u_i | i = 1, 2, \cdots, 2^{m^d/8}\}$.

Then we apply the local Fano method and check that we can obtain a constant lower bound of $\mathcal{P}(\hat{V} \neq V)$ for any estimator $\hat{V}$. Applying the local Fano methods, we know that

$$I(V; X) \leq \frac{1}{|\mathcal{V}|^2} \sum_z \sum_{v \neq v'} D_{KL}(P_v \| P'_v)$$

where $P_k$ joint distribution of the sampled data $(x, y)$, in specific, $x$ follows a uniform distribution on $[0, 1]^d$ and $y = f(X) + \sigma \epsilon$ where $\epsilon$ is independently sampled from a standard Gaussian distribution$N(0, 1)$. Then we have

$$KL(P_k \| P_{k'}) = \mathbb{E} \log(\frac{dP_k}{dP}) = \|\Delta u_k + u_k\|_{L_2}^2 = \frac{C\omega}{m^{2\alpha - 4}}$$

Using Fano inequality, when taking $\omega$ large enough, we know that

$$\mathcal{P}(\hat{V} \neq V) \geq 1 - \frac{I(V; X) + \log 2}{\log(|\mathcal{V}|)} \geq 1 - \frac{\frac{8C\omega}{m^{2\alpha - 4}}}{m^d \log 2} \geq 1/2$$

At the same time, we can estimate the separation of the hypothesis in the two different norms:

- Deep Ritz Method:

$$\int_{[0,1]^d} \|\nabla u_k - \nabla u_{k'}\|^2 dx = \frac{\kappa^2}{m^{2\alpha - 2 + d}} \sum_{j \in [m]^d} (\tau_j^{(k)} - \tau_j^{(k')})^2 \int_{\mathbb{R}^d} \|\nabla g(x)\|^2 dx \gtrsim \frac{1}{m^{2\alpha - 2}}$$

- Physic Informed Neural Network:

$$\int_{[0,1]^d} \|\Delta u_k - \Delta u_{k'}\|^2 dx = \frac{\kappa^2}{m^{2\alpha - 4 + d}} \sum_{j \in [m]^d} (\tau_j^{(k)} - \tau_j^{(k')})^2 \int_{\mathbb{R}^d} \|\Delta g(x)\|^2 dx \gtrsim \frac{1}{m^{2\alpha - 4}}$$

Plug in $m = [n^{\frac{1}{2\alpha - 4 + d}}]$, we know that with constant probability we have

$$\inf_H \sup_{u \in W_s^\infty(\Omega)} \mathbb{E} \|H(\{X_i, Y_i\}_{i=1, \cdots, n}) - u^*(f)\|_{H_1}^2 \gtrsim n^{-\frac{2\alpha - 2}{d + 2\alpha - 4}} \log(n)^2, \tag{F.3}$$

$$\inf_H \sup_{u \in W_s^\infty(\Omega)} \mathbb{E} \|H(\{X_i, Y_i\}_{i=1, \cdots, n}) - u^*(f)\|_{H_2}^2 \gtrsim n^{-\frac{2\alpha - 4}{d + 2\alpha - 4}} \log(n)^2. \tag{F.4}$$

$\square$

