# OpenReview forum: "Statistical Numerical PDE : Fast Rate, Neural Scaling Law and When it’s Optimal"
_NeurIPS.cc/2021/Workshop/DLDE — DLDE Workshop -- NeurIPS 2021 Spotlight_

### Official Review · Reviewer_h9go · 2021-10-08

**Confidence:** 3

**Review:**

Deep Ritz Method (DRM) and Physics Informed Neural Networks (PINNs) are deep learning methods to solve PDEs. The authors took the same case setting as [11] and [25], which is the elliptic/Schrödinger equation with hypercube boundary condition. In this setting, they show that DRM and PINNs can achieve an optimal bound of O(1/n) instead of the previously best bound of O(1/sqrt(n)). Their upper bounds matches the lower bound (min-max bound). DRM does not achieve the improved bound, however they show how to modify DRM to achieve the same min-max bound. The authors provide experiments in simple settings to show that their bounds matches with experimental results.

Significance of the work:
Given that they derive the optimal min-max bounds of the two major deep learning methods to solve PDEs, further improvement is unlikely, which makes this work very important theoretically. The assumptions do not seem too strong. However, this bound only applies to a very specific scenario (Schrödinger equation with hypercube boundary condition). Their modified version of DRM could improve practical results, which is very useful for practitioners.

line 106: Section ??

**Score:**

5: Excellent paper: should definitely be a contributed talk

---

### Official Review · Reviewer_y8fa · 2021-10-11
**Review for Statistical Numerical PDE : Fast Rate, Neural Scaling Law and When it’s Optimal**

**Confidence:** 1

**Review:**

The authors study the limits of the Deep Ritz Method (DRM) and Physics Informed Neural Networks (PINN) for the Schrödinger equation with hypercube boundary conditions and obtain new min-max optimal bounds for PINNs and for an ad-hoc modified version of the DRM. Experimental results back up the claims. The authors also provide much detail and proof in the appendix.

This work can be an important contribution to the scientific machine learning community by rigorously providing new optimal bounds for deep PDE solvers.

Typos:
1. Abstract (OpenReview page):  Schrödinger appears as  Schr\"odinger; there is also citation leftover from Latex as \citep{duan2021convergence,jiao2021convergence}
2. Line 34: "optimiality"
3. Line 688 to 673: more than one instance of "Lemma ??"

**Score:**

4: Very good paper

---

### Official Review · Reviewer_NYN2 · 2021-10-14
**Understanding the limits of neural scaling for deep learning PDEs**

**Confidence:** 1

**Review:**

This paper studies the statistical limits of deep learning for solving PDEs for high-dimensional data.  The authors present a thorough theoretical and empirical evaluation of neural PDE solvers.  The paper presents the work in the context of known theoretical frameworks for evaluating the performance as a function of the input dimension.  They describe the implications of the results in practice by comparing them with existing neural PDE solvers.  The implications are important for understanding the limits of their use.

**Score:**

3: Good paper

---

### Decision · Program_Chairs · 2021-10-14

**Decision:**

Accept (Spotlight)

**Comment:**

The paper develops bounds on the optimality of DRMs and PINNs used to solve a class of PDEs. The reviewers were positive about this submission and its theoretical contributions.